# AutoFigure: Generating and Refining Publication-Ready Scientific Illustrations

**Minjun Zhu**[1,2]*, **Zhen Lin**[2]*, **Yixuan Weng**[2]*,

**Panzhong Lu**[2], **Qiujie Xie**[2], **Yifan Wei**[2], **Sifan Liu**[2], **Qiyao Sun**[2], **Yue Zhang**[2]✉

[1]Zhejiang University    [2]School of Engineering, Westlake University

wengsyx@gmail.com, {zhuminjun, zhangyue}@westlake.edu.cn

**Project:**https://deepscientist.cc ⬤ AutoFigure    ⬤ AutoFigure-Edit    🤗 FigureBench

## Abstract

High-quality scientific illustrations are crucial for effectively communicating complex scientific and technical concepts, yet their manual creation remains a well-recognized bottleneck in both academia and industry. We present **FigureBench**, the first large-scale benchmark for generating scientific illustrations from long-form scientific texts. It contains 3,300 high-quality scientific text–figure pairs, covering diverse text-to-illustration tasks from scientific papers, surveys, blogs, and textbooks. Moreover, we propose **AutoFigure**, the first agentic framework that automatically generates high-quality scientific illustrations based on long-form scientific text. Specifically, before rendering the final result, **AutoFigure** engages in extensive thinking, recombination, and validation to produce a layout that is both structurally sound and aesthetically refined, outputting a scientific illustration that achieves both structural completeness and aesthetic appeal. Leveraging the high-quality data from FigureBench, we conduct extensive experiments to test the performance of **AutoFigure** against various baseline methods. The results demonstrate that **AutoFigure** consistently surpasses all baseline methods, producing publication-ready scientific illustrations. The code, dataset and huggingface space are released in `https://github.com/ResearAI/AutoFigure`.

## 1 Introduction

Scientific illustration is a crucial medium for science communication, serving as a complement to scientific texts (Fytas et al., 2021; Kim et al., 2022). It allows readers to quickly grasp the main ideas within minutes and helps prevent misinterpretation (Chang et al., 2025). However, creating effective scientific illustrations is challenging. It requires a deep logical understanding of **long-form scientific texts**, along with the distillation of critical information. Additionally, the visual presentation must balance **structural fidelity and image quality**, ultimately transforming the text into clear, accurate, and aesthetically pleasing illustrations. As a result, producing a high-quality illustration usually takes human researchers several days, requiring creators to possess both domain knowledge and professional design skills.

Research into the **automatic generation of scientific illustrations from long-form scientific texts** could greatly enhance the efficiency and accessibility of science communication. However, this area remains largely unexplored. While previous datasets like Paper2Fig100k (Rodriguez et al., 2023b), ACL-Fig (Karishma et al., 2023), and SciCap+ (Yang et al., 2024a) have advanced the field, they primarily focus on reconstructing figures from captions, short snippets, or existing metadata. In contrast, our work targets **Long-context Scientific Illustration Design**, a task that requires distilling an entire methodology from a long document (avg. >10k tokens) and autonomously planning the visual

---

[1]* The contributions are equal.

[2]✉ Corresponding Author.

[3]AutoFigure is the complete implementation presented in this paper. AutoFigure-Edit is a new version of AutoFigure that supports fully editable icons and text.

structure, rather than simply translating explicit drawing instructions. Parallel to these limitations in benchmarks, existing automated systems also face challenges in generative capability. Although progress has been made in the field of automated generation of visual scientific communication (e.g., PosterAgent (Pang et al., 2025) and PPTAgent (Zheng et al., 2025)), these methods primarily focus on understanding, extracting, and combining existing multimodal content from papers, rather than **understanding** the original text and **generating** corresponding visual content. Another line of work employs executable code as an intermediate state between scientific text and illustration. (Belouadi et al., 2023; 2024; 2025; Ellis et al., 2018). These approaches primarily optimize for structural and geometric correctness. However, as demonstrated by our quantitative evaluations in , they often face challenges in balancing these rigid constraints with the aesthetic fluency and readability required for publication standards, resulting in lower scores compared to AUTOFIGURE in visual design metrics. Meanwhile, mainstream end-to-end text-to-image (T2I) models often fail to effectively visualize long scientific texts. Although they generate aesthetically pleasing images, they struggle to preserve **structural fidelity** (Liu et al., 2025). In Figure 6, we compare the generation results of the aforementioned methods when faced with long-form scientific texts. Taken together, these limitations underscore the challenges of directly transforming long scientific texts into illustrations that are both **accurate** and **visually appealing**.

To address these challenges, we introduce AUTOFIGURE, an agentic framework based on the **Reasoned Rendering** paradigm. This paradigm breaks down the scientific illustration generation process into two distinct stages: (1) **Semantic Parsing and Layout Planning**, converting the unstructured long-form scientific text into a structured, machine-readable conditioning image with an associated style description. (2) **Aesthetic Rendering and Text Refinement**, which transforms the structurally optimized symbolic blueprint into a high-fidelity illustration, while addressing the common problem of blurry text rendering through an "erase-and-correct" strategy. We further propose a large-scale benchmark named **FigureBench** (Figure 1) to comprehensively evaluate the quality of the AI-generated scientific illustrations. It consists of **3,300** high-quality long-form scientist text–figure pairs, with 300 reserved as the test set and the remaining as the development set. For the critical test set, we randomly sample 400 papers from Research-14K (Weng et al., 2025) and extract the most relevant conceptual illustrations using GPT-5. After sixteen days of human annotation, 200 high-quality pairs are retained with a high Inter-Rater Reliability (IRR, Cohen's $\kappa = \textbf{0.91}$)[1]. To further enhance the diversity, an additional 100 samples are curated from scientific surveys, blogs, and textbooks, yielding **300 test instances** in total. Leveraging these expert-labeled data, we further finetune an automated filter to construct a large-scale development set comprising **3,000 illustration pairs**.

Finally, based on FigureBench, we design an evaluation protocol grounded in the **VLM-as-a-judge** paradigm. It combines **referenced scoring** and **blind pairwise comparison** to assess AI-generated scientific illustrations across multiple dimensions (e.g., aesthetic quality, accuracy). Through extensive quantitative and qualitative evaluations, including automated evaluations (§5.1), human evaluation (§5.2), and controlled ablation studies (§5.3), we demonstrate that AUTOFIGURE effectively resolves the trade-off between aesthetic fluency and structural fidelity. The generated scientific illustrations not only maintain high accuracy in structure and text but also achieve publication quality in layout and visual appeal. Qualitative examples are presented in Figures 6, 3 and Appendix Section E, showcasing the versatility of AUTOFIGURE in generating complex scientific illustrations (e.g., procedural flows, algorithmic pipelines) from a diverse range of academic texts.

In this paper, we construct FigureBench, the first large-scale benchmark specifically targeting Long-context Scientific Illustration Design, and propose a novel framework named AUTOFIGURE. With this framework, we achieve **the fully automated generation of high-quality scientific illustrations**. The effectiveness of AUTOFIGURE is also strongly validated by human expert evaluations, with up to **66.7%** of generated results judged to meet publication standards (Figure 4). We hope this work can provide researchers with a powerful automation tool, lay a solid foundation for the development of automatic scientific illustration models, and endow future "AI scientists" with excellent visual expression capabilities.

---

[1]Cohen's $\kappa$ assesses annotator agreement beyond random chance on labeling tasks.

**Table 1:** Comprehensive Analysis of the FigureBench.

| Category | Number (Total) | Text Tokens (Avg.) | Text Density (%, Avg.) | Components (Avg.) | Colors (Avg.) | Shapes (Avg.) |
|---|---|---|---|---|---|---|
| Paper | 3200 | 12732 | 42.1 | 5.4 | 6.4 | 6.7 |
| Blog | 20 | 4047 | 46.0 | 4.2 | 5.5 | 5.3 |
| Survey | 40 | 2179 | 43.8 | 5.8 | 7.0 | 6.7 |
| Textbook | 40 | 352 | 25.0 | 4.5 | 4.2 | 3.4 |
| **Total/Average** | **3300** | **10300** | **41.2** | **5.3** | **6.2** | **6.4** |

## 2 RELATED WORK

**Automated Scientific Visuals Generation.** Existing work on automated scientific visuals primarily explores the generation of artifacts like posters and slides. Modern agentic systems such as PosterAgent (Pang et al., 2025) and PPTAgent (Zheng et al., 2025) have advanced significantly beyond early summarization techniques (Qiang et al., 2019; Xu & Wan, 2022; Hu & Wan, 2014; Sravanthi et al., 2009). However, these systems are fundamentally designed to rearrange and summarize existing figures and textual content from a source document. Moreover, existing schematic-focused works such as SridBench (Chang et al., 2025) and FigGen (Rodriguez et al., 2023a) are often limited by their reliance on sparse inputs, such as captions, which lack sufficient structural information. Our work, instead, addresses the task of generating scientific illustrations from scratch based on a **long-form scientific context**, a critical step towards producing a complete and original scientific artifact.

**Text-to-Image Generation.** Recent progress in diffusion models (Song et al., 2021) have greatly improved the performance of T2I generation (Saharia et al., 2022; Ramesh et al., 2022). While text-based conditioning provides flexibility and user-friendliness, current models face particular challenges when dealing with scientific long-form texts, which often contain specialized terminology, complex structures, and intricate relationships between concepts. These texts not only span multiple sentences and hundreds of tokens but also require a deep understanding of domain-specific knowledge (Zheng et al., 2024). Effectively encoding such lengthy and detailed conditions, while ensuring precise alignment between the scientific text and generated images, remains a critical gap for generative models (Liu et al., 2025; Chen et al., 2024). To address this gap, we propose **FigureBench** for systematic evaluation and **AUTOFIGURE** for advancing automated scientific illustration.

**Automated Scientific Discovery.** The rise of AI Scientists (Lu et al., 2024; Yamada et al., 2025; Intology, 2025; Zhu et al., 2025b; Xie et al., 2025c), powered by Large Language Models (LLMs), is revolutionizing scientific discovery by autonomously managing the entire research workflow (Xie et al., 2025b; Starace et al.; Weng et al., 2026; Chan et al.; Wang et al., 2024a). This shift is substantiated by the growing acceptance of AI-generated papers at prestigious venues. For instance, manuscripts generated by the AI Scientist-v2 (Yamada et al., 2025) exceeded human acceptance thresholds at ICLR 2025 workshops, and Zochi (Intology, 2025) successfully authored papers accepted into the main proceedings of ACL 2025. This is further complemented by significant progress in producing textual artifacts such as reviews and surveys (Zhu et al., 2025a; Wang et al., 2024b). These developments signal that a human-level AI capable of uncovering novel phenomena may be imminent. However, this progress has exposed a critical bottleneck, as the inability to generate illustrations prevents AI Scientists from visually articulating their own findings. Automating this capability is the essential next step, empowering these systems to translate complex, machine-generated discoveries into an intuitive visual language that is fully comprehensible to human researchers.

## 3 FIGUREBENCH: A BENCHMARK FOR AUTOMATED SCIENTIFIC ILLUSTRATION GENERATION

Automatic scientific illustration aims to constructs a mapping function $G$ that takes long-form scientific text $T$ as input and generates a publication-quality illustration $I_{final}$. In this section, we introduce **FigureBench**, the first large-scale benchmark for generating scientific illustrations from long-form scientific texts. As depicted in Figure 1, FigureBench is curated to encompass a wide array of document types, including research papers, surveys, technical blogs, and textbooks, establishing a challenging and diverse testbed to spur research in automatic scientific illustration generation.

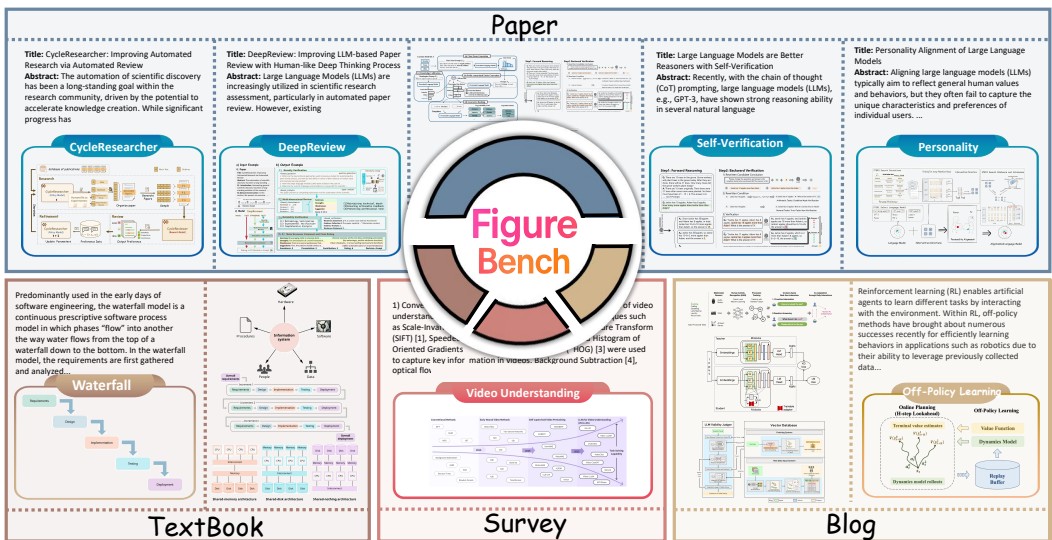

**Figure 1:** The composition of the FigureBench dataset. It features a rich collection of text-figure pairs from four distinct sources (Paper, Survey, Blog, and TextBook), demonstrating the benchmark's capability to evaluate automatic illustration generation across various domains and complexities.

**Data Curation.** To curate a high-quality test set for this task, we began by randomly sampling 400 scientific papers from the Research-14K dataset (Weng et al., 2025). For each paper, we used GPT-5 to select the single illustration that best represented its core methodology. This resulted in 400 initial paper-figure pairs. We then filtered these pairs, retaining only conceptual illustrations (i.e., excluding data-driven charts) where each key visual element was explicitly described in the source text. To ensure high quality and consistency, each remaining pair was evaluated by two independent annotators. Only pairs that were approved by both annotators were included in the final dataset. This rigorous annotation process yielded a high Inter-Rater Reliability (IRR) of 0.91, resulting in a final test set of 200 high-quality samples.

To further enhance the diversity of our test data, we manually curated an additional 100 samples from three distinct sources: surveys, technical blogs, and textbooks. For the survey category, we collected structural diagrams (e.g., roadmaps and taxonomies) from recent AI surveys published on arXiv. Textbook samples were sourced from open-licensed educational platforms like OpenStax for their pedagogical clarity, while blog samples were hand-collected from technical outlets such as the ICLR Blog Track to capture modern and accessible visual styles. The entire curation process strictly adhered to open-source licenses, with a detailed breakdown provided in Appendix A. Finally, we leveraged our high-quality curated set of 300 samples (200 from papers and 100 from diverse sources) to fine-tune a vision-language model. This model then served as an automated filter, which we applied to the larger Research-14K corpus (Weng et al., 2025), resulting in **a large-scale development set containing 3,000 scientific illustration samples**.

We explicitly distinguish the roles of these datasets: the Test Set is strictly reserved for evaluation, whereas the Development Set is designed for training, development and experimental purposes. Although AUTOFIGURE operates as an inference-only pipeline and does not utilize the Development Set for training, we provide this resource to facilitate future exploration of end-to-end or trainable methods by the community.

**Dataset Analysis.** To quantify the characteristics of FigureBench, we conduct a detailed statistical analysis, presented in Table 1. The analysis confirms the task's significant challenges. For instance, the average Text Tokens metric varies by over an order of magnitude between Textbooks (352) and Papers (12,732), highlighting the need for robust long-context reasoning. Additionally, the high average Text Density (41.2%), which indicates the proportion of the image area occupied by text, and the varied number of Colors (averaging 6.2), both statistically analyzed using the InternVL 3.5 model (Wang et al., 2025), underscore the challenge of balancing informational richness with visual clarity. Moreover, the mean number of Components (5.3) and Shapes (6.4) demonstrates the structural complexity. The collected paper data is also temporally recent from to 2025.

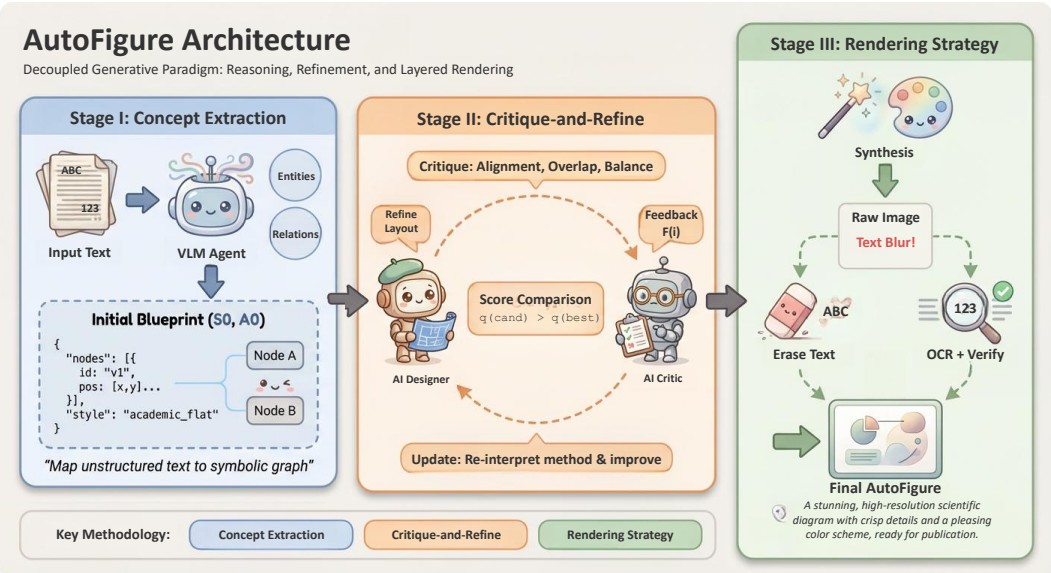

**Figure 2:** An Overview of the AUTOFIGURE, which decouples structural layout generation from aesthetic rendering. Stage 1 ensures structural fidelity by having a multi-agent system generate and iteratively self-correct a symbolic layout (SVG). Stage 2 renders the validated layout and employs an erase-and-correct module—using OCR and cross-verification—to guarantee perfect textual accuracy with high-fidelity vector overlays. **This figure is also produced by AUTOFIGURE and serves as a qualitative showcase of its generation quality.**

**Evaluation Metrics.** The evaluation of scientific illustrations is non-trivial, as traditional T2I metrics (e.g., FID (Jayasumana et al., 2024)) are usually misaligned with the requirements for logical and topological correctness. Therefore, our evaluation protocol leverages the **VLM-as-a-judge paradigm** for structural reasoning and long-context comprehension, consisting of two complementary methods: (1) **Referenced scoring**, where a VLM is provided with the full text, the ground-truth figure, and the generated image. It assesses the generated image across three dimensions with eight sub-metrics: **Visual Design** (aesthetic quality, visual expressiveness, professional polish), **Communication Effectiveness** (clarity, logical flow), and **Content Fidelity** (accuracy, completeness, appropriateness). The VLM outputs a score and textual reasoning for each sub-dimension, with the Overall score calculated as their average. (2) **blind pairwise comparison.** In this evaluation setting, the VLM receives the full text and two images (ground-truth and generated) in a randomized order, without knowledge of which is the original. It is asked to select a winner (A, B, or Tie) based on seven criteria including aesthetic quality, clarity, information sophistication, accuracy, completeness, appropriateness, and provide a final choice for the better figure. We note that VLM-as-a-judge paradigm can not fully replace human expertise (Lee et al., 2024; Xie et al., 2025a). To this end, we further conduct an **human evaluation**, for which we recruit ten first-authors to assess generated figures for their own work (§5.2), providing a gold-standard measure of real-world utility.

## 4 AUTOFIGURE

We introduce **AUTOFIGURE**, a **decoupled generative paradigm** for high-fidelity scientific illustration generation. Our approach tackles the challenge of producing **semantically accurate** and **visually coherent** figures by separating the reasoning and rendering processes. Our core innovation lies in a three-stage pipeline. First, we employ a large language model (LLM) for **conceptual grounding**, distilling unstructured text into a structured, symbolic blueprint. Second, a novel self-refinement loop—simulating a dialogue between an AI designer and critic—iteratively optimizes this blueprint for structural coherence and logical consistency. Finally, a dedicated rendering stage, featuring a unique erase-and-correct strategy to ensure textual legibility. The following sections detail each phase of this pipeline.

## 4.1 STAGE I: CONCEPTUAL GROUNDING AND LAYOUT GENERATION

Given a long-form scientific document $T$, Stage I produces (i) a machine-readable symbolic layout $S_0$ (e.g., SVG/HTML) that specifies the 2D geometry and topology of the schematic, and (ii) a style descriptor $A_0$. We additionally rasterize $S_0$ into a layout reference image $I_0$ that will be used to condition the renderer in Stage II.

**Concept Extraction and Symbolic Construction.** Given the input text $T$, the Concept-Extraction agent outputs (a) a distilled methodology summary $T_{\text{method}}$ and (b) a set of entities and relations that will be visualized as nodes and directed edges. We serialize this structure into a markup-based symbolic layout $S_0$ (SVG/HTML) and a category-conditioned style description $A_0$, where $C \in \{\text{PAPER}, \text{SURVEY}, \text{BLOG}, \text{TEXTBOOK}\}$ and $S_0$ encodes a directed graph $G_0 = (V_0, E_0)$. All stage prompts and the exact input–output schema are provided in Appendix M for reproducibility.

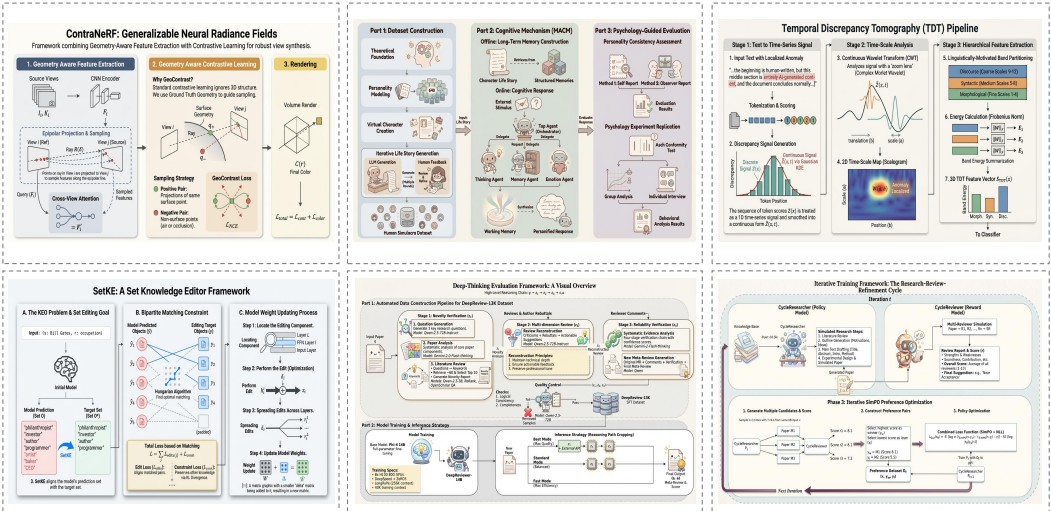

**Figure 3:** Examples showcasing the versatility of AUTOFIGURE in generating complex scientific illustrations from a diverse range of academic texts. Note that we employ a unified default style (*Delicate and cute cartoon comic style, using Morandi color palette*) solely to ensure visual consistency and readability for comparative analysis. This is a choice of presentation rather than a limitation of the method; users can freely specify or mix arbitrary styles as needed (see in Appendix L). We present a diverse range of results in Appendix Q to further illustrate our approach.

**Critique-and-Refine.** This step is the core of our "thinking" process, implementing a **self-refinement loop** that simulates a dialogue between an AI "designer" and an AI "critic", aiming to find the globally optimal layout through iterative search. First, the initial layout $(S_0, A_0)$ is evaluated to get an initial score $q_0$, which is set as the current best version: $(S_{best}, A_{best}) \leftarrow (S_0, A_0)$ and $q_{best} \leftarrow q_0$. Subsequently, in each iteration $i$, the system attempts to generate a superior solution:

$$F_{best}^{(i)} = \text{Feedback}(\Phi_{critic}(S_{best}, A_{best})) \tag{1}$$

$$\left(S_{cand}^{(i)}, A_{cand}^{(i)}\right) = \Phi_{gen}(T_{method}, F_{best}^{(i)}), \tag{2}$$

where the critic $\Phi_{critic}$ evaluates the best-performing layout $(S_{best}, A_{best})$ for alignment, balance, and overlap avoidance, producing textual feedback $F_{best}^{(i)}$. The generator $\Phi_{gen}$ then then uses this feedback to reinterprets $T_{method}$ and produce a candidate layout $(S_{cand}^{(i)}, A_{cand}^{(i)})$ with score $q_{cand}^{(i)}$. If $q_{cand}^{(i)} > q_{best}$, it replaces the current best. The loop continues until a preset limit of N iterations or until the score converges, yielding the final layout $(S_{final}, A_{final})$, a logically consistent, structurally coherent, and aesthetically balanced conditioning layout with style description.

**Table 2:** A comprehensive user evaluation across four generation tasks, with updated methods and scoring. Win-Rate is calculated through blind pairwise comparisons against the reference, indicating the percentage of times a method is selected as producing more suitable illustrations for the descriptive text.

| Method | Visual Design | | | Communication | | Content Fidelity | | | Overall | Win-Rate |
|---|---|---|---|---|---|---|---|---|---|---|
| | Aesthetic | Express. | Polish | Clarity | Flow | Accuracy | Complete. | Appropriate. | | |
| **BLOG** | | | | | | | | | | |
| HTML-Code | 5.61 | 4.50 | 5.79 | 7.42 | 7.53 | 7.26 | 6.34 | 6.76 | 6.40 | 30.0% |
| SVG-Code | 4.39 | 3.61 | 4.09 | 5.68 | 5.71 | 5.98 | 5.05 | 5.17 | 4.96 | 45.0% |
| GPT-Image | 3.80 | 3.00 | 3.60 | 5.83 | 5.70 | 4.62 | 3.92 | 4.67 | 4.39 | 10.0% |
| Diagram Agent | 1.95 | 1.47 | 1.61 | 2.16 | 2.05 | 2.34 | 1.76 | 2.00 | 1.92 | 0.0% |
| AUTOFIGURE | **7.53** | **7.25** | **7.44** | **8.04** | **8.38** | **7.32** | **6.65** | **8.23** | **7.60** | **75.0%** |
| **SURVEY** | | | | | | | | | | |
| Gemini-HTML | 4.77 | 3.59 | 4.88 | 6.99 | 6.52 | **8.04** | 7.04 | 5.55 | 5.92 | 37.5% |
| Gemini-SVG | 4.28 | 3.16 | 4.25 | 6.51 | 6.06 | 7.25 | 6.16 | 5.04 | 5.34 | 44.1% |
| GPT-Image | 3.65 | 2.85 | 3.71 | 6.28 | 5.79 | 5.87 | 4.59 | 4.26 | 4.63 | 17.5% |
| Diagram Agent | 2.11 | 1.55 | 1.77 | 2.69 | 2.67 | 2.86 | 2.06 | 2.07 | 2.22 | 0.0% |
| AUTOFIGURE | **6.91** | **6.31** | **6.65** | **7.50** | **7.44** | 7.54 | **6.75** | **6.83** | **6.99** | **78.1%** |
| **TEXTBOOK** | | | | | | | | | | |
| Gemini-HTML | 5.36 | 4.24 | 5.31 | 7.49 | 7.09 | 8.29 | 7.75 | 6.75 | 6.53 | 72.5% |
| Gemini-SVG | 4.90 | 3.99 | 4.81 | 6.91 | 6.74 | 8.03 | 7.33 | 6.28 | 6.12 | 76.9% |
| GPT-Image | 4.60 | 4.07 | 4.53 | 6.98 | 6.60 | 6.83 | 5.93 | 5.85 | 5.67 | 55.0% |
| Diagram Agent | 2.03 | 1.51 | 1.63 | 2.51 | 2.20 | 3.24 | 2.73 | 2.17 | 2.25 | 0.0% |
| AUTOFIGURE | **7.51** | **7.33** | **7.21** | **8.13** | **8.27** | **8.69** | **8.22** | **8.64** | **8.00** | **97.5%** |
| **PAPER** | | | | | | | | | | |
| HTML-Code | 5.90 | 5.04 | 5.84 | 7.17 | 7.38 | **6.99** | 6.37 | 6.15 | 6.35 | 11.0% |
| SVG-Code | 5.00 | 4.19 | 4.89 | 6.34 | 6.48 | 6.15 | 5.53 | 5.37 | 5.49 | 31.0% |
| GPT-Image | 4.24 | 3.47 | 4.00 | 5.63 | 5.63 | 4.77 | 4.08 | 4.25 | 4.51 | 7.0% |
| Diagram Agent | 2.25 | 1.73 | 2.04 | 2.67 | 2.49 | 2.11 | 1.72 | 1.94 | 2.12 | 0.0% |
| AUTOFIGURE | **7.28** | **6.99** | **6.92** | **7.34** | **7.87** | 6.96 | **6.51** | **6.40** | **7.03** | **53.0%** |

## 4.2 STAGE II: AESTHETIC SYNTHESIS AND TEXT POST-PROCESSING

The final stage translates the structurally optimized symbolic blueprint $(S_{final}, A_{final})$ into a high-fidelity illustration $I_{final}$.

**Style-Guided Aesthetic Rendering.** We use a transformation function $\Phi_{prompt}$ (LLM-based) to convert the $(S_{final}, A_{final})$ into an exhaustive text-to-image prompt, paired with a structural graph derived from $S_{final}$ (which precisely dictates the position and interconnection of all elements). These inputs are fed into a multimodal generative model to render an image $I_{polished}$ that is faithful to the layout structure and perfectly embodies the optimized aesthetic style.

**Ensuring Textual Accuracy.** We improve text legibility via an erase-and-correct process. First, a non-LLM eraser $\Phi_{erase}$ removes all text pixels from $I_{polished}$ to produce a clean background $I_{erased} = \Phi_{erase}(I_{polished})$. Second, a OCR engine $\Phi_{ocr}$ extracts preliminary strings and bounding boxes $(T_{ocr}, C_{ocr}) = \Phi_{ocr}(I_{polished})$. Third, a multimodal verifier $\Phi_{verify}$ aligns each OCR string with the ground-truth labels $T_{gt}$ parsed from $S_{final}$ and outputs a corrected text map $T_{corr} = \Phi_{verify}(T_{ocr}, T_{gt})$. Finally, we render $T_{corr}$ as vector-text overlays at $C_{ocr}$ on top of $I_{erased}$ to obtain $I_{final}$.

## 5 EXPERIMENTS

To comprehensively evaluate AUTOFIGURE, we conduct (i) automated evaluations on FigureBench (§ 5.1), (ii) a domain-expert study with paper-authors (§ 5.2), and (iii) controlled ablations isolating key modules (§ 5.3). Figure 3 provides representative qualitative results; beyond these in-text examples, the appendix contains substantially extended evidence, including: detailed qualitative case studies across diverse papers (Appendix § E); additional evaluations under open-source model deployments (Appendix G); an extended blind pairwise comparison with absolute quality judgments (Appendix H); module-level analyses of the text-refinement/post-processing component (Appendix I); efficiency and cost analysis under different deployment settings (Appendix J); a human-audited sanity check of automated dataset statistics (Appendix K); further results on style controllability and diversity (Appendix L); and further results on extended baselines(Appendix N).

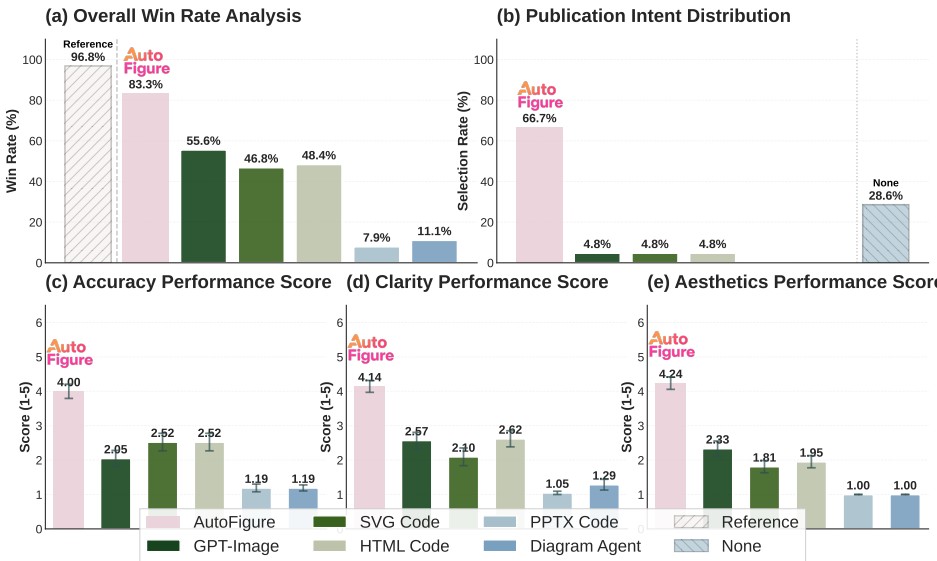

**Figure 4:** Human evaluation results from 10 first-author experts assessing AI-generated figures for 21 of their own publications. The comprehensive study required experts to perform three tasks: (a) a forced-choice holistic ranking of six AI models against the original reference to determine a win rate, (b) a publication intent selection, and (c-e) multi-dimensional scoring on a 1-5 Likert scale for accuracy, clarity, and aesthetics.

## 5.1 AUTOMATED EVALUATIONS

**Experimental Setup.** We assess AUTOFIGURE against three distinct types of baseline methods: (1) End-to-end text-to-image methods (Sun et al., 2024), where we used the GPT-Image model (Hurst et al., 2024) to directly generate a scientific schematic from the paper's text based on instructions; (2) Text-to-code methods, where we use a LLM to generate corresponding HTML Code and SVG Code (Rodriguez et al., 2025; Malashenko et al., 2025; Yang et al., 2024b), which are then automatically rendered into images; and (3) Multi-agent frameworks, represented by Diagram Agent (Wei et al., 2025), which automates workflow design. For AUTOFIGURE and other decoupled baselines, we use Gemini-2.5-Pro as the sketch model and GPT-Image as the rendering model.

As detailed in Table 2, AUTOFIGURE achieves the highest Overall score across all four document categories: Blog (7.60), Survey (6.99), Textbook (8.00), and Paper (7.03). Notably, AUTOFIGURE also dominates in Win-Rate evaluations through blind pairwise comparisons, achieving 75.0% for Blog, 78.1% for Survey, an exceptional 97.5% for Textbook, and 53.0% for Paper. The results demonstrate that **AUTOFIGURE consistently surpasses all baseline methods in both automated evaluations and human preferences**, showcasing a superior balance of visual quality, communicative effectiveness, and content fidelity. Moreover, AUTOFIGURE achieves the best performance in most sub-metrics under Visual Design Excellence and Communication Effectiveness, indicating its ability to produce schematics that are both attractive and easy to understand. The Win-Rate results particularly highlight the inherent limitations of existing paradigms: code-generation methods (Gemini-HTML/SVG) achieve moderate Win-Rates (30-77%) but sacrifice visual aesthetics for structural control, while the end-to-end model GPT-Image shows consistently low Win-Rates (7-55%) due to poor content accuracy. For instance, in the Paper category, the Aesthetic scores of text-to-code methods (5.90 and 5.00) are significantly lower than AUTOFIGURE's (7.28), limiting their Win-Rates to 11.0% and 31.0% respectively. Conversely, GPT-Image exhibits critical weakness in content accuracy, scoring the lowest among generative models in this metric for the Paper category (4.77), resulting in only 7.0% Win-Rate. The multi-agent framework, Diagram Agent, consistently achieves 0% Win-Rate across all categories while performing poorly in all dimensions, underscoring the profound difficulty of this task without a specialized, structured approach.

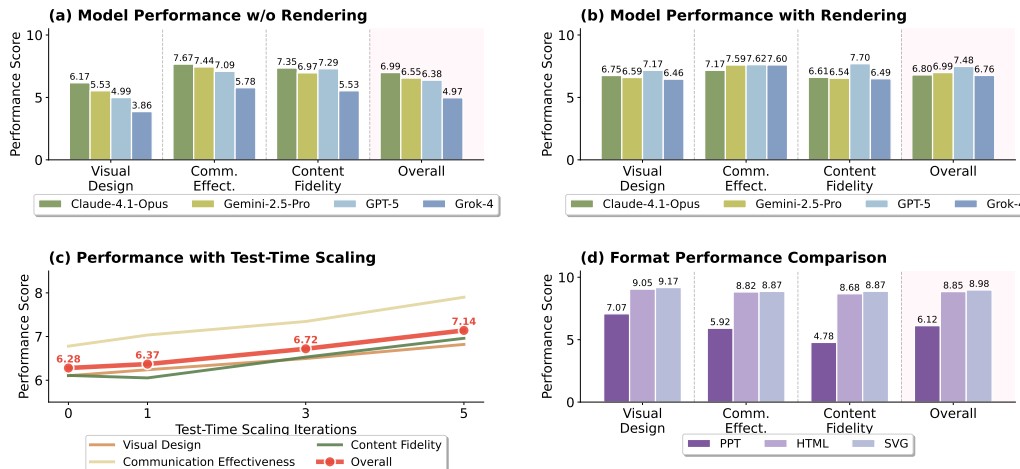

**Figure 5:** Ablation studies of the AUTOFIGURE framework. Subplots compare different backbone models on (a) pre-rendering symbolic layouts versus (b) final rendered outputs. Also shown are (c) performance scaling with increased test-time refinement iterations and (d) the impact of different intermediate sketch formats.

## 5.2 HUMAN EVALUATION WITH DOMAIN EXPERTS

**Experimental Setup.** To evaluate whether the figures generated by AUTOFIGURE meet the publication-ready standards of the relevant domain experts, we recruited 10 human experts to assess AI-generated figures based on their own first-author publications. The evaluation involved three tasks across 21 high-quality papers: (1) Multi-dimensional scoring: Each figure was rated on a 1–5 Likert scale for Accuracy, Clarity, and Aesthetics. (2) Forced-choice ranking: Experts ranked all AI-generated figures against the original human-authored references. (3) Publication intent selection: Experts indicated which figures they would choose to include in a camera-ready paper.

The results are shown in Figure 4, showing that **AUTOFIGURE's quality is judged far superior to other AI systems and closely approaches the human-created originals** by resolving the key trade-off between accuracy and aesthetics. The win-rate analysis in Figure 4(a) shows AUTOFIG-URE achieves an 83.3% win rate against oth er models, second only to the original human-authored reference (96.8%). Notably, as shown in Figure 4(b), 66.7% of experts are willing to adopt figures generated by AUTOFIGURE for a camera-ready version of their own papers, indicating that it can produce figures that meet the standards of real-world academic publishing. In contrast, the performance of baseline methods is highly polarized. GPT-Image achieves better aesthetics at the cost of low accuracy, while SVG Code has slightly better accuracy but poor aesthetics.

## 5.3 ABLATION STUDIES

**Analysis on pre-rendering symbolic layouts.** By comparing the pre-rendering scores in Figure 5(a) with the post-rendering scores in Figure 5(b), we observe a consistent and significant improvement in Visual Design and Overall scores for all backbone models. For instance, with GPT-5 as the reasoning core, the Overall score jumps from 6.38 to 7.480 after rendering. This proves the effectiveness of the final drawing phase. The decoupled rendering stage effectively enhances visual appeal without compromising the schematic's structural integrity and content fidelity.

**Analysis on refinement loop.** To investigate the impact of the critique-and-refine loop, we conduct a test-time scaling experiment that fixes the backbone models and varies the number of "thinking" iterations from 0 to 5. As shown in Figure 5(c), the overall performance score steadily rises from an initial 6.28 (zero iterations) to 7.14 after five iterations. This improvement demonstrates that the refinement loop is an effective optimization process.

**Analysis on reasoning models and intermediate formats.** The figure quality is also heavily influenced by the choices of the reasoning model and the intermediate data format. Figures 5(a) and 5(b) show that stronger reasoning models like Claude-4.1-Opus produce superior layouts compared

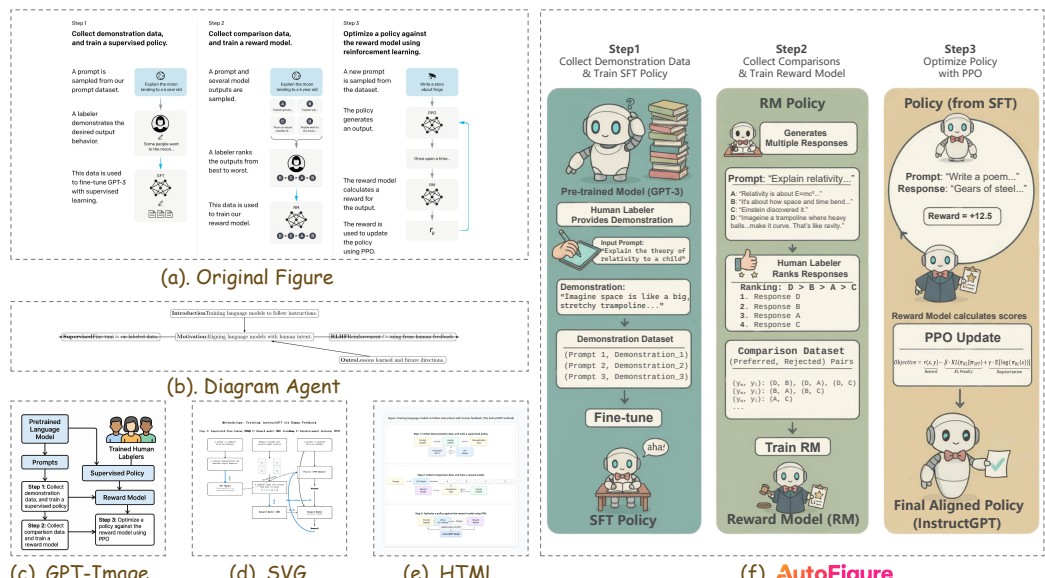

**Figure 6:** Qualitative comparison for generating a scientific illustration of the InstructGPT framework (Ouyang et al., 2022). We compare the original human design (a) with illustrations generated by various baseline methods (b-e) and our AUTOFIGURE (f). Notably, AUTOFIGURE achieves both high scientific fidelity and aesthetic appeal, yielding a publication-ready result.

to others. Furthermore, Figure 5(d) highlights the critical role of the intermediate representation, as expressive and structured formats like SVG (8.98) and HTML (8.85) can generate the entire figure in one coherent file, while PPT's (6.12) requirement for multiple incremental code insertions introduces inconsistencies that cause the final output to diverge from the original paper's content.

**Case study.** This case highlights AUTOFIGURE's advantage in jointly preserving *semantic structure* and *visual readability* for multi-stage procedural diagrams. Compared with the original figure 6 (a), DIAGRAM AGENT in Figure 6 (b) collapses the process into an overly thin, low-information chain, losing the essential stage separation and the data artifacts (demonstrations, comparisons, reward scores) that define the RLHF workflow. The end-to-end generator GPT-IMAGE in Figure 6 (c) captures only a coarse flow and exhibits inconsistent typography and crowded labeling, which undermines legibility and instructional value. Code-only baselines Figure 6 (d–e) better maintain a box-and-arrow skeleton, but the outputs remain visually sterile and fragmented (e.g., weak hierarchy, poor spacing, and limited affordances to emphasize key roles such as labelers and reward modeling). In contrast, AUTOFIGURE in Figure 6 (f) explicitly decomposes the content into three aligned stages (SFT, RM, PPO), renders consistent typographic hierarchy and spacing, and uses semantically grounded icons and callouts to make roles and data transformations immediately interpretable, yielding a polished infographic without sacrificing the core scientific fidelity.

## 6 CONCLUSION

Generating high-quality scientific illustrations from long-form scientific texts poses new challenges for existing automated scientific visuals generation technologies. To advance this field, this paper introduced FigureBench, a comprehensive benchmark consisting of 3,300 high-quality long-form scientist text–figure pairs, covering diverse types of scientific texts. Building upon FigureBench, we further proposed AUTOFIGURE, an agentic framework based on the Reasoned Rendering paradigm, which generates accurate and visually appealing illustrations in an iterative process. Through automatic evaluations grounded in the VLM-as-a-judge paradigm and human expert assessments, we demonstrated that AUTOFIGURE's ability to generate scientifically rigorous and aesthetically appealing illustrations that meet the standards of academic publishing. By automating a critical bottleneck in scientific communication, our work lays the groundwork for AI-driven scientific visual expression, enabling more efficient and accessible creation of publication-ready illustrations.

## ACKNOWLEDGEMENT

This publication has been supported by the National Natural Science Foundation of China (NSFC) Key Project under Grant Number 62336006 and 625B2152, and the 2025 Dean's Special "PhD Student Project" of the School of Engineering, Westlake University.

We sincerely thank Yiran Ding, Zhiyuan Ning, Fang Guo, Guangsheng Bao, Hongbo Zhang, and Shulin Huang for their contributions to this paper.

## ETHICS STATEMENT

We acknowledge the significant ethical considerations associated with powerful generative technologies like AUTOFIGURE. The primary risk involves the potential for misuse, where the system could be used to generate scientifically plausible but factually incorrect or misleading schematics to support false claims. To mitigate this risk, we are committed to a policy of transparency and responsible deployment. Our mitigation strategy is twofold. First, we explicitly declare that AUTOFIGURE is an assistive tool with limitations. This disclaimer, stating that the system is not a substitute for expert verification and may not produce perfectly reliable outputs, will be placed prominently within this paper and in the README file of the public code repository. Second, the open-source license governing AUTOFIGURE will include a mandatory attribution clause. This clause requires any academic publication using a figure generated by our tool to (a) include a specific section that discusses the role AI played in the work, and (b) explicitly caption the figure as having been generated by AUTOFIGURE. These requirements are designed to ensure transparency and accountability in the downstream use of our technology, fostering a research environment where AI tools are used to augment, not compromise, scientific integrity.

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

## A    DATA CURATION DETAILS FOR FIGUREBENCH

The curation of the FigureBench dataset was guided by a rigorous, multi-stage pipeline designed to ensure high quality, relevance, and full adherence to open-source principles. Our process began with the Research-14K dataset, from which we exclusively selected papers governed by permissive licenses (e.g., Research-Dataset-License and CC BY 4.0, as detailed in Table 3). This step ensured that all subsequent annotation and redistribution activities complied with the original authors' terms. From this licensed subset, an initial filtering pass using GPT-5 identified approximately 400 candidate text-figure pairs likely to contain high-quality schematic diagrams. To validate these candidates for our gold-standard set, we utilized an online annotation platform and two expert annotators with backgrounds in machine learning. Each annotator independently judged whether a figure accurately and effectively represented the core methodology of its paper. A stringent unanimous approval policy was enforced: a figure was accepted as a positive sample only if both annotators gave their approval. This process yielded 200 high-quality positive samples, which form the core of our test set, and 200 verified negative samples (those not unanimously approved) suitable for training discriminator models.

## B    HUMAN EVALUATION PROTOCOL

To rigorously assess the practical utility of AUTOFIGURE and other baseline models, we designed a comprehensive human evaluation study hosted on a custom annotation website. We recruited 10 annotators who have all previously published as first authors on academic papers in the computer

**Table 3:** Data source details for FigureBench

| Data Type | Source | License | Number |
|---|---|---|---|
| paper | Research-14k | Research-Dataset-License&CC BY 4.0 | 3200 |
| survey | Arxiv | CC BY | 40 |
| blog | CMU ML Blog + The BAIR Blog + ICLR Blogposts 2025 | CC BY 4.0 | 20 |
| textbook | OpenStax: *Foundations of Computer Science* OpenStax: *Information Systems* | CC BY 4.0 CC BY-NC-SA 4.0 | 40 |

science field. To ground their evaluation in a familiar context, each expert was presented with figures generated for their own past publications. The evaluation was structured into three distinct tasks, compelling the experts to assess the quality of outputs from multiple perspectives: multi-dimensional scoring, holistic ranking, and a publication-readiness selection.

### B.1 TASK 1: MULTI-DIMENSIONAL QUALITY SCORING

The initial task required experts to perform a detailed, multi-dimensional quality assessment of figures generated by six different AI models: AUTOFIGURE, Diagram Agent, GPT-Image, HTML Code, PPTX Code, and SVG Code. For each of the six generated figures, evaluators were asked to provide a rating on a five-star scale across three key dimensions: Accuracy, Clarity, and Aesthetics. The **Accuracy** dimension measured how faithfully the figure represented the core concepts, relationships, and technical details described in the original paper. A one-star rating indicated a complete deviation from the paper's content, while a five-star rating signified a perfect representation.

The **Clarity** dimension assessed the figure's legibility and effectiveness in communicating information, considering factors such as text readability, the completeness of legends, and the logic of the layout. A one-star rating was for figures that were confusing and difficult to interpret, whereas a five-star rating was for those that were immediately understandable. Finally, the **Aesthetics** dimension focused on the visual design quality, including the harmony of the color scheme, the refinement of graphical elements, and its overall professional appearance suitable for academic publication. The user interface facilitated this process with interactive star-rating components for each of the 18 required judgments (6 models × 3 dimensions), ensuring that evaluators provided a complete set of scores before proceeding.

### B.2 TASK 2: HOLISTIC COMPARATIVE RANKING

Following the detailed scoring, the second task asked evaluators to perform a holistic ranking of all figures. This task was designed to move beyond dimensional analysis and capture an overall judgment of quality. Crucially, the set of items to be ranked included not only the six AI-generated figures but also the original, human-created figure from the publication, serving as a ground-truth reference. Evaluators were instructed to consider all aspects of quality, including the previously scored dimensions as well as less tangible factors like innovativeness and overall fitness for the paper's context.

The ranking was implemented through a drag-and-drop interface where each figure was displayed on a movable card. These cards showed the model's name (or "Reference" for the original), a preview of the figure, and its current rank from 1 to 7. Experts could adjust the order by dragging the cards or using "move up" and "move down" buttons, with the interface providing smooth animations to reflect the real-time changes. The system enforced a strict ranking, with no ties allowed, compelling the evaluators to make definitive comparative judgments. This design allowed for a direct comparison of AI-generated outputs against the human-authored baseline, providing a clear measure of their competitive quality.

### B.3    Task 3: Publication Intent Selection

The final task framed the evaluation in the most practical terms by simulating the author's decision-making process for publication. Evaluators were asked: "If you were the author of this paper, which of these figures, if any, would you be willing to use in a camera-ready version of your publication?" This task required them to synthesize their quality assessments with practical considerations, such as stylistic fit with their paper, alignment with conventional academic standards, and potential impact on peer reviewers.

The interface presented each of the six AI-generated figures on a selectable card. Experts could choose one or more figures they deemed publication-ready. An explicit option, "I would not select any of the generated figures," was also provided to capture instances where none of the AI outputs met the required standard for publication. The interface provided clear visual feedback for selected items, such as a green checkmark or a highlighted border, and displayed a summary of the current selections at the bottom of the page. This binary-style decision provided a direct measure of each model's real-world applicability and acceptance rate from the perspective of the original author.

## C    Discussion and Future Outlook

Our work establishes a strong and generalizable foundation for Automated Scientific Schematic Generation (ASSG). By focusing on the Computer Science domain—a field with a diverse and rapidly evolving visual language—we have demonstrated that the "Reasoned Rendering" paradigm can successfully produce high-quality, publication-ready figures. This achievement serves as a robust proof-of-concept and a blueprint for future advancements in AI-driven scientific communication.

Building on this foundation, an exciting future direction is the extension of our framework to other scientific disciplines. The methodology established in FigureBench and AUTOFIGURE can be adapted to create specialized tools that understand the unique visual conventions of fields like biology, chemistry, and economics. For instance, future work could incorporate domain-specific knowledge to accurately generate intricate biological signaling pathway diagrams or standardized molecular structures. This path forward leads from a powerful generalist tool to a suite of expert AI illustrators, each tailored to empower researchers in their respective communities.

Furthermore, our framework masters the creation of high-fidelity static diagrams, which remain the cornerstone of formal scientific publishing. Having established this essential capability, the logical next frontier is to bring these static representations to life. The core principles of AUTOFIGURE—decoupling structural planning from aesthetic rendering—are perfectly suited for the generation of dynamic and interactive schematics. We envision future systems that can produce animated figures to illustrate processes over time or interactive diagrams that allow for user-driven exploration of complex models. This represents a transformative opportunity, moving beyond simply documenting scientific findings to creating rich, immersive experiences that can accelerate understanding and discovery.

## D    Use of Large Language Models

LLMs were utilized as assistive tools at various stages of this research and manuscript preparation to enhance productivity and quality. Our use of these tools was supervised, with the final responsibility for all content resting with the human authors. In the initial phase of our work, we employed LLM-based tools to assist with literature discovery. These tools helped in identifying and summarizing a broad range of relevant prior work, which facilitated a comprehensive understanding of the existing research landscape. During the implementation of the AUTOFIGURE framework, we utilized Claude to assist in writing and refining code segments. This process accelerated development and helped in debugging and optimizing our software components. For the preparation of the manuscript, LLMs (e.g. Gemini-2.5-Pro) were used for text polishing, including improving grammatical correctness, clarity, and readability. The core scientific ideas, methodologies, and arguments presented in this paper were conceived and articulated by the authors. Finally, upon completion of the draft, we subjected the manuscript to a secondary review process using the DeepReviewer (Zhu et al., 2025a) model. This tool helped to proactively identify potential weaknesses in our argumentation, exper-

imental setup, and presentation. We carefully considered the feedback provided by DeepReviewer and made targeted revisions to address the concerns we deemed valid, thereby strengthening the final version of this paper.

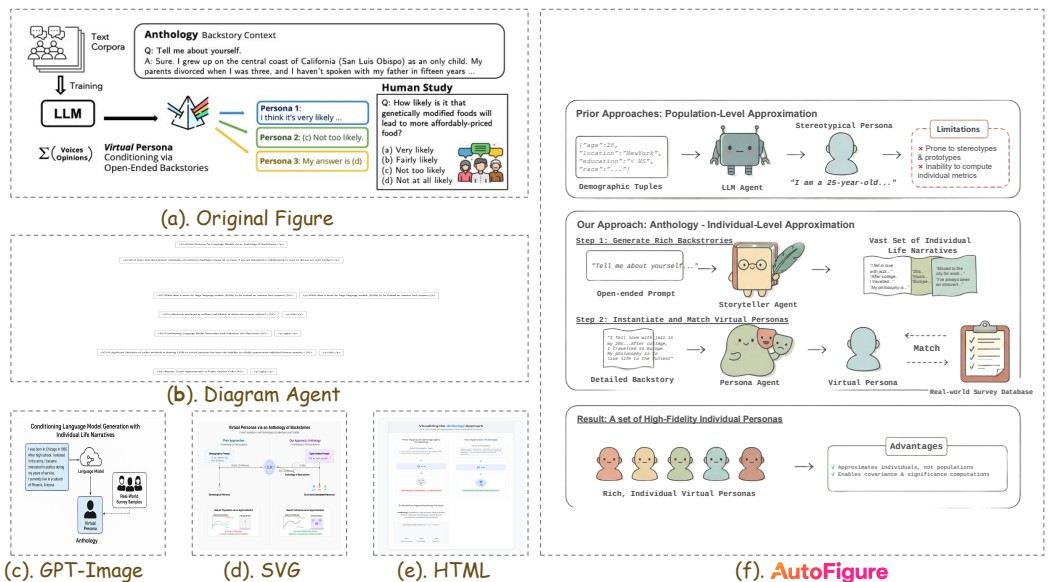

**Figure 7:** Qualitative comparison, contrasting baseline failures with AUTOFIGURE's superior output.

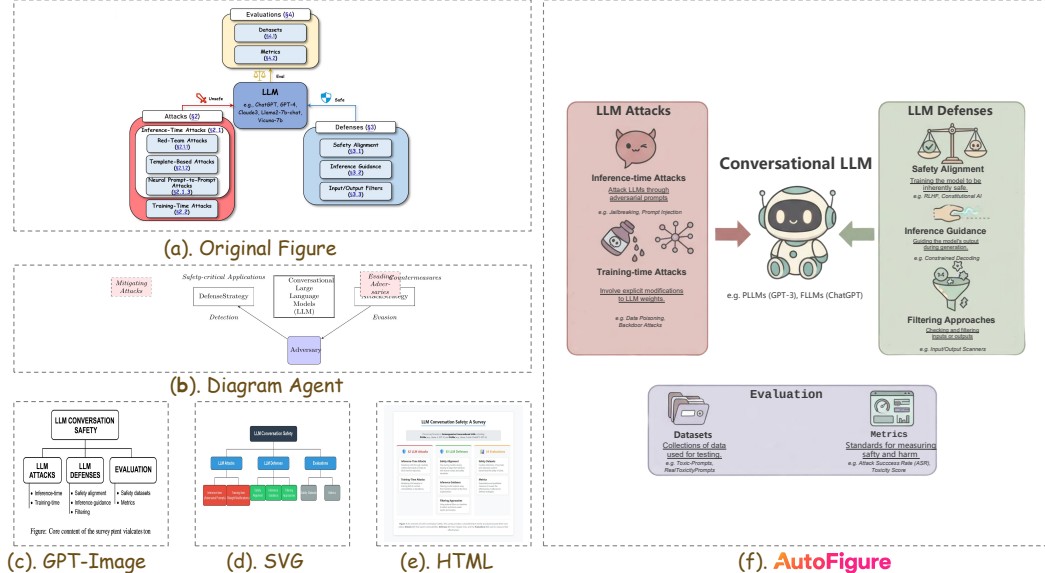

**Figure 8:** Analysis for the Survey category, showing AUTOFIGURE's ability to render complex relationships clearly and accurately.

# E  QUALITATIVE CASE STUDIES

To provide a more granular, qualitative understanding of AUTOFIGURE's capabilities, this section presents a targeted analysis of its performance against baselines across three distinct document types: a technical blog, an academic survey, and a textbook. These examples, shown in Figures 7, 8, and

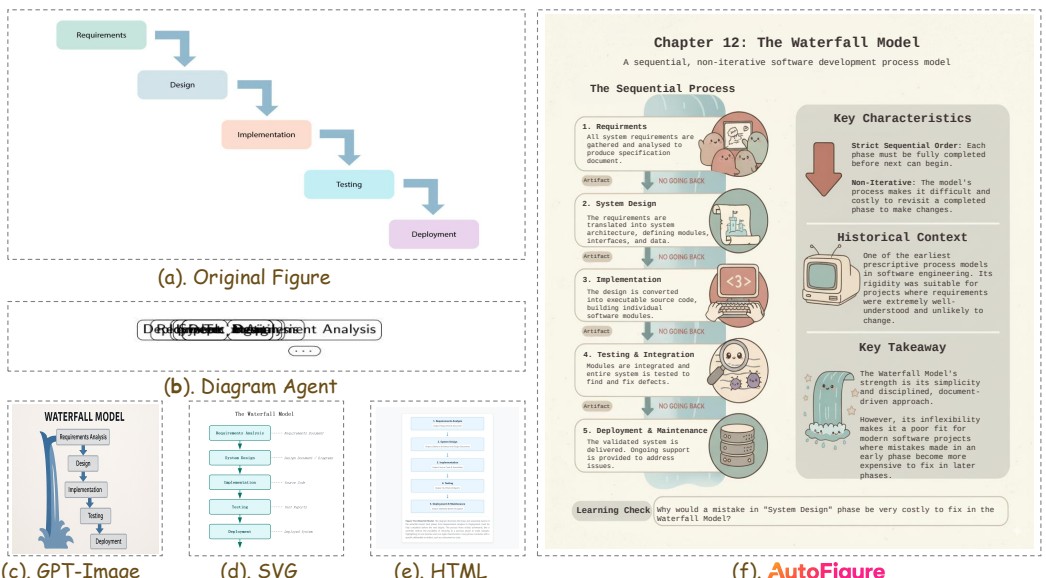

**Figure 9:** Case study for the Textbook category, demonstrating AUTOFIGURE's effectiveness in creating clear pedagogical illustrations.

9, visually substantiate the quantitative findings by highlighting the specific failure modes of baseline methods and demonstrating how AUTOFIGURE's "Reasoned Rendering" paradigm successfully overcomes them.

The blog case in Figure 7 illustrates the generation of a complex process diagram. The end-to-end model, GPT-Image, not only fails to render legible text but also hallucinates an entirely incorrect topic, producing a diagram for "Conditioned Language/Word Detection" instead of the intended "Virtual Persona Opinions." The code-based methods (SVG, HTML) manage to represent the basic flow but result in visually simplistic and unprofessional layouts that fail to capture the nuances of the original figure. In stark contrast, AUTOFIGURE correctly abstracts the source text into a coherent, multi-step narrative ("Prior Approaches," "Our Approach," "Result") and renders it using a clean, aesthetically pleasing design with thematic icons, demonstrating a sophisticated understanding of both content and presentation.

Figure 8 presents a more complex challenge: visualizing a taxonomy of LLM safety concepts. Here, the baselines fail catastrophically on structural fidelity. Diagram Agent produces a gross oversimplification, while GPT-Image and the code-based methods fail to capture the critical categorical relationships between Attacks, Defenses, and Evaluation metrics. Their outputs are either logically incorrect or presented as a flat, confusing flowchart. AUTOFIGURE, however, excels by reimagining the content as a clear infographic. It correctly organizes concepts into distinct, labeled groups (LLM Attacks, LLM Defenses, Evaluation) around a central "Conversational LLM" motif, using thematic icons (a bomb for attacks, scales for defenses) to enhance comprehension. This shows an ability not just to reproduce, but to logically restructure information for clarity.

Finally, the textbook example in Figure 9 highlights the importance of pedagogical clarity. The original figure is a simple, canonical diagram of the Waterfall Model. GPT-Image captures the basic downward flow but suffers from severe text-rendering artifacts, making the labels illegible and thus educationally useless. The SVG and HTML methods produce a structurally correct but visually sterile diagram. AUTOFIGURE's output is transformative; it not only generates a clear and accurate waterfall process with engaging icons but also enriches it by synthesizing key information from the source text into supplementary panels like "Key Characteristics," "Historical Context," and "Key Takeaway." This moves beyond mere illustration to create a comprehensive, high-quality educational artifact, demonstrating a deep contextual understanding of the task's purpose.

## F    LIMITATIONS AND FAILURE ANALYSIS

Despite the strong performance of AUTOFIGURE, we identify several persistent limitations. Most notably, we explicitly list *fine-grained text-rendering accuracy* as a primary bottleneck of the current system and, more broadly, of existing figure-generation pipelines on FigureBench. Even with our erase-and-correct post-processing, the system can still exhibit rare but consequential character-level errors under small font sizes, dense layouts, or visually complex backgrounds. A representative example is the spelling mistake "ravity" (missing "g" in "gravity") observed in one generated figure, which illustrates the gap between long-context semantic reasoning (where the pipeline largely succeeds) and pixel-/glyph-level fidelity (where even minor artifacts can occur). Importantly, we argue that the existence of such subtle errors *inversely highlights* the discriminative power and research value of FigureBench: it exposes a hard, unresolved frontier for the community, rather than indicating a fundamental flaw in our approach. Beyond text accuracy, a second limitation is the tension between aesthetic presentation and scientific rigor: our "concretization" behavior can occasionally drift beyond the strictly literal content if the source text is underspecified, and theoretical or vaguely phrased passages may lead the model to produce a visually plausible but imperfect conceptual structure (e.g., compressing nuanced distinctions or inadvertently imposing hierarchical relations on parallel concepts).

To make these limitations transparent and actionable, the revised manuscript adds a dedicated *Limitations and Failure Analysis* section with concrete failure cases and causal analysis, complementing our qualitative case studies (Appendix § E; Figures 7–9). While those case studies demonstrate that AUTOFIGURE substantially mitigates common baseline failure modes (e.g., illegible text, topic hallucination, and structural collapse), they also help delineate boundary conditions for our system: (i) when accurate rendering hinges on domain-specific conventions or external facts not explicitly stated in the input, the produced structure or labeling may be incomplete; (ii) when the input demands fine-grained ontological distinctions (e.g., multi-branch taxonomies), the model may favor a cleaner visual organization at the expense of faithfully preserving subtle relations; and (iii) when layouts are dense, small typographic errors can survive post-processing and harm educational utility. Looking forward, we note several promising directions to address these deeper reasoning and verification gaps, including incorporating external knowledge bases (retrieval-augmented grounding) and introducing domain verifiers (Domain Verifiers) that can enforce constraint checks over entities, relations, and terminology before final rendering. We expect such verification-oriented components—together with more robust constrained text rendering (e.g., vector-text overlays or tighter OCR-to-layout alignment)—to be key future steps toward closing the "reasoning vs. rendering" gap on FigureBench.

## G    PERFORMANCE EVALUATION ON OPEN-SOURCE MODELS

To ensure the reproducibility of our research and facilitate broader adoption with minimal deployment costs, we extended our evaluation to include state-of-the-art (SOTA) open-source and open-weight models. Specifically, we evaluated AUTOFIGURE using **Qwen3-VL-235B-A22B-Instruct**, **GLM-4.5V**, and **ERNIE-4.5-VL** as the reasoning backbones. This analysis aims to verify whether AUTOFIGURE can maintain high-quality generation without relying on proprietary commercial APIs.

**Comparative Analysis with Commercial Models.** As presented in Table 4, the experimental results demonstrate that top-tier open-source models are highly capable of driving the AUTOFIGURE framework. Notably, **Qwen3-VL-235B** achieved an Overall Score of **7.08**, which not only significantly outperforms other open-source baselines like GLM-4.5V (5.99) but also surpasses several leading commercial models, including Gemini-2.5-Pro (6.99), Claude-4.1-Opus (6.80), and Grok-4 (6.76). It ranks second only to GPT-5 (7.48) among all tested models. This finding strongly suggests that the open-source community has reached a maturity level sufficient for complex scientific illustration tasks.

**Detailed Capabilities and Win-Rates.** We further provide a granular breakdown of open-source model performance across specific sub-dimensions and their Overall Win-Rates in blind pairwise comparisons (Table 5). The performance gap between models highlights a strong correlation between the system's output quality and the backbone model's capabilities in visual reasoning and

Table 4: Performance comparison between Commercial and Open-Source models

| Model Type | Model | Visual Design | Comm. Effect. | Content Fidelity | Overall |
|---|---|---|---|---|---|
| Commercial | GPT-5 | 7.17 | 7.62 | 7.70 | 7.48 |
| | Gemini-2.5-Pro | 6.59 | 7.59 | 6.54 | 6.99 |
| | Claude-4.1-Opus | 6.75 | 7.17 | 6.61 | 6.80 |
| | Grok-4 | 6.46 | 7.60 | 6.49 | 6.76 |
| Open-Source | Qwen3-VL-235B | 7.57 | 7.01 | 7.18 | 7.08 |
| | GLM-4.5V | 6.09 | 6.53 | 6.13 | 5.99 |
| | ERNIE-4.5-VL | 3.04 | 2.89 | 2.68 | 2.64 |

instruction following. The success of Qwen3-VL confirms that by selecting a capable open-source backbone, AUTOFIGURE can be deployed as a cost-effective solution.

Table 5: Detailed dimensional breakdown and Overall Win-Rates for Open-Source models.

| Model | Visual Design | | | Comm. | | Content | | Win-Rate (Overall) |
|---|---|---|---|---|---|---|---|---|
| | Aesth. | Expr. | Polish | Clarity | Flow | Soph. | Fidel. | |
| Qwen3-VL-235B | 0.90 | 0.90 | 0.90 | 0.25 | 0.25 | 0.25 | 0.15 | 40.0% |
| GLM-4.5V | 0.70 | 0.70 | 0.70 | 0.45 | 0.65 | 0.25 | 0.20 | 55.0% |
| ERNIE-4.5-VL | 0.20 | 0.15 | 0.20 | 0.00 | 0.10 | 0.05 | 0.05 | 10.0% |

## H  EXTENDED BLIND PAIRWISE COMPARISON WITH ABSOLUTE QUALITY OPTIONS

To address the limitations of simple relative ranking (which restricts choices to "A", "B", or "Tie") and to investigate the absolute quality of the generated figures, we conducted an extended blind pairwise comparison. In this experiment, we updated the evaluation prompt to include **"Both Good"** and **"Both Bad"** options. This allows us to identify potential "race-to-the-bottom" scenarios (where a model wins only because the competitor is worse) or cases where models are indistinguishable in quality.

We performed this evaluation on the *Paper* subset, a challenging category requiring high structural fidelity. The results are summarized in Table 6.

Table 6: Extended pairwise comparison results on the *Paper* subset with "Both Good" and "Both Bad" options. AutoFigure demonstrates a high win rate while the low "Both Bad" count confirms a high quality baseline across most methods.

| Method | Visual Design | | | Communication | | Content | | Pairwise Decisions | | | | Overall |
|---|---|---|---|---|---|---|---|---|---|---|---|---|
| | Aesth. | Expr. | Polish | Clarity | Flow | Sophist. | Fidelity | Win | Lose | Good | Bad | |
| **AutoFigure** | **0.88** | **0.93** | **0.88** | 0.38 | 0.45 | **0.35** | 0.28 | **29** | 11 | 0 | 0 | **0.73** |
| Gemini-HTML | 0.63 | 0.53 | 0.63 | **0.58** | **0.53** | 0.30 | 0.25 | 21 | 18 | 1 | 0 | 0.53 |
| Gemini-SVG | 0.48 | 0.40 | 0.48 | 0.48 | 0.45 | 0.33 | **0.30** | 18 | 20 | **2** | 0 | 0.45 |
| GPT-Image | 0.25 | 0.25 | 0.25 | 0.08 | 0.08 | 0.00 | 0.00 | 4 | 36 | 0 | 0 | 0.10 |
| Diagram Agent | 0.00 | 0.00 | 0.00 | 0.00 | 0.00 | 0.00 | 0.00 | 0 | **39** | 0 | **1** | 0.00 |

The extended evaluation yields three critical insights: the negligible occurrence of "Both Bad" (1 instance) and "Both Good" (3 instances) selections confirms that the models meet a high baseline of readability while maintaining clear discriminability in quality, rather than suffering from a "race to the bottom." Within this validated framework, AutoFigure maintains dominance with a 72.5% win rate; its overwhelming superiority in visual design metrics (e.g., 0.93 vs. 0.53 in Expressiveness) significantly outweighs the structural clarity of code-based baselines like Gemini-HTML, securing the highest overall score (0.73) and demonstrating true publication readiness.

## I   ABLATION STUDY: CONTRIBUTION OF THE TEXT REFINEMENT MODULE

To systematically evaluate the contribution of individual components within our multi-stage pipeline, we conducted a focused ablation study on the **Stage 2 Text Refinement (Erase-and-Correct)** module. This complements the ablations on the Reasoning Backbone and Intermediate Format presented in the main text. We compared the full AutoFigure pipeline against a variant where the text erasure and correction step was removed (*w/o Text Refinement*) to quantify its impact on the final output quality.

**Table 7:** Ablation study results on the Text Refinement module. While the Overall score shows a moderate increase, the module significantly enhances visual design metrics, confirming its role in achieving publication-ready quality.

| Model | Visual Design | | | Communication | | Content Quality | | | Overall |
|---|---|---|---|---|---|---|---|---|---|
| | Aesth. | Expr. | Polish | Clarity | Flow | Acc. | Comp. | Appr. | |
| **AutoFigure (Full)** | **7.49** | **7.20** | **6.80** | **7.53** | **7.73** | 7.45 | **6.83** | 6.42 | **7.18** |
| w/o Text Refinement | 7.39 | 7.12 | 6.70 | 7.50 | 7.70 | **7.53** | 6.74 | **6.47** | 7.14 |

The comparative evaluation yields a clear conclusion regarding the necessity of the refinement stage. Although the *Overall* score improvement appears moderate (7.18 vs. 7.14), the granular breakdown reveals that the Text Refinement module drives significant gains in dimensions most critical to visual presentation: **Aesthetic Quality** (+0.10), **Professional Polish** (+0.10), and **Visual Expressiveness** (+0.08). These improvements confirm that the "Erase-and-Correct" strategy is pivotal for eliminating generative artifacts (such as blurred text) and elevating the figure from a "usable" draft to a professional, publication-ready illustration.

## J   EFFICIENCY AND COST ANALYSIS

To assess the practical viability and scalability of AutoFigure in real-world applications, we conducted a comprehensive analysis of inference latency and economic costs. We utilized typical long-form academic papers (average length >10k tokens) as input and compared two distinct deployment settings: (1) A commercial closed-source solution using the **Gemini-2.5-Pro API**, and (2) A local deployment using the open-source **Qwen-3-VL** model on a high-performance computing node equipped with NVIDIA H100 GPUs.

As detailed in Table 8, the choice of deployment strategy significantly impacts both generation speed and cost. When relying on the commercial API, generating a single publication-ready illustration takes approximately **17.5 minutes** with an average cost of **$0.20**. In contrast, deploying the system locally on H100 GPUs reduces the total generation time to **~9.3 minutes**—a nearly **2× speedup**—primarily due to the elimination of network latency and higher inference throughput during the intensive iterative reasoning phase (Stage 2). Furthermore, the local deployment model reduces the marginal cost per figure to effectively zero (excluding hardware amortization and electricity).

Our experiments indicate that deploying AutoFigure does not require prohibitively expensive supercomputing clusters. The performance metrics reported for the local setup (Qwen-3-VL) can be achieved using a computing node equipped with **2× NVIDIA H100 GPUs** or two standard **NVIDIA DGX Spark** servers (approximate value $3,000 each). This accessibility ensures that research labs can deploy AutoFigure locally to ensure data privacy and high throughput without recurring API costs.

## K   HUMAN SANITY CHECK ON AUTOMATED DATASET STATISTICS

We utilized InternVL-3.5 to automatically compute statistical metrics (Text Density, Components, Colors, and Shapes) for the FigureBench dataset. To address potential concerns regarding the reli-

**Table 8:** Breakdown of efficiency and cost for generating a single scientific illustration. Comparing Commercial API (Gemini-2.5) vs. Local Deployment (Qwen-3-VL on H100).

| Stage | Core Task | Gemini-2.5 (API) Time / Cost | Qwen-3-VL (Local) Time / Cost | Remarks |
|---|---|---|---|---|
| **Stage 1** | Concept Extraction & Method Distillation | **~22s /** < $0.01 | **~12s /** ~ $0.00 | Local inference eliminates network latency, doubling speed. |
| **Stage 2** | Layout Planning (Avg. 5 iterations) | ~660s / ~ $0.14 | **~390s /** ~ $0.00 | H100 throughput significantly accelerates the iterative critique-and-refine loop. |
| **Stage 3** | Aesthetic Rendering & Post-processing | ~370s / ~ $0.05 | **~250s /** ~ $0.00 | Code generation and local rendering/OCR are faster than API calls. |
| **Total** | **End-to-End Generation** | **~17.5 min /** ~ $0.20 | **~9.3 min /** ~ $0.00* | **Local deployment achieves ~2× speedup with negligible marginal cost.** |

\* Marginal cost excludes hardware amortization and electricity.

ability of these automated measurements and to quantify potential errors, we conducted a human-audited sanity check.

**Methodology.** We randomly sampled a subset of 21 text-figure pairs from the FigureBench test set. Expert annotators were tasked with manually verifying the four key metrics for each sample. The detailed breakdown of this human audit is presented in Table 9.

**Table 9:** Human-audited statistics on a random subset of 21 samples. The results serve as a sanity check for the automated statistics provided by InternVL-3.5.

| Paper ID | Text Density (%) | Connected Components | Color | Shape |
|---|---|---|---|---|
| 2212.09561 | 75 | 5 | 8 | 6 |
| 2304.01665 | 30 | 4 | 5 | 5 |
| 2304.03531 | 40 | 5 | 5 | 6 |
| 2305.04505 | 65 | 8 | 5 | 4 |
| 2305.15075 | 70 | 4 | 9 | 4 |
| 2510.0513 | 65 | 6 | 5 | 3 |
| 2310.05157 | 90 | 3 | 8 | 2 |
| 2402.13753 | 25 | 3 | 6 | 6 |
| 2402.16048 | 65 | 5 | 6 | 3 |
| 2402.1818 | 45 | 4 | 7 | 5 |
| 2404.1196 | 90 | 8 | 8 | 6 |
| 2405.06312 | 55 | 8 | 9 | 7 |
| 2408.11779 | 30 | 7 | 9 | 8 |
| 2409.07429 | 70 | 4 | 10 | 7 |
| 2411.00816 | 30 | 4 | 8 | 5 |
| 2412.11506 | 45 | 4 | 7 | 5 |
| 2502.10709 | 50 | 6 | 5 | 5 |
| 2502.13723 | 60 | 8 | 6 | 7 |
| 2503.06635 | 20 | 9 | 10 | 7 |
| 2503.08569 | 75 | 6 | 10 | 5 |
| 2504.20972 | 45 | 7 | 7 | 5 |
| **Average (Human)** | **54.29** | **5.62** | **7.29** | **5.29** |

**Comparison and Discussion.** Comparing the human-verified averages on this subset with the full-dataset automated statistics (Text Density 41.2%, Components 5.3, Colors 6.2, Shapes 6.4), we observe that the values are in the same order of magnitude and the relative deviations are within a

reasonable range. Specifically, both human and automated statistics indicate a high level of visual complexity (Components: 5.62 vs. 5.3; Colors: 7.29 vs. 6.2), confirming that the dataset presents a non-trivial challenge for generation models. Regarding text density, the human estimate (54.29%) is slightly higher than the automated measurement (41.2%), likely because human annotators tend to perceive the bounding box area of text blocks whereas the model calculates pixel-level density; nevertheless, both metrics consistently categorize the samples as "text-heavy" compared to standard image datasets. Overall, these results verify that the automated statistics generated by InternVL-3.5 are reliable at a macro level, effectively characterizing the difficulty distribution of FigureBench and supporting the validity of our dataset analysis.

## L    STYLE CONTROLLABILITY AND DIVERSITY

To address concerns regarding the apparent style uniformity in the main paper and to demonstrate the versatility of our framework, we conducted a controlled experiment on style controllability. We emphasize that the consistent visual style (Q-version avatars with Morandi color palette) used throughout the main text was a deliberate choice to ensure visual consistency and readability for comparative analysis, rather than a limitation of the model.

**Experimental Setup.** We kept the structural layout and textual content (Stage 1 output) fixed and only varied the style description prompt in Stage 2. We tested three distinct style prompts: 1) **Prompt 1 (Default):** "Delicate and cute cartoon comic style (using Morandi color palette)"; 2) **Prompt 2 (Creative):** "comic style"; and 3) **Prompt 3 (Minimalist):** "modern minimalist design".

**Quantitative Analysis.** Table 10 presents the automated multi-dimensional evaluation results. The Overall scores across the three styles are highly consistent (ranging from 7.18 to 7.27), indicating that altering the style descriptor does not negatively impact the structural integrity or logical flow of the illustration. Table 11 shows the results of the blind pairwise comparison (VLM-as-a-judge). The Win-Rates are similarly stable, confirming that AutoFigure can adapt to different aesthetic requirements without compromising content quality.

**Table 10:** Automated multi-dimensional scores under different style prompts. The consistent Overall scores demonstrate that AutoFigure maintains high quality across diverse aesthetic styles.

| Style Prompt | Aesthetic & Design | Visual Express. | Prof. Polish | Clarity | Logical Flow | Accuracy | Completeness | Appropriateness | Overall |
|---|---|---|---|---|---|---|---|---|---|
| Prompt 1 (Default) | 7.49 | 7.20 | 6.80 | 7.53 | 7.73 | 7.45 | 6.83 | 6.42 | 7.18 |
| Prompt 2 (Comic) | 7.32 | 7.24 | 6.78 | 7.58 | 7.78 | 7.63 | 7.02 | 6.82 | 7.27 |
| Prompt 3 (Minimalist) | 7.14 | 6.27 | 7.09 | 7.72 | 7.75 | 7.54 | 6.75 | 7.28 | 7.19 |

**Table 11:** Blind pairwise comparison results for different style prompts. Comparison metrics show robust performance across styles.

| Style Prompt | Visual Design | | | Comm. | | Content | | Decision Counts | | | | Overall |
|---|---|---|---|---|---|---|---|---|---|---|---|---|
| | Aesth. | Expr. | Polish | Clarity | Flow | Sophist. | Fidel. | Win | Lose | Good | Bad | |
| Prompt 1 (Default) | 0.85 | 0.85 | 0.85 | 0.40 | 0.50 | 0.35 | 0.35 | 13 | 7 | 0 | 0 | 0.65 |
| Prompt 2 (Comic) | 0.90 | 1.00 | 0.90 | 0.35 | 0.40 | 0.35 | 0.40 | 12 | 7 | 1 | 0 | 0.60 |
| Prompt 3 (Minimalist) | 0.65 | 0.65 | 0.65 | 0.65 | 0.65 | 0.45 | 0.35 | 13 | 5 | 1 | 1 | 0.65 |

## M    MINIMAL WORKING EXAMPLE: WORKFLOW FOR THE "A2P" PAPER

To illustrate the practical operation of AutoFigure, we detail the step-by-step generation process for the main figure of A2P (West et al., 2025) and corresponding artifacts in the supplementary material. The end-to-end workflow proceeds as follows:

**Input Analysis and Method Extraction:** AutoFigure first ingests the source document (supporting formats such as `.pdf`, `.txt`, `.md`, and `.tex`). In this instance, the `.tex` source file is processed

by **Gemini-2.5-Pro**, which analyzes the full text to extract and distill the core methodological contributions.

**Initial Layout Generation:** The extracted methodology is passed to the *Initial Design Agent* and *Initial Critic Agent*. These agents collaborate to generate a preliminary vector layout file (`iteration_0.svg`) along with its corresponding .png version (`iteration_0.png`) and provide an initial quality assessment score.

**Iterative Refinement:** The initial layout and score are fed into the "Critique-and-Refine" loop. The critique agent orchestrated a comprehensive optimization on the initial draft (`iteration_0`): 1) It corrected the erroneous arrow connections to ensure logical data flow towards the output; 2) It re-engineered the spatial arrangement by moving the 'Output' module to the right, establishing an intuitive left-to-right visual flow; 3) It expanded the 'Crucial Pre-processing' section to refine methodological details; 4) It consolidated the disorganized layout into three distinct, aligned columns; and 5) It resolved aesthetic artifacts, specifically fixing the text overflow in the conversation log steps. In this specific case, the design achieved the required quality threshold (score of 8.5) after the first iteration (`iteration_1`), triggering an early exit from the loop without further modification with its artifacts `layout.png` and `layout.svg`.

**Aesthetic Rendering:** The *Rendering Module* employs **Gemini-2.5-Pro** to translate the finalized SVG code into a descriptive text-to-image prompt. This prompt, along with the rasterized layout reference (`layout.png`), is sent to the image generation model (**Nano-Banana**) to synthesize the aesthetically polished illustration (`polished.png`).

**OCR Extraction:** To address potential text rendering artifacts, **EasyOCR** is used to detect text content and bounding box coordinates from `polished.png`, storing the data in a raw mapping file (`library.json`).

**Text Verification:** `library.json` and the ground-truth structure `layout.svg` are submitted to **Gemini-2.5-Pro** for verification. The model corrects any OCR errors or hallucinations in the extracted text using the SVG as the ground truth, producing a validated text mapping (`corrected_library.json`).

**Background Erasure:** The **ClipDrop** API is applied to `polished.png` to remove the original (potentially blurred) text, resulting in a clean background image (`erased.png`).

**Final Composition:** Finally, **Gemini-2.5-Pro** utilizes the coordinates and content from `corrected_library.json` to programmatically overlay precise, vector-quality text onto the `erased.png` background (generating a final presentation slide), thereby producing the final, publication-ready scientific illustration (`figure.pptx`).

## N  EXTENDED BASELINE EXPERIMENTAL RESULTS

To comprehensively evaluate the performance positioning of AutoFigure, we expanded our comparative experiments on the **Paper** category by incorporating two additional classes of baselines: TiKZ-based code generation methods and Agentic presentation systems. Specifically, we introduced **TikZero** and **TikZero+** (Belouadi et al., 2025) as representatives of the Automatikz paradigm, which attempts to directly generate compilable LaTeX TiKZ code from scientific text, and **AutoPresent** (Ge et al., 2025) as a representative of presentation generation agents that focus on arranging content into slide layouts. Note that we excluded systems like Paper2Poster (Pang et al., 2025) and PPTAgent (Zheng et al., 2025) from this specific benchmark, as they strictly require original source images for layout arrangement rather than generating conceptual illustrations from pure text. The quantitative results of this expanded comparison are presented in Table 12, which clearly contrasts the capabilities of these different paradigms.

The results reveal a dominant performance by AutoFigure (Overall: 7.03, Win-Rate: 53.0%), while the new baselines struggle significantly. TikZ-based methods (TikZero/TikZero+) scored extremely low (Overall < 1.5), a failure that extends beyond syntax errors to a fundamental limitation of the end-to-end code generation paradigm; forcing an LLM to linearly serialize high-dimensional scientific structures (dozens of entities and complex topological flows) into LaTeX code imposes an excessive cognitive load, causing the model to deplete its reasoning capacity on low-level coordinate calculations rather than macro-level logical construction. In contrast, AutoFigure's decoupled

**Table 12:** Extended baseline comparison results under the **Paper** category.

| Method | Visual Design | | | Communication | | Content Fidelity | | | Overall | Win-Rate |
|---|---|---|---|---|---|---|---|---|---|---|
| | Aesthetic | Express. | Polish | Clarity | Flow | Accuracy | Complete. | Approp. | | |
| **AutoFigure** | **7.28** | **6.99** | **6.92** | **7.34** | **7.87** | 6.96 | **6.51** | **6.40** | **7.03** | **53.0%** |
| HTML-Code | 5.90 | 5.04 | 5.84 | 7.17 | 7.38 | **6.99** | 6.37 | 6.15 | 6.35 | 11.0% |
| SVG-Code | 5.00 | 4.19 | 4.89 | 6.34 | 6.48 | 6.15 | 5.53 | 5.37 | 5.49 | 31.0% |
| GPT-Image | 4.24 | 3.47 | 4.00 | 5.63 | 5.63 | 4.77 | 4.08 | 4.25 | 4.51 | 7.0% |
| AutoPresent | 2.74 | 1.79 | 2.00 | 2.87 | 2.91 | 3.15 | 2.60 | 2.35 | 2.55 | 10.0% |
| Diagram Agent | 2.25 | 1.73 | 2.04 | 2.67 | 2.49 | 2.11 | 1.72 | 1.94 | 2.12 | 0.0% |
| TikZero+ | 1.52 | 1.25 | 1.38 | 1.90 | 1.93 | 1.20 | 1.10 | 1.35 | 1.45 | 0.0% |
| TikZero | 2.00 | 1.50 | 1.00 | 1.00 | 1.50 | 1.00 | 1.00 | 1.00 | 1.25 | 0.0% |

"Reasoning-then-Rendering" strategy effectively bypasses this bottleneck. Similarly, while Auto-Present (Overall 2.55) outperformed TikZ methods, it still lagged significantly behind AutoFigure because such agents are primarily designed for *arranging* existing textual and visual assets into slides rather than *designing* explanatory schematics from scratch, lacking the specialized reasoning modules required to translate abstract scientific text into visual logic.

## O    QUALITATIVE ANALYSIS ON CHALLENGES IN THE "PAPER" CATEGORY

Our quantitative evaluation revealed that the "Paper" category exhibits lower win rates compared to "Survey" or "Textbook" categories. To investigate the underlying causes, we conducted a deep qualitative analysis using the InstructGPT paper as a representative case study. We attribute this performance gap primarily to the hierarchical complexity of the information and the necessity for novel, bespoke design patterns that characterize research papers.

A primary challenge lies in the hierarchical density of information. Unlike textbook diagrams that often explain a single isolated concept, illustrations in research papers, such as the InstructGPT framework, frequently need to visualize information across three distinct depth levels simultaneously. This includes the macro-level workflow (e.g., the transition from SFT to RM and PPO), the micro-level procedural sub-steps within each phase (e.g., constructing demonstration datasets, ranking candidate outputs), and the fine-grained entity details (e.g., specific roles like human labelers, pre-trained models, or loss terms like KL penalty). For AutoFigure, extracting this multi-layered structure from long-form text presents a massive challenge during the semantic parsing stage. The model must perform complex reasoning to determine which information constitutes a "critical node" for visualization versus what should be condensed into textual descriptions, and subsequently decide how to spatially arrange these nested relationships in a 2D layout. This cognitive load is significantly higher than that required for standard pedagogical schematics.

Furthermore, unlike surveys or textbooks which often rely on established schemas (such as tax-onomy trees or canonical flowcharts), scientific paper illustrations are typically bespoke designs intended to represent a unique, novel pipeline. For instance, the original InstructGPT diagram employs specific color coding and positional grouping to uniquely delineate its three stages, a visual structure tailored specifically to its methodology. Consequently, AutoFigure cannot rely on learn-ing stable visual templates or "pattern matching" from the training data. Instead, it must engage in "design from scratch," conceptualizing a custom topology for a pipeline that has no prior visual precedent. This high degree of freedom leads to a prevalent trade-off in our generated results: the model may either merge sub-steps to maintain a clean layout, resulting in penalties for incomplete information, or attempt to preserve every detected node, leading to a cluttered layout that reduces readability. This acute trade-off between structural completeness and aesthetic clarity explains the lower win rates observed in the Paper category compared to domains with more standardized visual grammars.

# P   HUMAN-LLM CORRELATION STUDY

To ensure the methodological rigor of our evaluation and address potential concerns regarding bias in the VLM-as-a-judge paradigm, we conducted a dedicated Human-LLM correlation study. Specifically, we recruited human experts to rigorously score 95 generated samples covering five different generation methods across all evaluation dimensions, including Visual Design, Communication Effectiveness, and Content Fidelity. We then calculated the statistical correlation between these human scores and the automated scores assigned by the VLM.

The experimental results demonstrate that the VLM evaluator is highly aligned with human scores, both numerically and in terms of ranking consistency. First, regarding the **Overall Score Correlation**, the **Pearson Correlation coefficient** reached $r = 0.659$ ($p < 0.001$), indicating a significant linear positive correlation between the VLM scoring trends and human experts, confirming the model's ability to accurately capture quality differences.

Second, regarding **Ranking Consistency**, the **Spearman Rank Correlation** was $\rho = 0.593$, and **Kendall's Tau** was $0.497$. These metrics indicate that the VLM maintains robust consistency with human judgment in determining the relative ranking of different models. Notably, the **Mean Ranking Error (MRE)** was calculated to be only $0.98$, implying that the ranking deviation between the VLM and humans is, on average, less than one position.

Third, we performed **Sub-metric Validation**. Statistically significant positive correlations were observed across the three core dimensions: Accuracy ($r = 0.646$), Aesthetics ($r = 0.587$), and Clarity ($r = 0.559$). This demonstrates the robustness of the VLM across different evaluation perspectives. Collectively, this analysis confirms that the VLM-as-a-judge method adopted in this work is a verified and reliable evaluation proxy that accurately reflects human preferences.

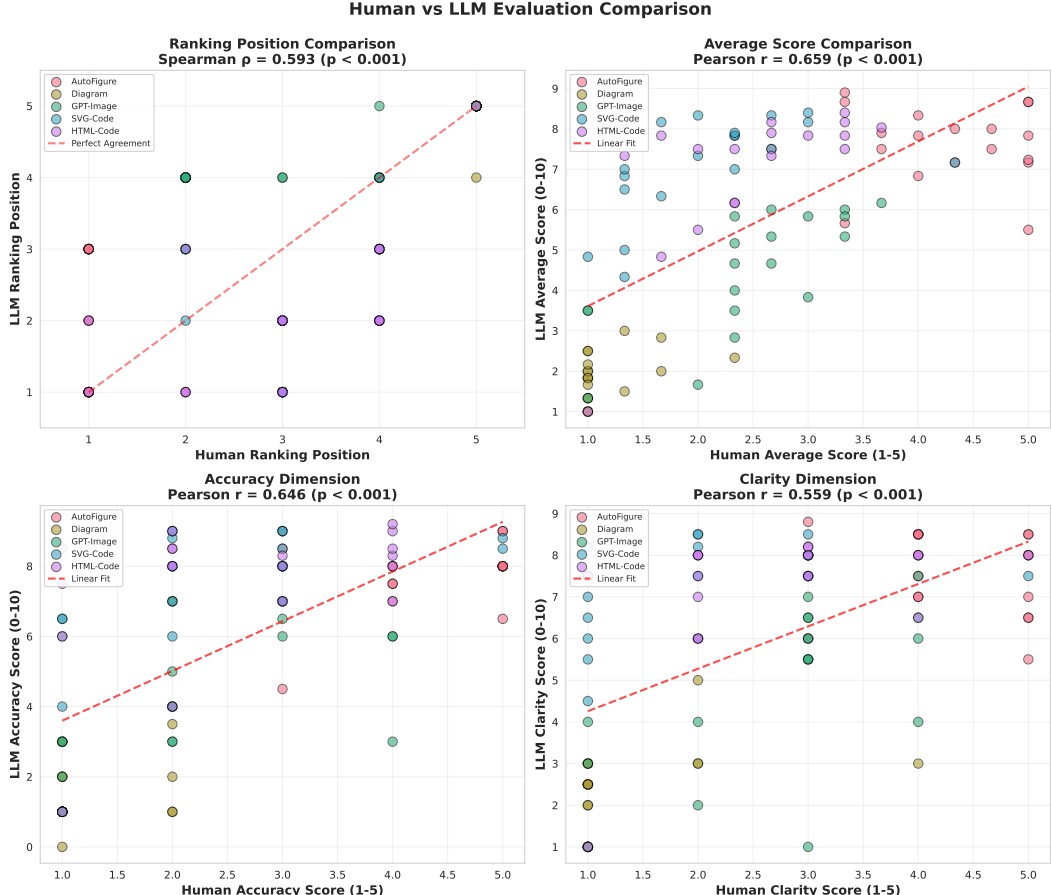

Figure 10: **Human-LLM correlation study validating the VLM-as-a-judge paradigm.** Subplots illustrate the alignment between human expert ratings and VLM automated scores across **(Top-Left)** ranking positions, **(Top-Right)** overall average scores, **(Bottom-Left)** accuracy dimension, and **(Bottom-Right)** clarity dimension.

# Q CASES

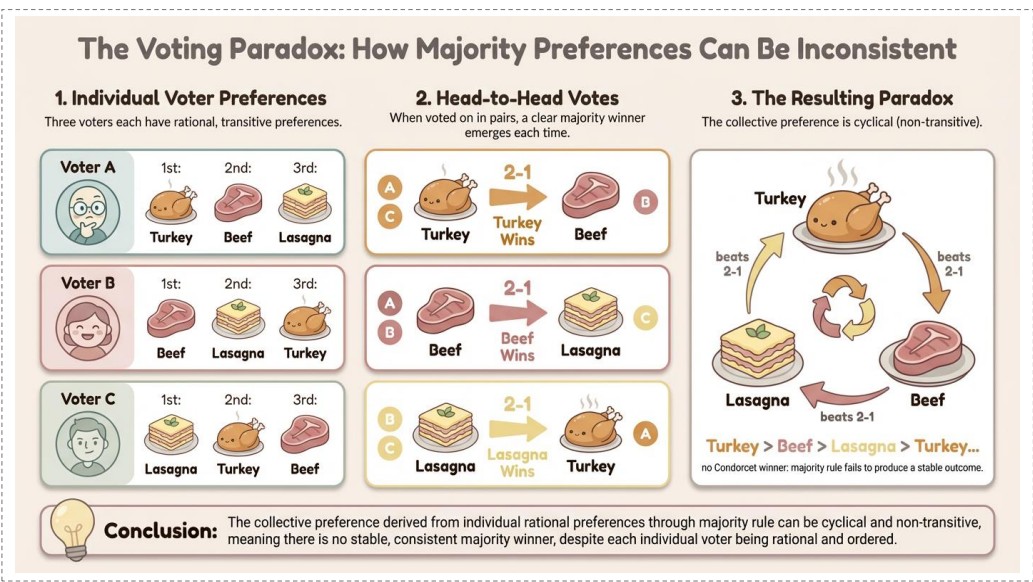

**Figure 11:** This is Case 1 from Textbook. The original text is located at [Link].

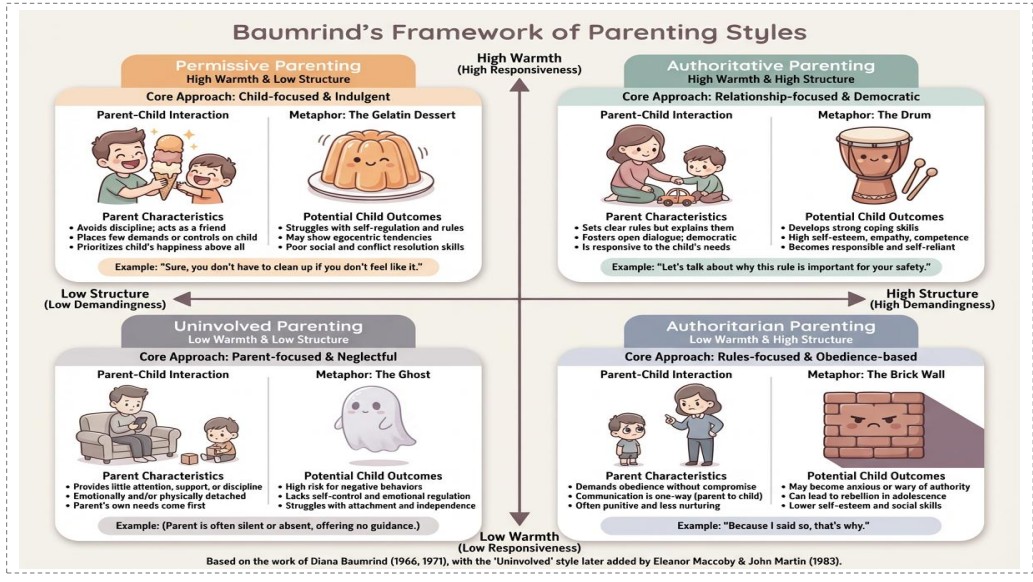

**Figure 12:** This is Case 2 from Textbook. The original text is located at [Link].

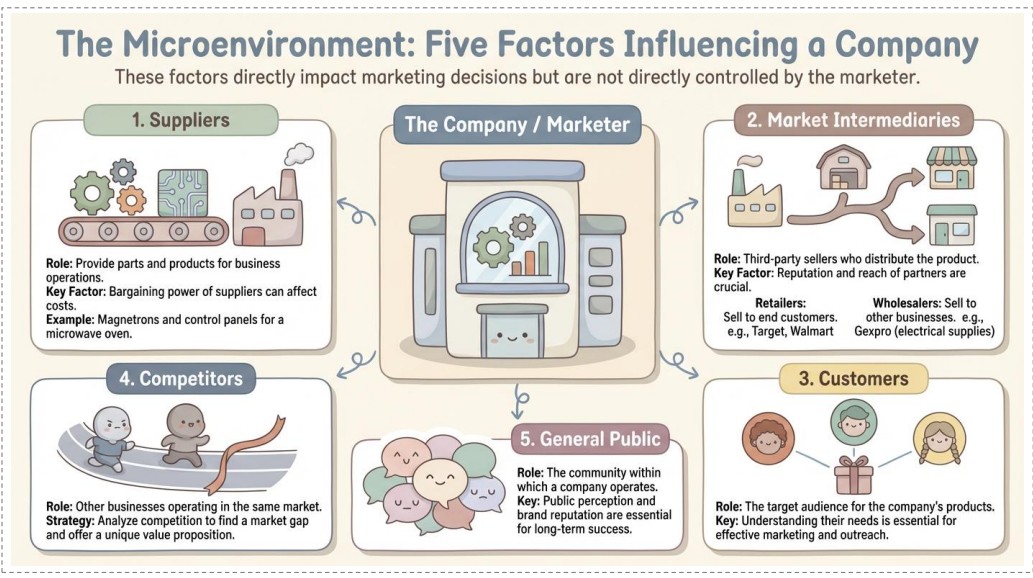

**Figure 13:** This is Case 3 from Textbook. The original text is located at [Link].

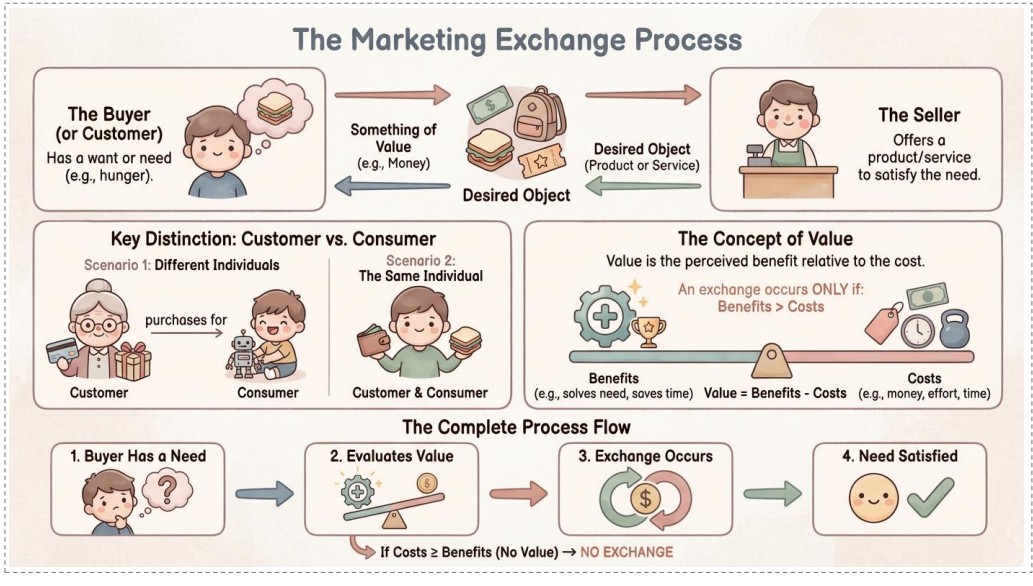

**Figure 14:** This is Case 4 from Textbook. The original text is located at [Link].

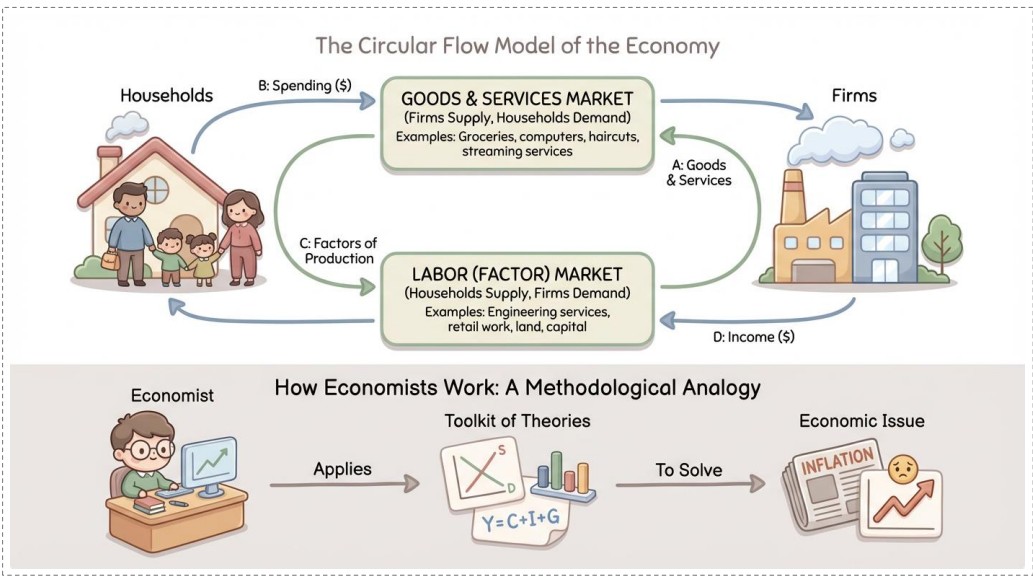

**Figure 15:** This is Case 5 from Textbook. The original text is located at [Link].

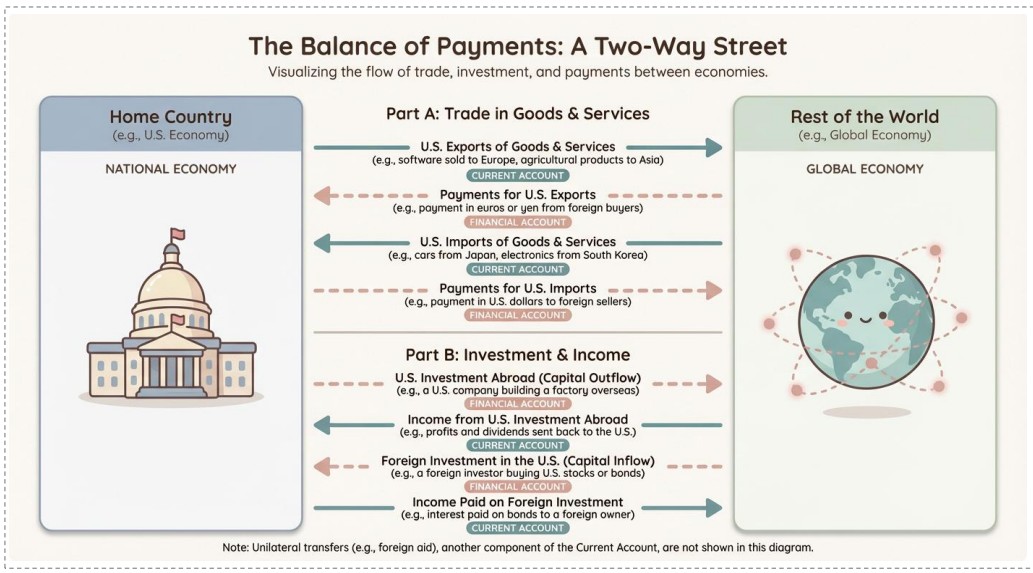

**Figure 16:** This is Case 6 from Textbook. The original text is located at [Link].

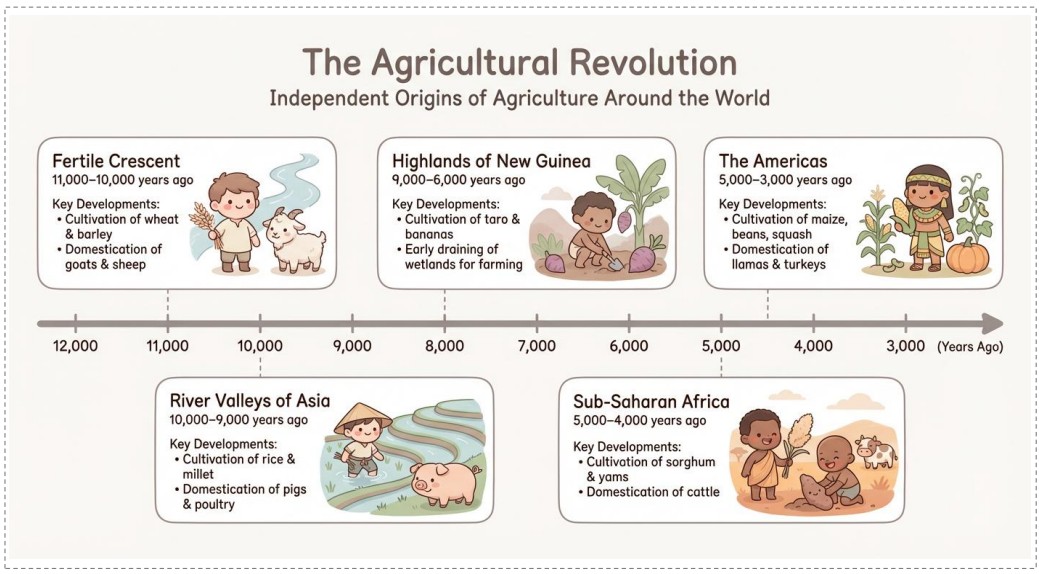

**Figure 17:** This is Case 7 from Textbook. The original text is located at [Link].

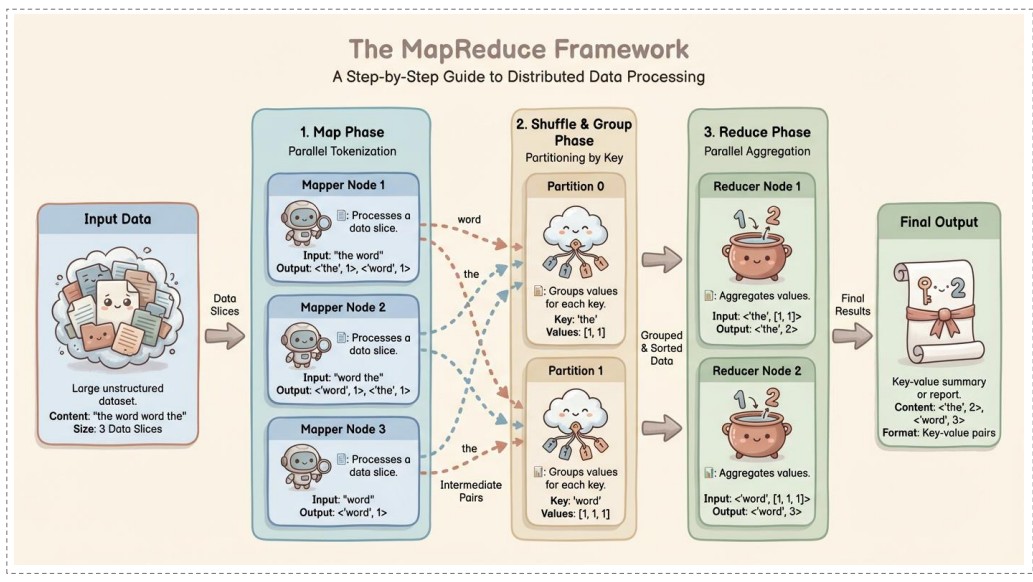

**Figure 18:** This is Case 8 from Textbook. The original text is located at [Link].

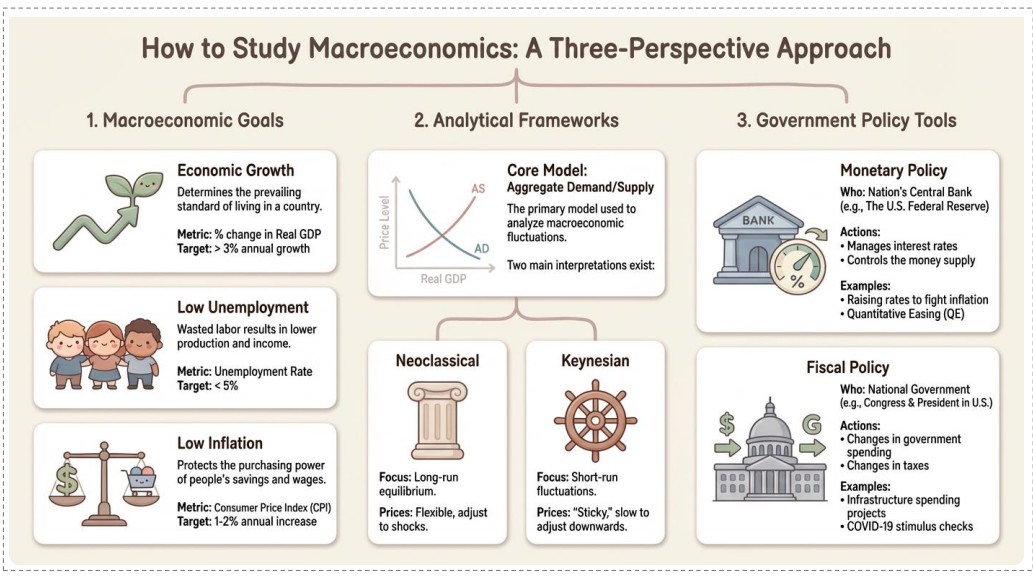

**Figure 19:** This is Case 9 from Textbook. The original text is located at [Link].

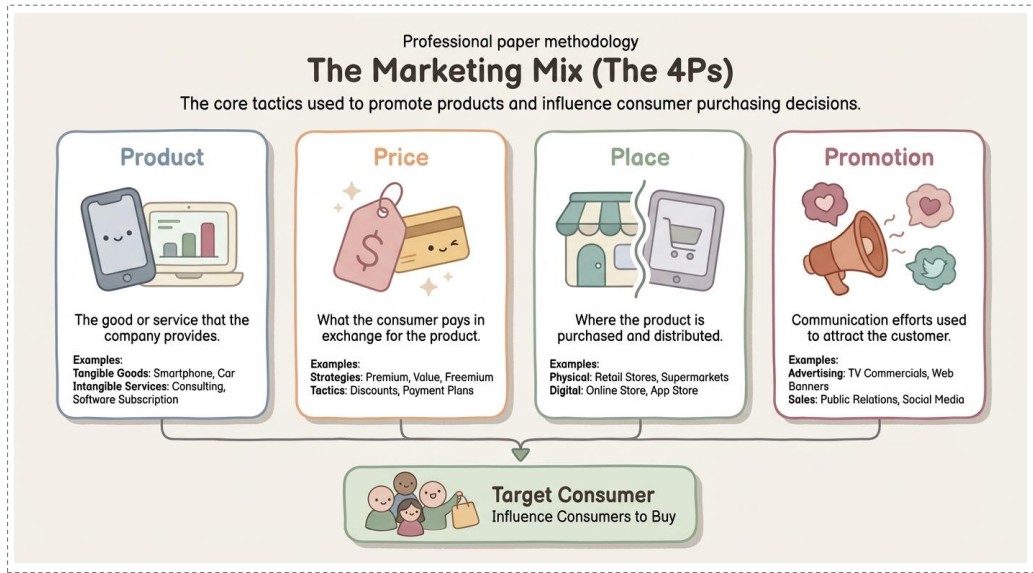

**Figure 20:** This is Case 10 from Textbook. The original text is located at [Link].

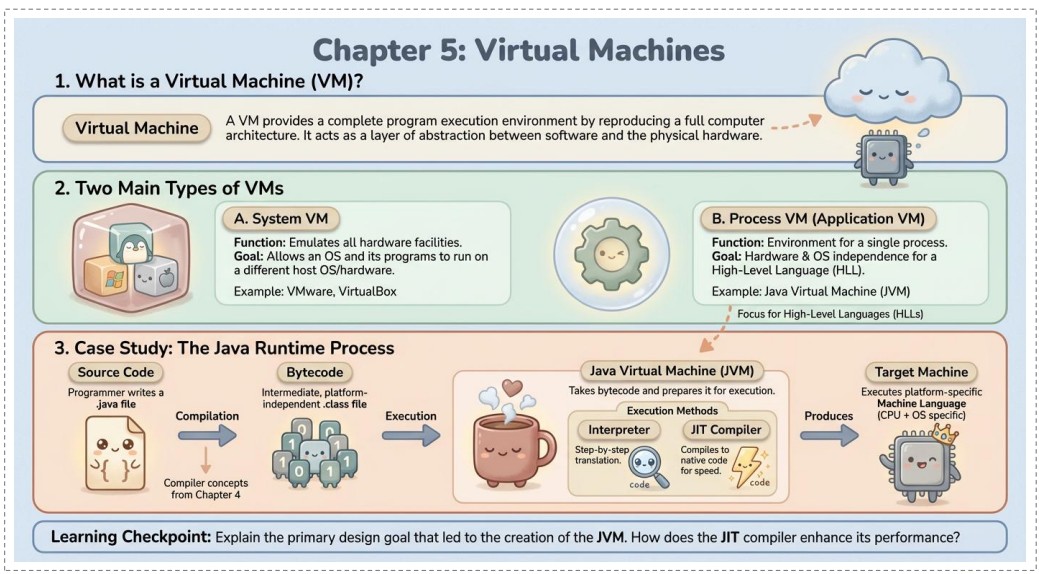

**Figure 21:** This is Case 11 from Textbook. The original text is located at [Link].

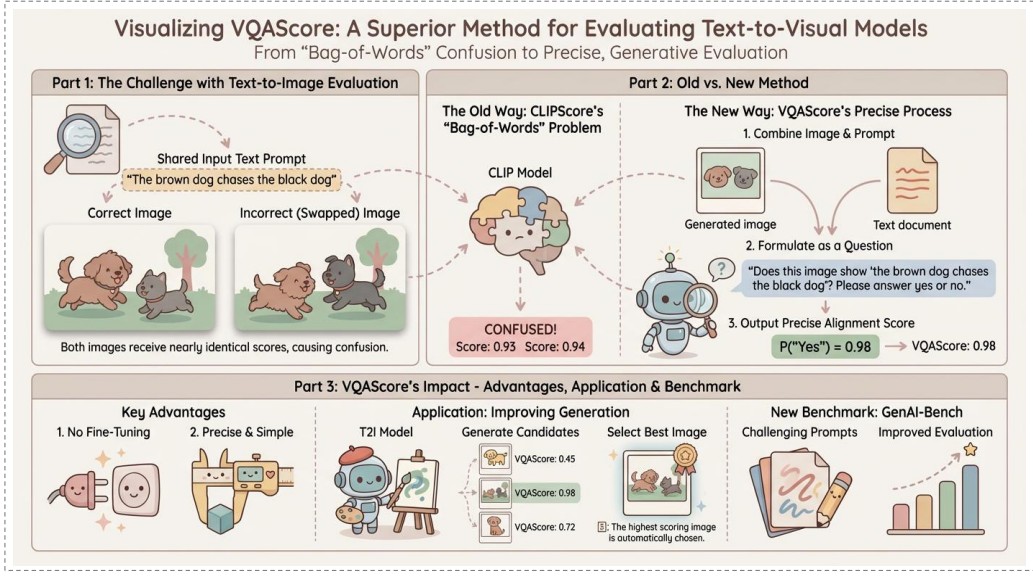

**Figure 22:** This is Case 1 from Blog. The original text is located at [Link].

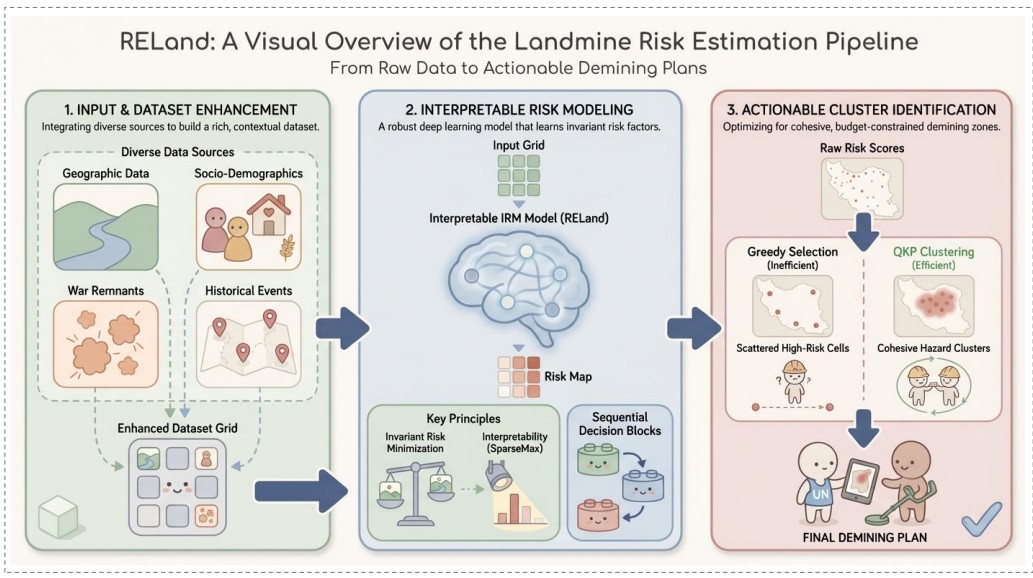

**Figure 23:** This is Case 2 from Blog. The original text is located at [Link].

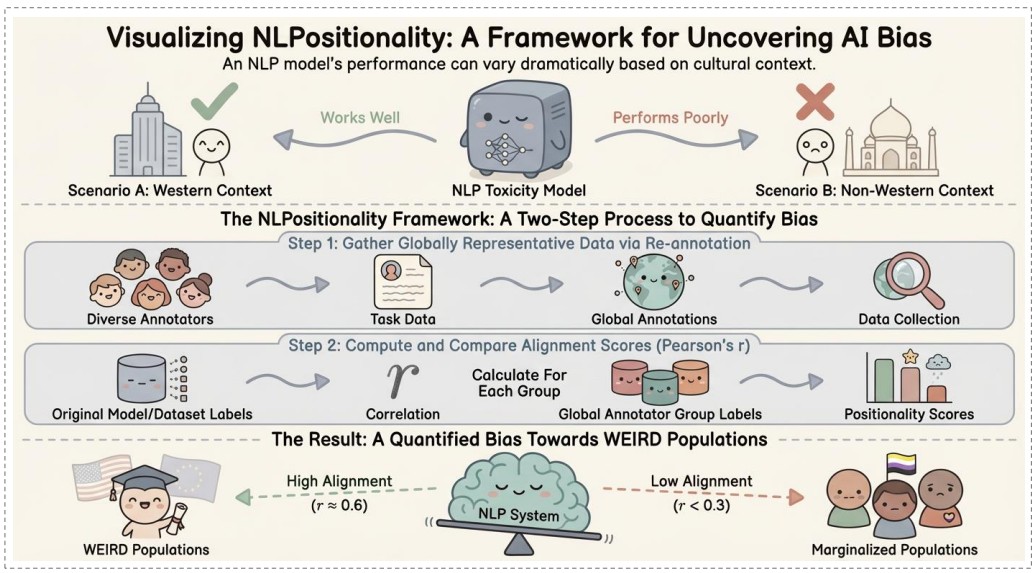

**Figure 24:** This is Case 3 from Blog. The original text is located at [Link].

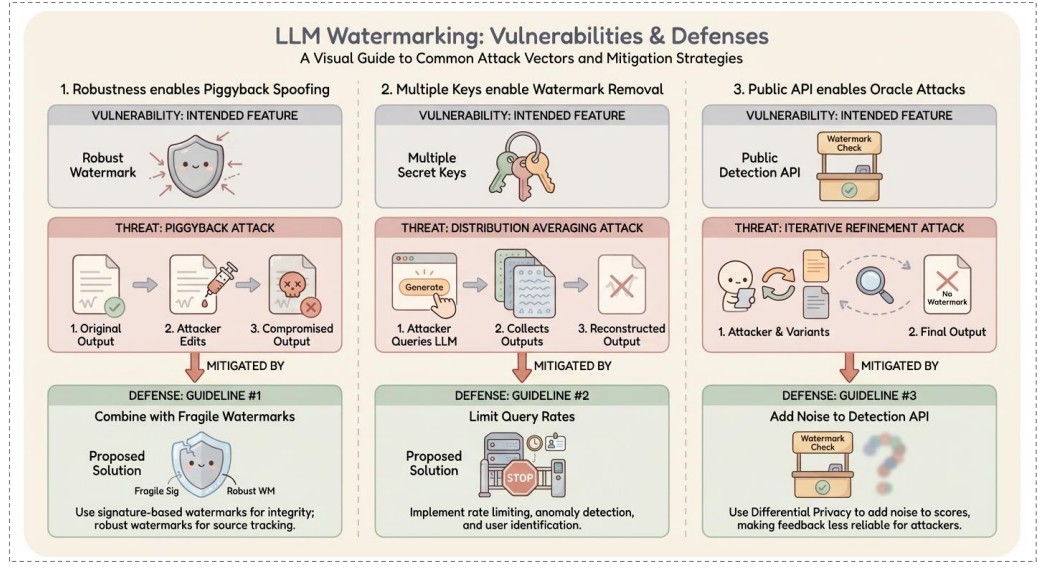

**Figure 25:** This is Case 4 from Blog. The original text is located at [Link].

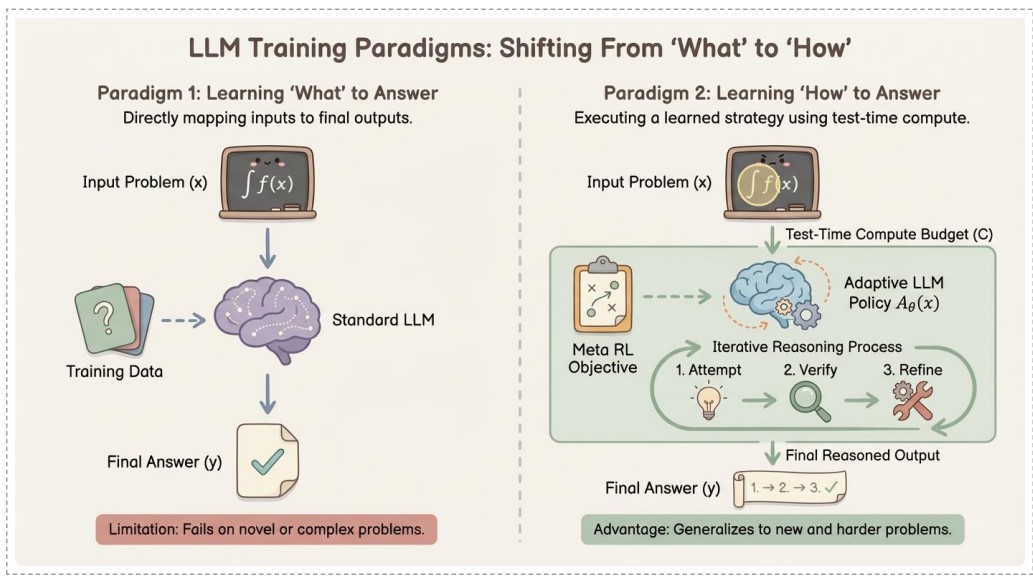

**Figure 26:** This is Case 5 from Blog. The original text is located at [Link].

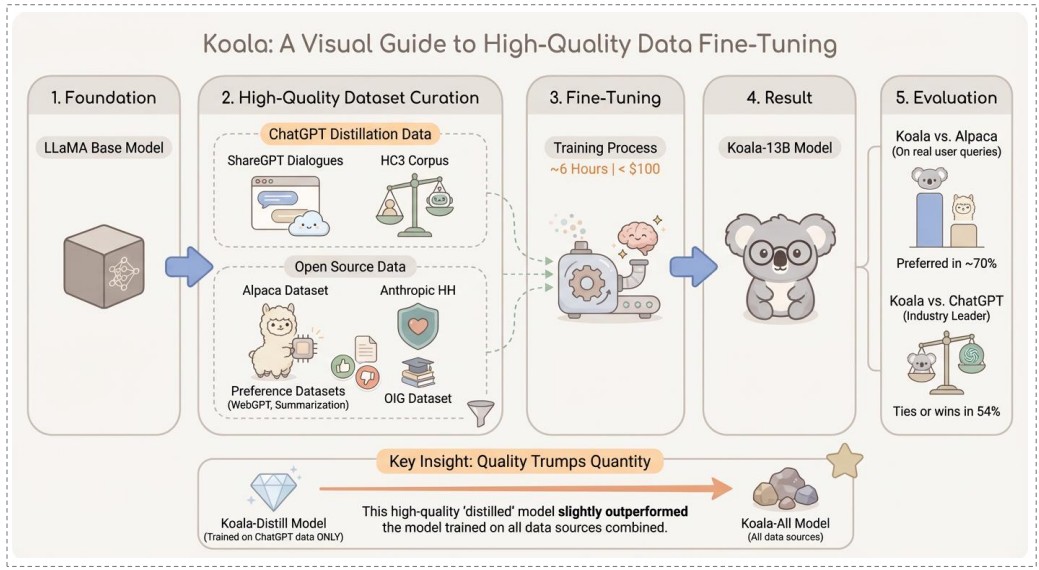

**Figure 27:** This is Case 6 from Blog. The original text is located at [Link].

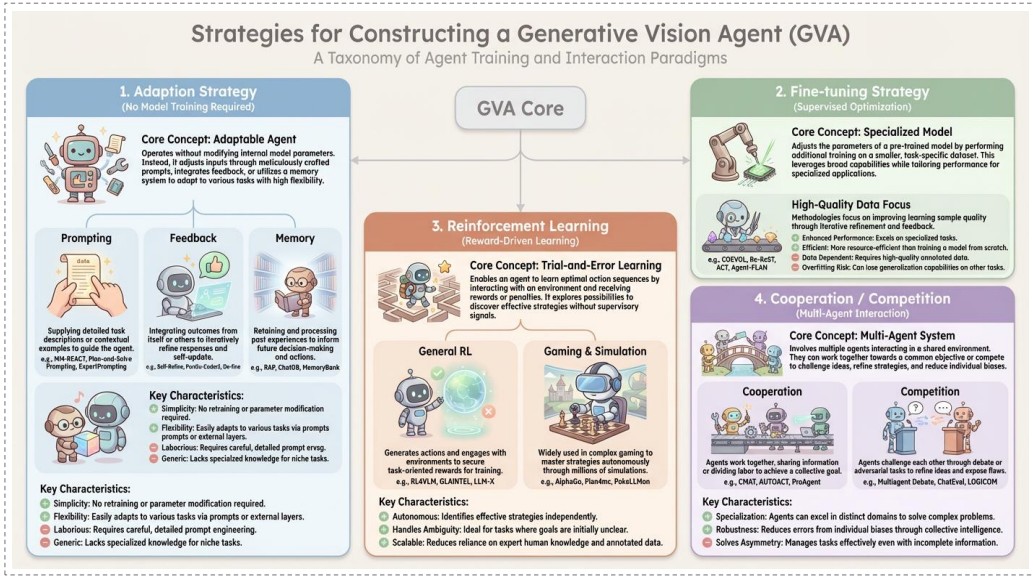

**Figure 28:** This is Case 1 from Survey. The original text is located at [Link].

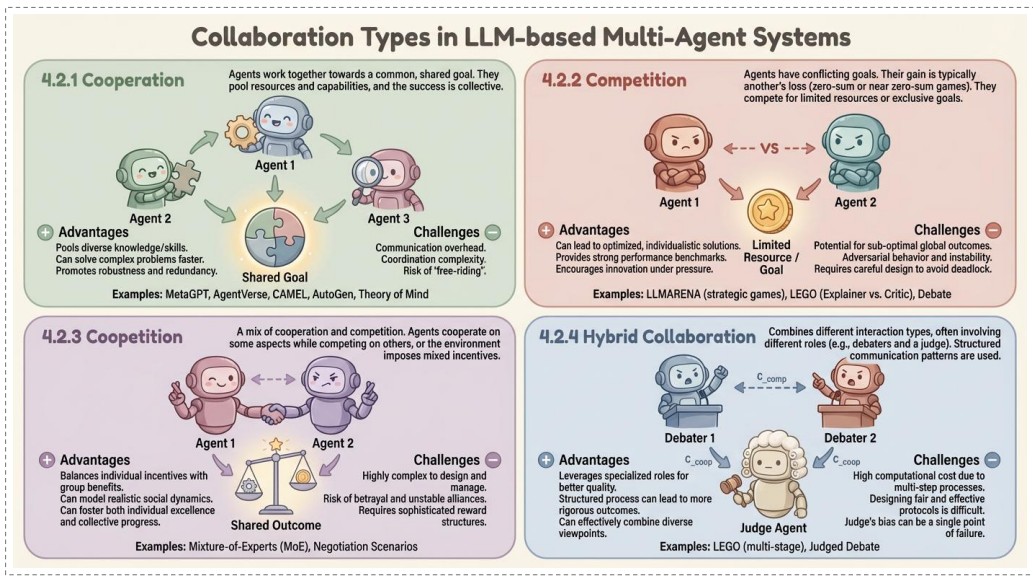

**Figure 29:** This is Case 2 from Survey. The original text is located at [Link].

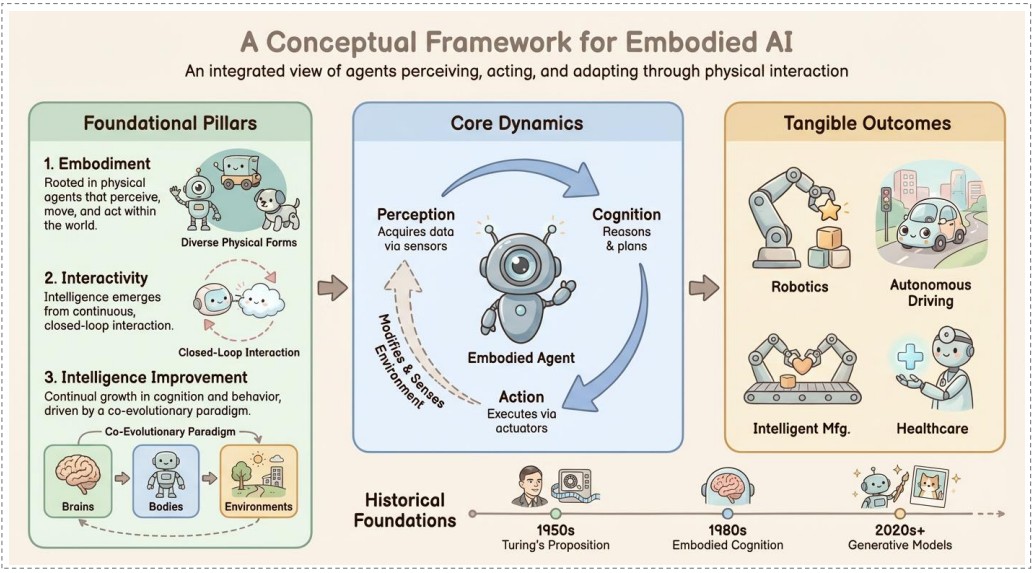

**Figure 30:** This is Case 3 from Survey. The original text is located at [Link].

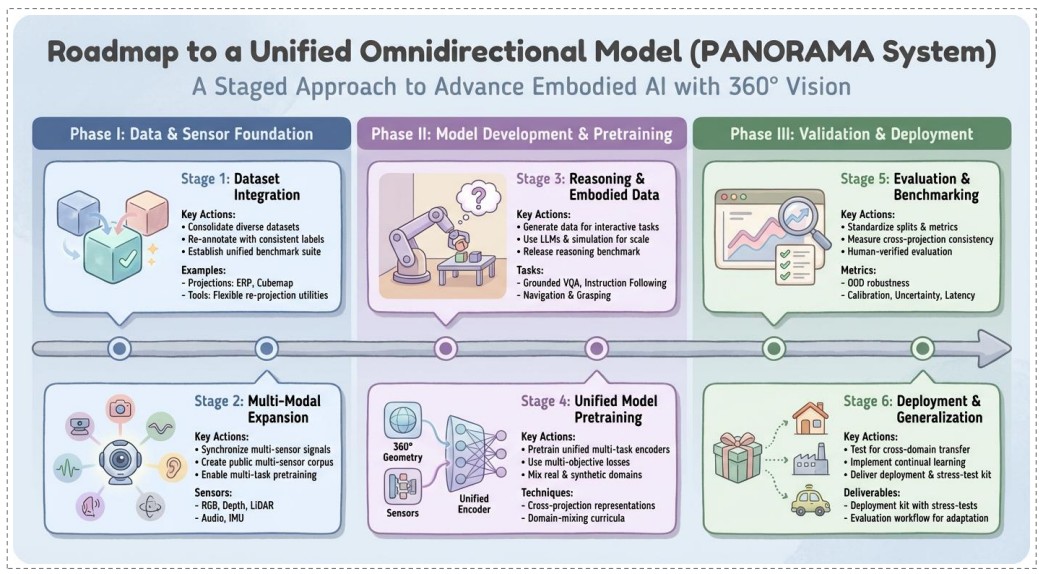

**Figure 31:** This is Case 4 from Survey. The original text is located at [Link].

