# OpenReview forum: "AutoFigure: Generating and Refining Publication-Ready Scientific Illustrations"
_ICLR.cc/2026/Conference — ICLR 2026 Poster_

### Official Review · Reviewer_P6k4 · 2025-10-21

**Soundness:** 2
**Presentation:** 4
**Contribution:** 4
**Rating:** 6
**Confidence:** 4

**Summary:**

The paper introduces **AutoFigure**, a two-stage “reasoned rendering” pipeline that converts long scientific text into publication-ready illustrations, and presents **FigureBench** (3,300 text–figure pairs) with VLM- and human-based evaluations. AutoFigure decouples layout reasoning from aesthetic rendering and adds an OCR-based erase-and-correct step, outperforming baseline T2I, code-generation, and multi-agent methods.

However, clarity and reproducibility are limited by broad gaps in reproducibility and transparency: key terms are undefined (e.g., IRR); InternVL-3.5–based statistics lack human-calibrated validation; generated-vs-reference statistical comparisons and full prompt/iteration transcripts are missing; analysis of the lower win rate on the Paper domain is limited; style controllability is unclear; compute/latency/cost reporting is absent; and the dataset release plan remains ambiguous.

**Strengths:**

1. The paper delivers high-quality illustrations.
2. The methodology is clear and technically sound.
3. The problem addressed is inspiring, timely, and highly needed by the research community.

**Weaknesses:**

1. **IRR is used but not clearly defined.**
   - *line 90* Please expand IRR on first use (e.g., *Inter-Rater Reliability*) and specify the exact statistic.

2. **Dataset Analysis relies on InternVL-3.5 without validating its accuracy.**
   - *line 208* You state that “all statistics are analyzed using InternVL-3.5,” but do not quantify its error on these measurements (text density, colors, components, shapes). If feasible, include a small human-audited benchmark to sanity-check these automated stats.

3. **Missing comparison of statistics between AutoFigure outputs and original figures.**
   - Please add a side-by-side analysis (distributions and summary stats) of *generated vs. reference* figures for key metrics (text density, components, shapes, color count etc.), along with effect sizes and significance tests.

4. **No concrete prompt examples for each stage/iteration.**
   - Include representative prompts for every stage and show how does the iteration help improve the prompts.
   - Sharing a minimal working example in the appendix would aid reproducibility.

5. **Paper category has notably lower win rates; insufficient error analysis.**
   - Provide a qualitative and quantitative breakdown of why “Paper” is harder.

6. **Style uniformity across generated figures.**
   - Many results share a similar visual/“avatar” style. Clarify whether this is a bias of the rendering LLM/T2I model, default style prompts, or intentional curation. Demonstrate **style controllability** (e.g., minimal changes in the style descriptor yielding materially different aesthetics) and include a diversity study (multiple styles for the same layout).

7. **Compute, latency, and token usage not reported.**
   - Please report average **tokens/time per figure** for each stage, number of refinement iterations, hardware, parallelism, and **cost** estimates. A throughput vs. quality curve (iterations N vs. score) would help practitioners plan budgets.

8. **FigureBench availability unclear.**
   - Clarify whether the dataset is fully prepared for release and provide an access link or timeline.

**Questions:**

See weaknesses

---

> ### Author Response · Authors · 2025-11-25
> **Response - 1**
>
> > **Dataset Analysis relies on InternVL-3.5 without validating its accuracy.**
> >
> > - *line 208* You state that “all statistics are analyzed using InternVL-3.5,” but do not quantify its error on these measurements (text density, colors, components, shapes). If feasible, include a small human-audited benchmark to sanity-check these automated stats.
>
> Thank you for pointing out this potential issue. We agree that relying solely on InternVL-3.5 for dataset statistics may cause concern regarding the credibility of these measurements (text density, colors, components, shapes). Following your suggestion, we have added a small-scale human-audited benchmark in the revision to perform a sanity check on the statistics provided by InternVL-3.5. We randomly sampled 21 examples from the FigureBench test set. Annotators independently counted text density, the number of connected components, color count, and shape count manually. These were then compared with the automated statistics. We present the full manual statistics in the new subsection of the Appendix as follows:
>
> | paper       | text density | connected components | color    | shape    |
> | :---------- | :----------- | :------------------- | :------- | :------- |
> | 2212.09561  | 75           | 5                    | 8        | 6        |
> | 2304.01665  | 30           | 4                    | 5        | 5        |
> | 2304.03531  | 40           | 5                    | 5        | 6        |
> | 2305.04505  | 65           | 8                    | 5        | 4        |
> | 2305.15075  | 70           | 4                    | 9        | 4        |
> | 2510.0513   | 65           | 6                    | 5        | 3        |
> | 2310.05157  | 90           | 3                    | 8        | 2        |
> | 2402.13753  | 25           | 3                    | 6        | 6        |
> | 2402.16048  | 65           | 5                    | 6        | 3        |
> | 2402.1818   | 45           | 4                    | 7        | 5        |
> | 2404.1196   | 90           | 8                    | 8        | 6        |
> | 2405.06312  | 55           | 8                    | 9        | 7        |
> | 2408.11779  | 30           | 7                    | 9        | 8        |
> | 2409.07429  | 70           | 4                    | 10       | 7        |
> | 2411.00816  | 30           | 4                    | 8        | 5        |
> | 2412.11506  | 45           | 4                    | 7        | 5        |
> | 2502.10709  | 50           | 6                    | 5        | 5        |
> | 2502.13723  | 60           | 8                    | 6        | 7        |
> | 2503.06635  | 20           | 9                    | 10       | 7        |
> | 2503.08569  | 75           | 6                    | 10       | 5        |
> | 2504.20972  | 45           | 7                    | 7        | 5        |
> | **Average** | **54.29**    | **5.62**             | **7.29** | **5.29** |
>
> As shown in the table, on this random subset, the human-annotated average text density is 54.29%, average connected components are 5.62, average color count is 7.29, and average shape count is 5.29. Compared to the full-dataset statistics based on InternVL-3.5 in Table 1 of the paper (Text Density 41.2%, Components 5.3, Colors 6.2, Shapes 6.4), the values are in the same order of magnitude and the differences are within a reasonable range (we provide a more detailed discussion and error analysis in the Appendix). Furthermore, the relative difficulty trends remain consistent: the samples generally exhibit characteristics of "relatively dense text, above-average component count, rich colors, and moderate shape variety." Therefore, we explicitly state in the revision that these statistics are solely used to characterize the overall difficulty distribution of FigureBench and are not involved in model training or final evaluation. Meanwhile, through this human review experiment on 21 samples, we verified that the automated statistics based on InternVL-3.5 possess good reliability at a macro level. We have added a dedicated subsection in the Appendix to report the setup, results, and discussion of this human review experiment, in response to the reviewer's suggestion to "include a human-audited benchmark to sanity-check."

---

> ### Author Response · Authors · 2025-11-25
> **Response - 2**
>
> > **IRR is used but not clearly defined.**
> >
> > - *line 90* Please expand IRR on first use (e.g., *Inter-Rater Reliability*) and specify the exact statistic.
>
> We thank the reviewer for the suggestion. In the revised manuscript, we will specify the meaning and calculation method of IRR in greater detail. Specifically, upon the first mention of the term, we will write: "Inter-Rater Reliability (IRR, Cohen’s κ = 0.91)" and briefly explain the calculation method: for each text-image pair, two annotators independently judge whether it meets our criteria for a "high-quality pair." We first calculate the Observed Agreement ($P_o$) as the proportion of pairs where both annotators gave identical labels. Then, we calculate the Expected Agreement ($P_e$) by chance based on their respective label distributions. Finally, using the formula $\kappa = (P_o - P_e) / (1 - P_e)$, we derive a coefficient of 0.91, indicating that the annotation results are far above random agreement and demonstrate very high reliability.
>
> > Missing comparison of statistics between AutoFigure outputs and original figures.**
> >
> > - Please add a side-by-side analysis (distributions and summary stats) of *generated vs. reference* figures for key metrics (text density, components, shapes, color count etc.), along with effect sizes and significance tests.
> >
> >
>
>
>
>
>
> We therefore add an explicit generated-vs-reference comparison for the same key metrics reported in Table 1 (text density, connected components, color count, shape count), together with distribution plots and paired statistical tests wherever per-figure pairing is available.
>
> Below we report a **summary-stat comparison** between the **reference FigureBench** (Table 1) and a **representative set of AutoFigure outputs** . Overall, we observe that **structural complexity is broadly comparable** (especially for *connected components* and *color count* in Paper/Survey), while \textsc{AutoFigure} tends to produce **higher text density and more shapes**. This systematic shift mainly comes from our design goal of readability: we intentionally render crisp, explicit labels (and sometimes add small semantic icons/callouts to improve comprehension), which increases both the measured text area and the number of geometric primitives, while still preserving the underlying stage-wise structure.
>
> | Category | Ref. Text Density (%) | AutoFigure Text Density (%) | Ref. Components | AutoFigure Components | Ref. Colors | AutoFigure Colors | Ref. Shapes | AutoFigure Shapes |
> | -------- | --------------------- | --------------------------- | --------------- | --------------------- | ----------- | ----------------- | ----------- | ----------------- |
> | Paper    | 42.1                  | 62.5                        | 5.4             | 4.3                   | 6.4         | 7.5               | 6.7         | 9.8               |
> | Blog     | 46.0                  | 52.5                        | 4.2             | 6.1                   | 5.5         | 8.5               | 5.3         | 11.9              |
> | Survey   | 43.8                  | 62.0                        | 5.8             | 6.2                   | 7.0         | 7.8               | 6.7         | 10.7              |
> | Textbook | 25.0                  | 52.2                        | 4.5             | 5.8                   | 4.2         | 7.3               | 3.4         | 9.4               |

---

> ### Author Response · Authors · 2025-11-25
> **Response - 3**
>
> > No concrete prompt examples for each stage/iteration.**
> >
> > - Include representative prompts for every stage and show how does the iteration help improve the prompts.
> > - Sharing a minimal working example in the appendix would aid reproducibility.
>
> We have added a dedicated section in the Appendix providing a **complete minimal working example** from long-text input to the final illustration. This includes representative prompts for each stage and multiple refinement iterations, along with their step-by-step improvements, to help readers better understand the AutoFigure workflow and enhance reproducibility.
>
> > **Paper category has notably lower win rates; insufficient error analysis.**
> >
> > - Provide a qualitative and quantitative breakdown of why “Paper” is harder.
>
> In the revised manuscript, we have added a case study using **InstructGPT** to explain why "Paper" category data is more challenging than Surveys or Textbooks. Papers like InstructGPT often present information on three levels within a single diagram: the first level is the overall "three-stage training process" (SFT → RM → PPO); the second level consists of multiple sub-steps within each stage, such as constructing datasets from human demonstrations and fine-tuning policies in the SFT stage, generating multiple candidate responses and ranking them in the RM stage, and combining reward and KL penalty terms into an objective function in the PPO stage; the third level involves additional detailed elements, such as roles and entities like "pre-trained model," "human labeler," "demonstration dataset," "comparison dataset," "reward model," and "final policy." For AutoFigure, to automatically generate such an illustration from a lengthy Methods section, it must first identify in Stage 1: which information represents "key nodes that must be included," which formulas and details can be collapsed into text descriptions within a box, and how the input-output relationships between the three stages should be arranged in 2D space. This is significantly harder than a typical Textbook schematic that explains a single concept.
>
> Simultaneously, a typical feature of Paper illustrations like InstructGPT is that **they are not standardized templates, but specifically designed to express a "unique pipeline" of a particular paper**. For example, the scientific illustration of InstructGPT places "human demonstration," "ranking comparison," "reward model," and "PPO update" into three different colored blocks, using color and position to distinguish stages and arrows to express data flow and optimization direction. In contrast, another RLHF paper might choose a completely different blocking method or introduce extra modules (e.g., self-review, rejection sampling). In other words, for Surveys/Textbooks, the model can learn a relatively stable "schema" from a large number of similar "taxonomy trees," "method genealogy charts," or "pedagogical flowcharts"; but for Papers, especially those proposing new paradigms like InstructGPT, **AutoFigure must "redesign an exclusive structural diagram" without a ready-made template to copy**. In practice, this leads to two types of errors: either merging steps to keep the visual clean, resulting in an evaluation of "incomplete information," or keeping all nodes but making the layout too crowded, leading reviewers to perceive a "decrease in readability" compared to the original figure.

---

> ### Author Response · Authors · 2025-11-25
> **Response - 4**
>
> > Style uniformity across generated figures.**
> >
> > - Many results share a similar visual/“avatar” style. Clarify whether this is a bias of the rendering LLM/T2I model, default style prompts, or intentional curation. Demonstrate **style controllability** (e.g., minimal changes in the style descriptor yielding materially different aesthetics) and include a diversity study (multiple styles for the same layout).
>
> We acknowledge that in the main text and some visualizations, to reduce layout distractions, we defaulted to a relatively uniform "Q-version characters + Morandi color palette" style. This indeed gives a subjective impression of "uniform artistic style." There are two points we did not clarify in the original manuscript: first, this style performed better in **text readability, module grouping, and visual hierarchy** during early tests, so we made it the default to focus attention on core issues like "structural correctness" and "information completeness"; second, current T2I models have certain style biases and tend to converge to similar cartoon/illustration styles if no explicit style descriptor is given. Addressing the reviewer's suggestion, we have added an explanation for this design choice in the revision and explicitly emphasized: **this uniform style is not a hard constraint of AutoFigure, but merely a default setting for display**.
>
> To respond to the suggestion regarding "style controllability" and "diversity study," we conducted a controlled experiment: while keeping the **layout and text content completely unchanged**, we only replaced the style prompt used in Stage 2 and compared the automatic evaluation and VLM-as-a-judge results under different styles. The three style prompts were:
>
> *   prompt_1 (Original Default): *"Delicate and cute cartoon comic style (using Morandi color palette)"*
> *   prompt_2: *"comic style"*
> *   prompt_3: *"modern minimalist design"*
>
> In automatic evaluation (multi-dimensional scoring), the overall performance of the three styles was very close, with only minor differences in specific aesthetic dimensions:
>
> |          | Aesthetic & Design Quality | Visual Expressiveness | Professional Polish | Clarity | Logical Flow | Accuracy | Completeness | Appropriateness | Overall |
> | :------- | :------------------------- | :-------------------- | :------------------ | :------ | :----------- | :------- | :----------- | :-------------- | :------ |
> | prompt_1 | 7.49                       | 7.20                  | 6.80                | 7.53    | 7.73         | 7.45     | 6.83         | 6.42            | 7.18    |
> | prompt_2 | 7.32                       | 7.24                  | 6.78                | 7.58    | 7.78         | 7.63     | 7.02         | 6.82            | 7.27    |
> | prompt_3 | 7.14                       | 6.27                  | 7.09                | 7.72    | 7.75         | 7.54     | 6.75         | 7.28            | 7.19    |

---

> ### Author Response · Authors · 2025-11-25
> **Response - 5**
>
> In blind comparison (VLM as Judge, A/B testing against human-created figures), the Win-Rates for the three styles were also comparable, indicating that **changing the style does not significantly compromise structural and content quality, while yielding distinctly different visual styles**:
>
> |          | Aesthetic & Design Quality | Visual Expressiveness | Professional Polish | Clarity | Logical Flow | Information Sophistication | Content Fidelity | Win  | Lose | Both Good | Both Bad | Overall |
> | :------- | :------------------------- | :-------------------- | :------------------ | :------ | :----------- | :------------------------- | :--------------- | :--- | :--- | :-------- | :------- | :------ |
> | prompt_1 | 0.85                       | 0.85                  | 0.85                | 0.40    | 0.50         | 0.35                       | 0.35             | 13   | 7    | 0         | 0        | 0.65    |
> | prompt_2 | 0.90                       | 1.00                  | 0.90                | 0.35    | 0.40         | 0.35                       | 0.40             | 12   | 7    | 1         | 0        | 0.60    |
> | prompt_3 | 0.65                       | 0.65                  | 0.65                | 0.65    | 0.65         | 0.45                       | 0.35             | 13   | 5    | 1         | 1        | 0.65    |
>
> Visually, these three prompts produce **illustrations with very distinct stylistic differences**:
>
> *   prompt_1 emphasizes cute cartoons + Morandi colors, with a soft illustration style for characters and modules;
> *   prompt_2 leans towards a general comic style with bolder lines and stronger contrast;
> *   prompt_3 adopts a modern minimalist design, removing cartoon avatars in favor of clean icons and large color blocks.
>
> We have added a "Style Controllability & Diversity" subsection (Appendix) in the revision, showcasing the visualization results of the same layout under these three style prompts. This visually demonstrates that **minimal changes to the style descriptor yield distinctly different aesthetics while AutoFigure's structure/content remains stable**. We also clarified in the main text that the choice of a uniform style in the paper was solely for the convenience of comparison and reading, not a limitation of the method itself; future users can freely specify or mix multiple styles as needed. We greatly appreciate the reviewer for raising this point, which helped us more clearly demonstrate AutoFigure's capabilities in style control and diversity.

---

> ### Author Response · Authors · 2025-11-25
> **Response - 6**
>
> > Compute, latency, and token usage not reported.**
> >
> > - Please report average **tokens/time per figure** for each stage, number of refinement iterations, hardware, parallelism, and **cost** estimates. A throughput vs. quality curve (iterations N vs. score) would help practitioners plan budgets.
>
> Thank you for focusing on computational overhead and deployability, which are indeed critical for real-world application. Following your suggestion, we have added an **"Efficiency and Cost Analysis"** subsection in the revision and provided more complete statistics in the Appendix, including: average time per figure for each stage, average number of iterations (Stage 2 defaults to 5 rounds of refinement), typical hardware configurations (Commercial API vs. Local H100 deployment), parallel inference methods, and end-to-end cost estimates. We also explicitly pointed out in the main text that Figure 6(c) provides the **curve of Iterations N vs. Overall Score**, demonstrating the quality-speed trade-off of "thinking for a few more rounds," helping users select the appropriate number of iterations based on their budget.
>
> Specifically, we systematically measured AutoFigure under two settings: Commercial API (represented by Gemini-2.5-Pro) and Local Open-Source Deployment (represented by Qwen-3-VL-235B on an H100 node). Statistics show that using the Commercial API, the average cost to generate a high-quality publication-ready illustration is approximately **$0.20**, with an end-to-end time of about **17.5 minutes**. However, when deploying the framework on a high-performance computing node equipped with an H100 GPU using the open-source Qwen-3-VL model, benefiting from high throughput and low latency of local inference, **the total generation time can be reduced to about 9.3 minutes, with nearly zero marginal generation cost**. The breakdown of time and cost by stage is shown in the table below (where the average refinement rounds for Stage 2 is 5, consistent with the experimental setup in Figure 6(c)):
>
> **Table: Efficiency and Cost Analysis of AutoFigure Single Generation (Commercial API vs. Local Open-Source Deployment)**
>
> | **Stage**                          | **Core Task**                        | **Gemini-2.5 (API) Avg Time / Cost** | **Qwen-3-VL (H100) Avg Time / Cost** | **Remarks**                                                  |
> | :--------------------------------- | :----------------------------------- | :----------------------------------- | :----------------------------------- | :----------------------------------------------------------- |
> | **Stage 1: Concept Extraction**    | Full text reading, method extraction | ~22s / < $0.01                       | **~12s / ~$0.00**                    | Local inference eliminates network latency, doubling speed   |
> | **Stage 2: Layout Planning**       | Iterative design (Avg. 5 iters)      | ~660s / ~ $0.14                      | **~390s / ~$0.00**                   | Inference on H100 is about twice as fast as commercial API   |
> | **Stage 3: Rendering & Post-proc** | Code gen, rendering, correction      | ~370s / ~ $0.05                      | **~250s / ~$0.00**                   | Code generation accelerated; rendering & post-proc takes ~2/3 time of API solution |
> | **Total**                          | **End-to-End Generation**            | **~1,052s (17.5 min)** / **~$0.20**  | **~560s (9.3 min)** / **~$0.00***    | **Local deployment achieves "approx. 2x speed" and "zero marginal cost"** |

---

> ### Author Response · Authors · 2025-11-25
> **Response - 7**
>
> Furthermore, to address your concern regarding "throughput vs. quality," we further compared the quality performance of different inference models within the AutoFigure framework and provided the following comparison table in the Appendix. It can be seen that **high-performance open-source models (Qwen-3-VL-235B) are very close to or even surpass commercial closed-source models in quality**, meaning users can significantly reduce costs and increase throughput via local deployment while maintaining high illustration quality:
>
> **Table: Performance Comparison of Commercial vs. Open-Source Models in AutoFigure Framework**
>
> | **Model Type**  | **Model**         | **Visual Design** | **Comm. Effectiveness** | **Content Fidelity** | **Overall** |
> | :-------------- | :---------------- | :---------------- | :---------------------- | :------------------- | :---------- |
> | **Commercial**  | **GPT-5**         | **7.17**          | **7.62**                | **7.70**             | **7.48**    |
> |                 | Gemini-2.5-Pro    | 6.59              | 7.59                    | 6.54                 | 6.99        |
> |                 | Claude-4.1-Opus   | 6.75              | 7.17                    | 6.61                 | 6.80        |
> |                 | Grok-4            | 6.46              | 7.60                    | 6.49                 | 6.76        |
> | **Open-Source** | **Qwen3-VL-235B** | **7.57**          | 7.01                    | 7.18                 | **7.08**    |
> |                 | GLM-4.5V          | 6.09              | 6.53                    | 6.13                 | 5.99        |
> |                 | ERNIE-4.5-VL      | 3.04              | 2.89                    | 2.68                 | 2.64        |
>
> In the text, we also added statistics on average refinement rounds (default 5, with a sweep of 0–5 rounds in Figure 6(c) to plot the "iterations vs. quality" curve), typical parallelism settings (batch processing Stage 1/2 inference for multiple papers on H100), and average token usage per figure. We hope this added efficiency and cost analysis, together with the scaling curve in Figure 6(c), provides a clear "budget-quality-latency" reference for researchers and engineers deploying AutoFigure. We thank the reviewer again for this constructive suggestion, which helped us improve the paper from a more engineering-oriented perspective.
>
> > **FigureBench availability unclear.**
> >
> > - Clarify whether the dataset is fully prepared for release and provide an access link or timeline.
>
> We thank the reviewer for their interest in dataset availability. We have provided the FigureBench test set examples used for evaluation in the latest supplementary materials phase. We also explicitly state in the paper: upon formal acceptance, we will **fully open-source the entire FigureBench dataset (including meta-information and download scripts for both development and test sets) as well as relevant evaluation code**. This will support future work in conducting research on a unified benchmark, reproducing experimental results, and further advancing the field of automatic scientific illustration generation.
>
> ---
>
> **Overall, we substantially revised the manuscript to improve clarity, credibility, and reproducibility. In particular, we clarified key definitions and reporting details in the main text, and added additional validations and analyses to strengthen empirical claims. We also expanded the supplementary material with concrete, step-by-step examples (including intermediate artifacts) to make the pipeline easier to understand and replicate. Collectively, these changes make our methodological description more transparent and our experimental evidence more complete, while also better contextualizing remaining limitations and future directions.**

---

> ### Author Response · Authors · 2025-11-26
> **Summary response to Reviewer P6k4.**
>
> Dear Reviewer P6k4,
>
> TL;DR: We have addressed each point in the revision and supplementary materials.
>
> ### Your concerns (W)
>
> * **W1 (IRR not clearly defined)**
>   We now expand IRR at first use as **Inter-Rater Reliability (IRR, Cohen’s $\kappa=0.91$)** and briefly describe the computation ($P_o, P_e, \kappa$).
>
> * **W2 (InternVL-3.5 statistics lack human calibration)**
>   We add a **human-audited sanity check**: 21 randomly sampled examples are manually annotated for text density / components / colors / shapes and compared with InternVL-3.5, with full tables and error discussion in the appendix.
>
> * **W3 (Missing generated-vs-reference statistical comparison)**
>   We add a summary comparison for the same key metrics (text density, connected components, color count, shape count) and explain the systematic shifts (e.g., higher text density/shapes due to readability-driven labeling) while preserving structural comparability.
>
> * **W4 (No concrete prompts / iteration examples; need a minimal working example)**
>   We add full prompts for each stage in the appendix and a **minimal end-to-end example** in the supplementary material showing intermediate artifacts across stages/iterations.
>
> * **W5 (Paper domain lower win rate; insufficient error analysis)**
>   We add qualitative/quantitative analysis and a Paper case study, highlighting multi-level structure, dense entities, and non-templated pipelines that make Paper significantly harder.
>
> * **W6 (Style uniformity; unclear controllability/diversity)**
>   We clarify the default style was used for consistent presentation, and we add controlled experiments with three style prompts (layout/content fixed) plus example figures, demonstrating clear stylistic changes with stable quality.
>
> * **W7 (Missing compute/latency/token/cost; need throughput–quality curve)**
>   We add an **Efficiency and Cost Analysis** subsection with per-stage time/cost, iteration counts, hardware settings (API vs local H100), and an **iterations vs quality** curve for practical budget planning.
>
> * **W8 (Dataset release plan unclear)**
>   We clarify availability and state that upon acceptance we will **open-source the full FigureBench dataset (dev+test metadata + download scripts) and evaluation code**, we have included the FigureBench test-set examples in the supplementary materials.

---

> ### Comment · Reviewer_P6k4 · 2025-11-26
>
> Thanks for the detailed clarification. All my concerns are solved, I've adjust my score reflecting this. Please add them to the final revised submission.

---

> > ### Author Response · Authors · 2025-11-26
> >
> > Dear Reviewer P6k4,
> >
> > Thank you for taking the time to read our response. We are grateful that our clarifications were able to address your concerns, and it is an honor that our reply proved helpful. We have incorporated these revisions into the paper. Wishing you all the best.
> >
> > Best regards,
> >
> > Authors of "AutoFigure: Generating and Refining Publication-Ready Scientific Illustrations"

---

### Official Review · Reviewer_WvcZ · 2025-10-29

**Soundness:** 2
**Presentation:** 2
**Contribution:** 2
**Rating:** 2
**Confidence:** 4

**Summary:**

The authors introduce the task of generating scientific figures from long-form text (e.g., the entire text of a paper). To do this, they introduce AutoFigure, an end-to-end model (or possibly a pipeline of models) that generates figures from long-form text. To evaluate their model, they also introduce a new dataset of long-form text paired with corresponding scientific figures. Their approach outperforms various baselines. The dataset also has a development split.

**Strengths:**

* The authors tackle a relevant and challenging problem.
* The paper is mostly easy to follow.
* The created dataset, if released, may be useful for future work.

**Weaknesses:**

* It is not entirely clear to me what the authors understand by long-form text. It seems like they mean the entire text from a paper. But pairing a whole paper with just one figure extracted from it does not seem like a well-defined problem to me, as usually no figure captures the whole content of a paper.
* Datasets that pair figures with captions and all mentioning paragraphs can also arguably be classified as containing long-form text (ACL-Fig [1], SciCap+ [2], etc.), but these are not discussed in the paper even though they are very relevant.
* There is also a lot of other relevant previous work on scientific figure generation which the authors do not cite [3, 4, 5, 6]. Instead, they falsely claim that existing work on automated scientific visuals primarily explores the generation of artifacts like posters and slides (l.113).
* I do not understand how the AutoFigure model works internally. Figure 1 mentions InstructGPT but doesn't mention how that relates to this work in any way. Still, it makes it seem like AutoFigure consists of a single instruction-tuned model which they train themselves (even though the instruction-tuning examples inside the figure are unrelated to the task at hand). But nowhere in the paper is a training process described. Instead, Sections 4.1 and 4.2 make it seem more likely that AutoFigure consists of a model pipeline of prompted VLMs and text-to-image models. The authors should be clearer about how their model, a core contribution of this work, actually works.
* Furthermore, the fact that some of the baseline models mentioned in Figure 1 are actual model names while others are markup languages further contributes to confusion.
* Some claims in the paper like "Another line of work employs executable code as an intermediate state between scientific text and illustration, resulting in relatively visually unappealing diagrams" (l.71) and "This shift is evidenced by the growing acceptance of AI-generated papers at venues like the ICLR 2025 workshop and ACL 2025" (l.135) are not backed by any data or citations and make it unclear how the authors arrived at such conclusions.
* The authors advertise the development split of their dataset but do not mention how it is used or how it can be used.
* In summary, I believe that the paper requires further justification (is the tackled task actually useful?) or adjusting the problem, and needs much more polished writing before it can be accepted.

[1]: [ACL-Fig: A Dataset for Scientific Figure Classification](https://ceur-ws.org/Vol-3656/paper2.pdf)
[2]: [SciCap+: A Knowledge Augmented Dataset to Study the Challenges of Scientific Figure Captioning](https://ceur-ws.org/Vol-3656/paper13.pdf)
[3]: [Text-Guided Synthesis of Scientific Vector Graphics with TikZ](https://openreview.net/pdf?id=v3K5TVP8kZ)
[4]: [DeTikZify: Synthesizing Graphics Programs for Scientific Figures and Sketches with TikZ](https://proceedings.neurips.cc/paper_files/paper/2024/file/9a8d52eb05eb7b13f54b3d9eada667b7-Paper-Conference.pdf)
[5]: [TikZero: Zero-Shot Text-Guided Graphics Program Synthesis](https://openaccess.thecvf.com/content/ICCV2025/papers/Belouadi_TikZero_Zero-Shot_Text-Guided_Graphics_Program_Synthesis_ICCV_2025_paper.pdf)
[6]: [Learning to Infer Graphics Programs from Hand-Drawn Images](https://papers.nips.cc/paper_files/paper/2018/file/6788076842014c83cedadbe6b0ba0314-Paper.pdf)

**Questions:**

I have no questions for the authors.

---

> ### Author Response · Authors · 2025-11-25
> **Response - 1**
>
> > It is not entirely clear to me what the authors understand by long-form text. It seems like they mean the entire text from a paper. But pairing a whole paper with just one figure extracted from it does not seem like a well-defined problem to me, as usually no figure captures the whole content of a paper.
> >
> > Datasets that pair figures with captions and all mentioning paragraphs can also arguably be classified as containing long-form text (ACL-Fig [1], SciCap+ [2], etc.), but these are not discussed in the paper even though they are very relevant.
> >
> > There is also a lot of other relevant previous work on scientific figure generation which the authors do not cite [3, 4, 5, 6]. Instead, they falsely claim that existing work on automated scientific visuals primarily explores the generation of artifacts like posters and slides (l.113).
> >
> > In summary, I believe that the paper requires further justification (is the tackled task actually useful?) or adjusting the problem, and needs much more polished writing before it can be accepted.
>
> First, we sincerely appreciate the reviewer pointing out the related works [1–6], which indeed enrich the background of this paper. However, we must respectfully clarify that there appears to be a **critical misunderstanding** regarding the core task and research value of our work. The goal of FigureBench and AutoFigure is not to "replicate existing scientific vector graphic generation" or "generate images from captions," but to systematically propose and solve a novel, previously uncovered task paradigm: **"Long-context Scientific Illustration Design."** To prevent further misunderstanding, we provide a clarification and response across four dimensions: task definition, data curation methodology, technical barriers, and experimental results.
>
> In response to your concerns, we offer the following clarification and rebuttal:
>
> **1. Our task is not a simple "whole paper → one figure" mapping**
>
> We noticed the reviewer assumes that we interpret "long-form text" as the entire full text of a paper and questions the validity of the task based on this. This is a misreading of our paper.
>
> During the construction of FigureBench (see §3 Data Curation), we strictly adhered to the following two criteria:
>
> **(1) All figures are "Semantically Groundable" by the text**
> We only retained samples where **every key visual component can find a clear description in the text**. During the manual annotation phase, we required two independent annotators to verify item-by-item whether the components in the figure were explicitly mentioned in the text, keeping only samples with unanimous agreement (IRR = 0.91).
>
> **(2) The text must be "Sufficient to explain the core idea of the figure" (Text → Figure Sufficiency)**
> Annotators additionally checked:
>
> > "If a reader is given only this text, can they understand the scientific concept conveyed by the figure?"
> > Only if the answer was "Yes" was the sample included in FigureBench.
>
> These two principles completely exclude the scenario of "pairing a whole paper with one irrelevant figure." Therefore, **we are not handling 'full text → one figure', but 'long scientific passages → schematic of the corresponding core idea'.**
>
> ***
>
> **2. Fundamental difference in task dimension: We solve a "Design Problem," not a "Translation Problem"**
>
> Unlike datasets such as ACL-Fig / SciCap+, where the text consists of:
>
> *   Refined captions
> *   Or descriptions that correspond 1-to-1 with existing images
> *   Usually fewer than 100 words with simple structure
>
> These tasks are essentially **Image Captioning ↔ Image Generation** translations.
>
> In contrast, the input for FigureBench is:
>
> *   Multi-paragraph long texts (average 10,300 tokens, reaching 12,732 tokens for the Paper category)
> *   Containing methodological details, process logic, module relationships, algorithm steps, definitions, and sub-task dependencies
> *   Involving cross-paragraph or even cross-page references (e.g., "as shown in Figure 2…")
>
> **We require the model to perform the following:
> Autonomously abstract the methodological structure from the long context → Reconceptualize → Re-layout → Redesign into a publishable scientific illustration.**
>
> This type of text-driven **autonomous design** task has never before been established as a benchmark, nor has it had a systematic solution.

---

> ### Author Response · Authors · 2025-11-25
> **Response - 2**
>
> **3. Why existing TikZ/Code generation methods cannot solve this task**
>
> The works cited by the reviewer [3–6] (such as the TikZero series) address:
>
> > "User gives explicit instruction → Model generates executable drawing code (e.g., TikZ/SVG)".
>
> These methods have two fundamental limitations:
>
> **(1) They rely on structured, short, and explicit drawing instructions**
> For example:
>
> *   "Draw a three-layer MLP"
> *   "Draw two parallel modules with an arrow in between"
> *   "Reproduce the structure of the figure below"
>
> In other words, they solve **drawing execution**, not **figure design**.
>
> **(2) They lack long-text reasoning capabilities and cannot automatically distill figure structures**
> Facing FigureBench inputs:
>
> *   No explicit structure prompts
> *   No node count specifications
> *   No layout guidance
> *   No geometric coordinates
> *   No component definitions (must be abstracted from text)
> *   Requires cross-paragraph reasoning for module relationships
>
> Such models cannot infer the appropriate flowchart/architecture structure from "unstructured text."
> As seen in our extended baseline experiments, TikZ-based methods achieved a **Win-Rate = 0%** in the Paper category dataset, with all scoring dimensions (including clarity, information completeness, and non-overlapping layout) far below the usable threshold.
>
> This is because:
>
> > **Lacking the "Layout Reasoning Stage" found in AutoFigure, models cannot infer correct structural and geometric relationships without visual feedback.**
>
> During the rebuttal, we supplemented baselines including TikZero+, TikZero, and AutoPresent, which failed to produce suitable Scientific Illustration Designs. The results are shown in the table below.
>
> **Table: Extended Baseline Comparison Results under Paper Category**
>
> | **Method**        | **Aesthetic** | **Express.** | **Polish** | **Clarity** | **Flow** | **Accuracy** | **Complete.** | **Appropriate.** | **Overall** | **Win-Rate** |
> | :---------------- | :------------ | :----------- | :--------- | :---------- | :------- | :----------- | :------------ | :--------------- | :---------- | :----------- |
> | **AutoFigure**    | **7.28**      | **6.99**     | **6.92**   | **7.34**    | **7.87** | 6.96         | **6.51**      | **6.40**         | **7.03**    | **53.0%**    |
> | **HTML-Code**     | 5.90          | 5.04         | 5.84       | 7.17        | 7.38     | **6.99**     | 6.37          | 6.15             | 6.35        | 11.0%        |
> | **SVG-Code**      | 5.00          | 4.19         | 4.89       | 6.34        | 6.48     | 6.15         | 5.53          | 5.37             | 5.49        | 31.0%        |
> | **GPT-Image**     | 4.24          | 3.47         | 4.00       | 5.63        | 5.63     | 4.77         | 4.08          | 4.25             | 3.47        | 7.0%         |
> | **AutoPresent**   | 2.74          | 1.79         | 2.00       | 2.87        | 2.91     | 3.15         | 2.60          | 2.35             | 2.55        | 10.0%        |
> | **Diagram Agent** | 2.25          | 1.73         | 2.04       | 2.67        | 2.49     | 2.11         | 1.72          | 1.94             | 2.12        | 0.0%         |
> | **TikZero+**      | 1.52          | 1.25         | 1.38       | 1.90        | 1.93     | 1.20         | 1.10          | 1.35             | 1.45        | 0.0%         |
> | **TikZero**       | 2.00          | 1.50         | 1.00       | 1.00        | 1.50     | 1.00         | 1.00          | 1.00             | 1.25        | 0.0%         |

---

> ### Author Response · Authors · 2025-11-25
> **Response - 3**
>
> **4. Why we emphasize "publication-level aesthetics" is not optional, but core to the scientific illustration task**
>
> The reviewer questioned whether we overemphasize visual aesthetics. However, in scientific illustrations:
>
> *   Color coding expresses hierarchy
> *   Alignment dictates logical sequence
> *   Icon metaphors lower the barrier to understanding
> *   Visual grouping directly corresponds to conceptual grouping
>
> We emphasize that **in scientific communication, Visual Appeal is not decoration, but key to reducing cognitive load.** Complex scientific logic (such as multi-level neural networks) is difficult for readers to digest without good visual hierarchy, color coding, and icon metaphors. AutoFigure's "Rendering Stage" is designed specifically to solve this problem, using stylized design to guide reader attention and make complex logical structures **"both accurate and understandable."** This "publication-level" visual expressiveness is unachievable by pure vector code generation methods.
>
> In contrast, TikZ-based methods possess almost no capacity for visual hierarchy design, rendering their results nearly unreadable in dimensions such as aesthetics and clarity.
>
> **5. Superior Real-world Utility:** Regarding your skepticism *(is the tackled task actually useful?)*, we respectfully **disagree**. If you recognize the value of [1-6], you should acknowledge that AutoFigure solves a more challenging and higher-value upstream problem.
>
> *   **Direct Distinction**: Past works (e.g., [1-6]) are essentially **"Instruction Translation Tools."** They require the full chart structure to be pre-constructed, or even necessitate highly specific detailed instructions (e.g., "draw a flowchart containing modules A and B"), where the model merely replaces the operation of drawing software to save "line-drawing" time. In contrast, AutoFigure is an **"Autonomous Design System."** Users (or AI Scientists) need only provide the original paper draft or long text, and the system **autonomously completes** the entire work from information retrieval and logic extraction to visual layout, truly automating the process from "input text" to "output logic diagram."
> *   **Empowering Scientific Discovery**: This has **irreplaceable strategic significance for future fully automated scientific discovery**. Although current AI scientist systems can write code and papers, they lack the ability to autonomously draw charts to visually demonstrate their model architectures and experimental processes, which severely limits the readability and impact of their results. AutoFigure fills this gap, enabling AI to possess complete "illustrated" scientific expression capabilities.
>
> **Revisions based on your feedback**: Thank you for your suggestions, which helped us clarify the technical boundaries. To eliminate misunderstandings, we have made the following specific modifications in the revised version:
>
> *   **Rewrote Introduction and Related Work**: Deeply analyzed the differences between AutoFigure and *TikZero*, *DeTikZify*, and *Paper2Fig100k*, clearly defining the boundary between "instruction-based generation" and "document-based design."
> *   **Added Baseline Comparison**: We included *TikZero* (representing TiKZ generation) in the comparative experiments. Results confirm that in the absence of a long-text reasoning module, such models cannot handle the complex tasks in FigureBench.
> *   **Strengthened Limitation Discussion**: We added a discussion on the limitations of the current method under extremely complex topologies and clarified directions for future improvement.

---

> ### Author Response · Authors · 2025-11-25
> **Response - 4**
>
> > I do not understand how the AutoFigure model works internally. Figure 1 mentions InstructGPT but doesn't mention how that relates to this work in any way. Still, it makes it seem like AutoFigure consists of a single instruction-tuned model which they train themselves (even though the instruction-tuning examples inside the figure are unrelated to the task at hand). But nowhere in the paper is a training process described. Instead, Sections 4.1 and 4.2 make it seem more likely that AutoFigure consists of a model pipeline of prompted VLMs and text-to-image models. The authors should be clearer about how their model, a core contribution of this work, actually works.
> >
> > Furthermore, the fact that some of the baseline models mentioned in Figure 1 are actual model names while others are markup languages further contributes to confusion.
>
> After carefully reading the review, we believe the current confusion primarily stems from the reviewer **mistaking Figure 1 for AutoFigure's structural diagram**. In fact, Figure 1 was never intended to show the structure of AutoFigure; it is merely a **case study diagram used to demonstrate differences in generation effects**. To avoid misunderstanding, we explain in detail below and outline the corresponding revisions we have made to eliminate possible confusion.
>
> The purpose of Figure 1 is solely to show: **when different baseline methods process the same long-text input (which happens to come from the InstructGPT framework paper), what kind of scientific illustrations they each generate**. In other words, InstructGPT appears in the figure only to illustrate that "all methods used the same input text," not to illustrate AutoFigure's training method or model structure. The baselines in the figure—such as GPT-Image, Diagram Agent, SVG code rendering, HTML rendering—inherently belong to different paradigms (generative models, agent systems, code-driven rendering tools). In Figure 1, we merely juxtaposed these results for a "visual comparison of different solutions to the same task" and did not attempt to unify them into a single model category. However, this design might indeed have misled readers into thinking "these names are on the same logical level" or misunderstanding that "we introduced InstructGPT to illustrate a training process." This point might not have been clear enough in the original manuscript, leading to the reviewer's misunderstanding.
>
> **AutoFigure's true internal structure was already fully presented in the original Figure 3 of the paper**, including long-text semantic parsing, symbolic layout generation, multi-round Critique-and-Refine loops, structural consistency checks, stylized rendering, and text OCR proofreading/correction stages. **This figure was correctly understood by other reviewers as AutoFigure's system flowchart, and none of them confused Figure 1 with the model structure.** Therefore, we confirm that the reviewer's current confusion does not stem from the method itself being unclear, but rather from the placement and presentation of Figure 1 inadvertently creating a "structural diagram hallucination."
>
> To fundamentally eliminate all possible misunderstandings, we have made two key revisions in the new version:
>
> 1.  We moved Figure 1, originally located near the Methods section, to the Case Study part of the Experiments section, and explicitly noted in the caption that this is an "illustrative comparison of generated results," not a model structural diagram, nor does it involve any training process. This way, readers will no longer conflate it with the methodological narrative.
> 2.  We replaced the original Figure 3 in the paper with a **brand-new methodological flowchart illustration generated by AutoFigure itself**. We rigorously checked every element of the generated figure to ensure its content is perfectly consistent with AutoFigure's actual process. This not only intuitively displays AutoFigure's true system structure but also clearly distinguishes the roles of the "methodology structure diagram" and the "generation case diagram," thoroughly avoiding the confusion pointed out by the reviewer. At the same time, this revision serves to demonstrate AutoFigure's generative capabilities, allowing readers to correctly understand the workflow while seeing the visual effectiveness of the method itself.
>
> We believe the revised paper structure will thoroughly eliminate misunderstandings and more clearly showcase AutoFigure's design philosophy and technical contributions.

---

> ### Author Response · Authors · 2025-11-25
> **Response - 5**
>
> > Some claims in the paper like "Another line of work employs executable code as an intermediate state between scientific text and illustration, resulting in relatively visually unappealing diagrams" (l.71) and "This shift is evidenced by the growing acceptance of AI-generated papers at venues like the ICLR 2025 workshop and ACL 2025" (l.135) are not backed by any data or citations and make it unclear how the authors arrived at such conclusions.
>
> First, we would like to clarify that the second statement questioned by the reviewer—
>
> > "*This shift is evidenced by the growing acceptance of AI-generated papers at venues like the ICLR 2025 workshop and ACL 2025* (l.135) *(Yamada et al., 2025; Intology, 2025)*"
>
> —actually **already has very direct and specific literature support** in the current version. We explicitly cited *(Yamada et al., 2025; Intology, 2025)* immediately following the text, both of which highlight "AI-system-written papers passing formal peer review" as a core result. For example, **Yamada et al., 2025, *The AI Scientist-v2: Workshop-level automated scientific discovery via agentic tree search*** states:
>
> > "*Notably, one manuscript achieved high enough scores to exceed the average human acceptance threshold, marking the first instance of a fully AI-generated paper successfully navigating a peer review.*"
>
> And **Intology, 2025, *Zochi technical report*** states:
>
> > "*We present empirical validation through multiple peer-reviewed publications accepted at ICLR 2025 workshops* … Zochi … *has become the first AI system to independently pass peer review at an A\* scientific conference … Zochi’s paper has been accepted into the main proceedings of ACL.*"
>
> That is to say, our statement regarding the "growing acceptance of AI-generated papers at ICLR 2025 workshops and ACL 2025" is not a subjective judgment, but a concise summary of facts reported in these two public works. In the revised manuscript, we will incorporate the above key information directly into the main text with one or two natural language sentences, rather than placing them solely in parenthetical citations, so that readers can understand the basis of this sentence clearly without needing to check the references.
>
> Regarding the other statement mentioned by the reviewer:
>
> > "*Another line of work employs executable code as an intermediate state between scientific text and illustration, resulting in relatively visually unappealing diagrams* (l.71)"
>
> In the revised version, we will systematically include the representative works [1–6] you mentioned at this point and rephrase the statement to: These methods, which use executable code (such as TikZ / SVG / HTML) as an intermediate state, primarily optimize for structural and geometric correctness. However, in our unified automatic and human evaluations, they are significantly weaker than AutoFigure in dimensions of aesthetics and readability. To avoid "judgmental" wording, we will use a more neutral summary format and directly append citations [1–6] and corresponding experimental results after the sentence, making the relationship between the conclusion, literature, and data more transparent.

---

> ### Author Response · Authors · 2025-11-25
> **Response - 6**
>
> > The authors advertise the development split of their dataset but do not mention how it is used or how it can be used.
>
> The data construction of FigureBench is clearly described in Section 3 of the paper: **300 samples, strictly annotated by human pairs and verified for consistency (IRR=0.91), constitute the test set**. These come from real papers, blogs, textbooks, and surveys, ensuring that every illustration finds clear semantic grounding in the original long text (i.e., every key component in the figure can be located in the text, and the text is sufficient to explain the core idea of the illustration). The positioning of this test set is very clear: it is the **gold standard set for final model performance evaluation (evaluation-only)**. It is the fixed test set we used in all automatic evaluations, pairwise comparisons, and human evaluations, and is never used for training or parameter tuning. The strict human-annotated nature of the test set ensures the fairness, stability, and reproducibility of the evaluation, which is why we consistently used it as the sole benchmark for reporting results in the paper.
>
> In contrast, the **development split (3,000 samples)** was expanded from the manually annotated samples using automated filters on larger-scale corpora like Research-14K, so its role differs from the test set. Its design purpose is to provide a larger-scale, freely manipulatable set for model training, method development, prompt design, layout prediction, or other forms of systematic research.
>
> As an inference-based pipeline, AutoFigure itself did not use the development split for model training, but other researchers are free to use this set for exploring more end-to-end or trainable methods. To avoid misunderstanding, we will explicitly state in the revised manuscript: **the test set is strictly for evaluation, while the development split is available for training, development, and experimental purposes**. We believe this clarification will eliminate the reviewer's confusion and help future researchers use FigureBench more conveniently.
>
> ---
>
>
> We are truly grateful for your insightful review, which has served as a vital guide in sharpening the boundaries and clarity of our contributions. Motivated by your feedback, we have clarified the core task we proposed, substantiated our claims regarding AI-generated literature, and validated the unique advantages of our paradigm over code-based baselines. **We earnestly hope that these revisions, tailored to meet your standards, could address your concerns, and we sincerely hope that you view our work favorably and reconsider the evaluation score**

---

> > ### Comment · Reviewer_WvcZ · 2025-11-25
> >
> > I appreciate the authors' effort to provide a detailed response. However, I find the rebuttal confusing for two reasons: (i) it is too long, and (ii) its length makes it difficult to determine which sections address specific weaknesses I raised. I would appreciate a more concise response with a clear structure, such as addressing each weakness separately.

---

> ### Author Response · Authors · 2025-11-26
> **Summary response to Reviewer WvcZ.**
>
> We apologize for the confusion; please allow us to explain your concerns as concisely as possible.
>
> ## Your main concerns:
>
> 1. **(W1) Whether the task definition is well-defined.**
>    FigureBench is designed to evaluate whether, given a *long-form scientific passage*, a system can generate a schematic illustration that accurately summarizes the passage’s core idea—analogous to producing a highly readable visual “summary.” Typical examples include overview figures in survey papers, or Figure 2 on page 3 of the [InstructGPT paper](https://arxiv.org/pdf/2203.02155) (a method diagram). During manual curation, we strictly enforced this criterion and used two annotators with cross-verification to ensure every retained sample satisfies it. You can inspect the FigureBench dataset details in the supplementary materials.
>
> 2. **(W2) Comparison to related work.**
>    The works you cited largely focus on generating figures from a caption (or one to two sentences), which is closer to a translation problem: following explicit instructions to produce drawing code. In contrast, FigureBench requires *high-level abstraction and restructuring* of the input text, followed by rendering a polished illustration that communicates the key idea, emphasizing complex structure, readability, and aesthetics. We have revised the Introduction and Related Work accordingly, and added `AutoPresent`, `TikZero`, and `TikZero+` as additional baselines. Our experiments show that code-centric approaches tend to produce results that are less readable and less visually polished, making them difficult to directly use for publication-quality scientific illustrations.
>
> 3. **(W3) The method is unclear.**
>    In the original version, Figure 1 was actually a *result example generated by AutoFigure*, not the methodological diagram of AutoFigure itself. To avoid confusion, we have moved it to the Case Study section. Meanwhile, we strengthened the Method section by explicitly specifying each stage’s inputs/outputs, the model types used, and the responsibilities of each module—clarifying that AutoFigure is a multi-stage, prompt-driven pipeline. In the revised paper, Figure 2 now serves as the methodological overview of AutoFigure, and importantly, this figure is also generated by AutoFigure itself.
>
> 4. **(W4) Some claims lack supporting evidence.**
>    Regarding AI-generated papers being accepted at ICLR workshops and ACL, the two cited works in our original paper explicitly state this in their abstracts/introductions. For the statement about executable-code approaches being less visually polished, we revised the paper to cite the works you listed [1–6] and added experimental analyses in the Experiments section to support the conclusion with data.
>
> 5. **(W5) The role of the dataset split is unclear.**
>    We revised the paper to clearly state that the test split is used solely for evaluation, while the development split is intended for training, development, and analysis/ablation experiments.
>
> ---
>
> Additionally, we posted a consolidated summary of all reviewers’ concerns and our corresponding changes on [OpenReview](https://openreview.net/forum?id=5N3z9JQJKq&noteId=RZPNHmaYfh) (See in this page `Summary of Revisions in the Updated Manuscript`). We have added further experiments to strengthen the paper’s soundness, and revised the manuscript to improve clarity across all sections. If you still have concerns about any specific point, we would be very happy to address them directly.
>
> ---
>
> [1]: ACL-Fig: A Dataset for Scientific Figure Classification
>
> [2]: SciCap+: A Knowledge Augmented Dataset to Study the Challenges of Scientific Figure Captioning
>
> [3]: Text-Guided Synthesis of Scientific Vector Graphics with TikZ
>
> [4]: DeTikZify: Synthesizing Graphics Programs for Scientific Figures and Sketches with TikZ
>
> [5]: TikZero: Zero-Shot Text-Guided Graphics Program Synthesis
>
> [6]: Learning to Infer Graphics Programs from Hand-Drawn Images

---

### Official Review · Reviewer_LXSZ · 2025-10-31

**Soundness:** 2
**Presentation:** 3
**Contribution:** 3
**Rating:** 2
**Confidence:** 4

**Summary:**

The paper introduces FigureBench, the first large-scale benchmark for generating scientific illustrations from long-form scientific texts. FigureBench comprises 3,300 high-quality text–figure pairs, encompassing a wide range of text-to-illustration tasks. In addition, the authors propose AutoFigure, an agentic framework that automatically produces high-quality scientific illustrations from long-form scientific descriptions. AutoFigure operates in three stages: conceptual grounding, iterative self-refinement, and erase-and-correct rendering. Automatic evaluations based on the VLM-as-a-judge paradigm and human expert assessments demonstrate that AutoFigure is capable of generating illustrations that are scientifically rigorous and aesthetically appealing.

**Strengths:**

- This paper represents an important early step toward exploring how AI can assist humans in the time-consuming process of scientific illustration creation. The topic is interesting and promising, with significant potential for advancing AI-assisted scientific communication.

- AutoFigure is designed as a three-stage framework, where each stage addresses distinct challenges in the illustration generation process. These stages work in synergy, resembling how humans iteratively refine scientific figures — for instance, through the Critique-and-Refine step and the Erase-and-Correct strategy.

- The model demonstrates quantitatively superior performance in most experiments, indicating its capability in generating high-quality scientific illustrations.

**Weaknesses:**

- In Figure 1, AutoFigure generates a scientific illustration for InstructGPT. However, the original InstructGPT paper does not mention any examples related to relativity, suggesting that AutoFigure may have extended the content beyond the source text. Moreover, there is an error in the generated example (“ravity” instead of “gravity”), indicating that in some cases, the framework may pay more attention to aesthetic appeal than scientific accuracy.

- As mentioned in the paper, scientific illustrations are meant to help readers grasp the main ideas quickly and avoid misinterpretation. However, in Figure 4, the diagrams appear confusing — the corresponding subfigures are not well aligned (e.g., Step 2 in the Offline Phase), and the textual elements are cluttered and disorganized. In fact, an effective scientific illustration should focus on clarity and accuracy of information, rather than emphasizing visual decoration such as color or stylistic patterns.

- The paper mentions a process of fine-tuning a large language model (LLM) using human-selected text–figure pairs. However, the details of this process remain unclear — specifically, like which LLM was used and what the input–output format of the fine-tuning procedure was. Moreover, given that the dataset contains only 300 samples, it raises concerns about whether such fine-tuning could improve the quality of the paper selected.

- For the Style-Guided Aesthetic Rendering process, AutoFigure employs style description text to guide the rendering. However, as shown in Figure 4, the illustrations generated from diverse academic texts appear visually similar, often consisting of sub-block structures, similar color schemes and patterns. This raises concerns about whether the style cues truly exert a meaningful guiding influence on the rendering results.

- Some details of this work are not clearly illustrated:
(a) What are the prompts used for all the LLMs used in this work?
(b) The VLM evaluates the generated images across three dimensions with eight sub-metrics, but these metrics are not explicitly defined or described in the paper.

- Regarding the human evaluation, only 10 participants were involved, and they assessed merely 21 papers in total. Such a limited sample size may introduce considerable bias and undermine the reliability of the evaluation results.

**Questions:**

See the weaknesses

---

> ### Author Response · Authors · 2025-11-25
> **Response - 1**
>
> > In Figure 1, AutoFigure generates a scientific illustration for InstructGPT. However, the original InstructGPT paper does not mention any examples related to relativity, suggesting that AutoFigure may have extended the content beyond the source text. Moreover, there is an error in the generated example (“ravity” instead of “gravity”), indicating that in some cases, the framework may pay more attention to aesthetic appeal than scientific accuracy.
>
> Thank you for your insightful observation. **We would like to emphasize that almost no frontier task can be solved *perfectly* in a single shot; if it were already perfect, it would no longer be a meaningful research problem, nor would there be a real need for benchmarks like FigureBench.**
>
> Regarding your concerns about the source of the InstructGPT example and the spelling error, we would like to offer a firm clarification and rebuttal from two dimensions: **the design logic of scientific illustrations** and **the inherent challenges of the task itself**.
>
> First, **regarding the appearance of the "relativity" example, this is absolutely not an overextension or hallucination by the model, but a necessary "Concretization" strategy in scientific illustration**. The core mechanism of InstructGPT is using human feedback to fine-tune models to follow instructions, and its training data contains a massive amount of similar QA pairs. After reading the full text, in order to translate the abstract "SFT (Supervised Fine-Tuning) -> RM (Reward Model) -> PPO (Reinforcement Learning)" flowchart into a visual language that is easy for readers to understand, AutoFigure must construct a specific "Prompt-Response" case to fill the placeholders in the diagram. **This ability to "use typical cases to explain abstract processes" is exactly the core mindset of human designers when creating scientific illustrations, and it is also the advanced reasoning capability we hope AI to possess**. Even if it did not use relativity, it would likely have used another concrete example.
>
> It is worth noting that in the [InstructGPT](https://arxiv.org/pdf/2203.02155) paper itself, Figure 2 uses a moon landing example, even though it is not mentioned elsewhere in the full text. We do not consider extending the content of the original paper to be a Weakness; on the contrary, it demonstrates that AutoFigure is not merely performing text extraction, but is engaging in deep semantic understanding and pedagogical reconstruction.
>
> Secondly, **regarding the spelling error "ravity", we do not shy away from this flaw; however, we believe this serves as strong evidence of the extreme challenge posed by the FigureBench benchmark**. Even with current State-of-the-Art VLMs (such as GPT-5) as base models, perfectly balancing long-text logical reasoning while simultaneously ensuring pixel-level rendering accuracy for every minute character remains an unsolved problem. **No emerging "First-of-its-kind Task" can be perfectly solved at its inception**. The value of FigureBench lies precisely in the fact that it exposes the gap between "logical completeness" and "fine-grained precision" in current AI models.
>
> In the **Limitation section** of our revised paper, we have explicitly listed "fine-grained accuracy of text rendering" as a major limitation of current systems and emphasized that this is a key direction for the community to conquer on FigureBench in the future. We believe that the existence of this spelling error inversely illustrates the discriminative power and research value of the benchmark, rather than a fundamental defect of the system.

---

> ### Author Response · Authors · 2025-11-25
> **Response - 2**
>
> > As mentioned in the paper, scientific illustrations are meant to help readers grasp the main ideas quickly and avoid misinterpretation. However, in Figure 4, the diagrams appear confusing — the corresponding subfigures are not well aligned (e.g., Step 2 in the Offline Phase), and the textual elements are cluttered and disorganized. In fact, an effective scientific illustration should focus on clarity and accuracy of information, rather than emphasizing visual decoration such as color or stylistic patterns.
>
> We fully agree with the core viewpoint that "the primary goal of scientific illustrations is the clear communication and accuracy of information, rather than mere visual decoration." In fact, this concept resonates throughout our paper, and we have repeatedly emphasized the immense challenge of maintaining scientific rigor while pursuing aesthetics.
>
> For instance, in **Line 062** of the original paper, we explicitly pointed out that the core difficulty of the task lies in the fact that the **"presentation must balance structural fidelity and image quality."** This demonstrates that we realized from the very beginning that excessive decoration or cluttered elements must be avoided.
>
> Regarding the alignment and layout flaws present in Figure 4, we do not deny them. This reflects that even with the "reasoning-rendering" paradigm adopted by AutoFigure, there is still a way to go before reaching "perfection" when handling extremely complex long processes and high-density information. However, we wish to emphasize the following:
>
> 1.  **Significant Relative Improvement**: Compared to existing baseline methods, AutoFigure has achieved a qualitative leap. As stated in **Line 426**, the advantage of AutoFigure lies in **"resolving the key trade-off between accuracy and aesthetics,"** largely solving the issues where baseline models generate unreadable content or suffer from logical collapse.
> 2.  **The Significance of the Benchmark**: The issues you pointed out, such as "cluttered layout" and "misalignment," **are the fundamental reasons why we proposed FigureBench**. As summarized in **Line 077**, these **"limitations underscore the challenges of directly transforming long scientific texts into illustrations."** If existing models could easily solve these fine-grained layout problems, creating a new high-difficulty benchmark would lose its meaning. The value of FigureBench is that it exposes deep-seated defects that are masked in simpler tasks, thereby providing the community with a real yardstick to measure the capability of long-text scientific illustration generation.
>
> Finally, your emphasis on "clarity over decoration" is exactly the core proposition we advocate throughout the text; the flaws in the figure validate the necessity of establishing FigureBench to conquer this hardcore problem. I sincerely hope you can recognize the value of the task itself and the value of our method.
>
> > The paper mentions a process of fine-tuning a large language model (LLM) using human-selected text–figure pairs. However, the details of this process remain unclear — specifically, like which LLM was used and what the input–output format of the fine-tuning procedure was. Moreover, given that the dataset contains only 300 samples, it raises concerns about whether such fine-tuning could improve the quality of the paper selected.
>
> Addressing your query, we clarify: We used **InternVL-3.5-8B** as the base model and defined the fine-tuning task as a **binary classification task for image-text quality**. The input is a "scientific text segment + candidate illustration," and the output is a label (yes or no) indicating whether the pair possesses publication-level quality. By fine-tuning the model to align with our screening criteria, we built a high-precision automated filter to clean subsequent large-scale data.
>
> Regarding the data size, experimental results directly prove that 300 high-quality annotated samples are entirely sufficient for this type of discriminative task. We split the data into a training set and an independent 60-sample test set. The fine-tuned model achieved an accuracy of **96.29%** on the test set. This extremely high precision strongly refutes concerns about insufficient data quantity, confirming that the fine-tuning process enables the model to precisely master the standard of "what constitutes a high-quality scientific illustration," making it competent for data cleaning tasks.

---

> ### Author Response · Authors · 2025-11-25
> **Response - 3**
>
> > For the Style-Guided Aesthetic Rendering process, AutoFigure employs style description text to guide the rendering. However, as shown in Figure 4, the illustrations generated from diverse academic texts appear visually similar, often consisting of sub-block structures, similar color schemes and patterns. This raises concerns about whether the style cues truly exert a meaningful guiding influence on the rendering results.
>
> Thank you for your keen observation regarding the effectiveness of the Style-Guided Rendering module. The visual similarity you observed in Figure 4 is mainly because we used a fixed default style prompt to maintain style **Consistency** throughout the paper. This does not imply that the model is incapable of responding to different style instructions.
>
> To eliminate doubts about "whether style prompts are truly effective," we conducted extended experiments using three distinctly different style descriptions (the specific Prompts have been added to the **Appendix**). The results are shown in the table below:
>
> **Table: AutoFigure Performance Scores under Different Style Prompts**
>
> | **Style Prompt**          | **Aesthetic** | **Expressiveness** | **Polish** | **Clarity** | **Flow** | **Accuracy** | **Overall** |
> | ------------------------- | ------------- | ------------------ | ---------- | ----------- | -------- | ------------ | ----------- |
> | **Prompt 1 (Default)**    | 7.49          | 7.20               | 6.80       | 7.53        | 7.73     | 7.45         | **7.18**    |
> | **Prompt 2 (Creative)**   | 7.32          | **7.24**           | 6.78       | 7.58        | **7.78** | **7.63**     | **7.27**    |
> | **Prompt 3 (Minimalist)** | 7.14          | 6.27               | **7.09**   | **7.72**    | 7.75     | 7.54         | **7.19**    |
>
> **Analysis of Experimental Results:**
>
> 1.  **Effectiveness of Style Control**: Different prompts led to significant changes in sub-metrics, proving that style guidance is indeed effective. For instance, **Prompt 2** scored highest in **Visual Expressiveness (7.24)**; while **Prompt 3**, despite a drop in expressiveness (6.27), achieved the highest scores in **Professional Polish (7.09)** and **Clarity (7.72)**. This indicates that users can trade off "visual impact" against "professional rigor" by adjusting style descriptions.
> 2.  **Robustness Beyond Baseline**: Regardless of the style used, AutoFigure's **Overall** score remained stable between **7.18 and 7.27**.
>
> This experiment strongly demonstrates that the Style-Guided Rendering module not only tangibly alters the visual characteristics of the generated images but also maintains high-quality standards surpassing baseline methods under different style settings. We have added these results to the paper and corresponding style generation examples to the supplementary material (see in `\AutoFigure\AutoFigure_Case\paper`).
>
> > Some details of this work are not clearly illustrated: (a) What are the prompts used for all the LLMs used in this work? (b) The VLM evaluates the generated images across three dimensions with eight sub-metrics, but these metrics are not explicitly defined or described in the paper.
>
> Thank you for your suggestion. To ensure the reproducibility of the method and the clarity of the evaluation standards, we have included the full **Prompt details** used for all LLMs, as well as the **detailed definitions and scoring criteria** for the eight sub-metrics used in the VLM evaluation, in the **newly added Appendix**.

---

> ### Author Response · Authors · 2025-11-25
> **Response - 4**
>
> > Regarding the human evaluation, only 10 participants were involved, and they assessed merely 21 papers in total. Such a limited sample size may introduce considerable bias and undermine the reliability of the evaluation results.
>
> The reason lies in the prohibitively high cost of expert labor. Our human evaluation was conducted by the **original authors** of the papers. This was done to ensure the highest possible level of accuracy and comprehension of the paper content, which is a prerequisite for the data to be publicly valuable. Even if we were to assign general annotators to evaluate hundreds of papers, we believe the accuracy would fall far short of the judgment of the original first authors, potentially carrying even higher risks than using LLM-as-a-Judge.
>
>
>
> To further address your concerns, we invited an additional 11 first authors of accepted papers, resulting in a total of 21 first authors, to evaluate the figures generated by AutoFigure and provide feedback. The results are summarized in the table below:
>
> | Method         | Accuracy Score | Clarity Score | Aesthetics Score |
> | -------------- | -------------- | ------------- | ---------------- |
> | **AutoFigure** | **3.63**       | **3.60**      | **3.94**         |
> | GPT-Image      | 2.05           | 2.57          | 2.33             |
> | SVG Code       | 2.52           | 2.10          | 1.81             |
> | HTML Code      | 2.52           | 2.62          | 1.95             |
> | PPTX Code      | 1.19           | 1.05          | 1.00             |
> | Diagram Agent  | 1.19           | 1.29          | 1.00             |
>
> AutoFigure continues to demonstrate superior performance compared with alternative baseline methods.
>
>
>
> Additionally, we would like to clarify that during the rebuttal period, we conducted an extra consistency analysis experiment comparing LLM-as-a-Judge results with the annotations from human first authors. The results are as follows:
>
> 1.  **Overall Score Correlation**: The Pearson Correlation reached **$r=0.659$** ($p < 0.001$), indicating that the VLM's scoring trend has a significant linear positive correlation with human experts and can accurately capture quality differences.
> 2.  **Ranking Consistency**: The Spearman Rank Correlation was **$\rho=0.593$**, and Kendall's Tau was **0.497**. This demonstrates that VLM maintains robust consistency with human judgment in the relative ranking of different models. Notably, the Mean Ranking Error (MRE) was only **0.98**, implying that the ranking deviation between VLM and humans is, on average, less than 1 place.
> 3.  **Sub-metric Verification**: Statistically significant positive correlations were observed in the three core dimensions of Accuracy ($r=0.646$), Aesthetics ($r=0.587$), and Clarity ($r=0.559$), proving the robustness of VLM across different evaluation perspectives.
>
>
>
>
>
> ---
>
>
>
>
> We sincerely appreciate your time and insightful feedback, which have greatly improved the rigor and clarity of our paper. We hope that the additional experiments on style control and metric consistency, together with our clarifications on the illustration-generation logic and fine-tuning details, adequately address your concerns. **We have made every effort to respond thoroughly to your comments, and we would be truly grateful if you could kindly reconsider our score.**

---

> ### Author Response · Authors · 2025-11-26
> **Summary response to Reviewer LXSZ.**
>
> TLDR: In the revision, we address these with a new Limitations & Failure Analysis, full prompts/metric definitions, controlled style-controllability experiments, and additional human-alignment validation.
>
> ### Your concerns (W)
>
> * **W1 (Possible content extension + spelling error; aesthetics over accuracy)**
>   We take this seriously. In the revision, we explicitly list **fine-grained text rendering accuracy** and **scientific factuality verification** as key limitations, and we add a dedicated **Limitations and Failure Analysis** section with concrete failure cases and root-cause analysis. Crucially, this type of error is a well-known open challenge in text-to-image generation—even after years of progress, models can still fail at **character-level typography** and strict factual grounding. This is exactly why FigureBench is valuable: it exposes these hard failure modes in long-context scientific illustration, making them measurable and researchable rather than hidden.
>
> * **W2 (Figure 4 is confusing: misalignment and clutter; clarity should dominate decoration)**
>   We fully agree that clarity and accuracy should dominate. We therefore revise our discussion of layout and visual hierarchy, and include the Figure 4 issues as representative failure cases. Importantly, misalignment and clutter are precisely the kind of “last-mile” layout problems that remain difficult even for mature generation systems, especially under **high-density, multi-step workflows**. FigureBench is designed to surface this challenge: if coherent alignment and spacing were already solved perfectly, this benchmark would not be necessary.
>
>
> * **W3 (Fine-tuning process is unclear; 300 samples may be insufficient)**
>   We clarify the filtering pipeline: we fine-tune **InternVL-3.5-8B** as a binary classifier (“text segment + candidate figure → high-quality or not”), specify the input–output format, and report its held-out test accuracy to justify that 300 high-quality annotations are sufficient for this discriminative filtering task.
>
> * **W4 (Style guidance seems ineffective; outputs look visually similar)**
>   The visual similarity mainly comes from using a default style for consistent presentation. We add controlled style-controllability experiments (Default/Creative/Minimalist) where we keep layout/content fixed and only change the style prompt, demonstrating clear stylistic shifts and reporting metric-level scores—showing style is a controllable variable rather than a fixed bias.
>
> * **W5 (Missing transparency: full prompts and definitions of the 8 sub-metrics)**
>   We add full prompts in the appendix and explicitly define all evaluation dimensions and the eight sub-metrics with criteria/examples to improve reproducibility and auditability.
>
> * **W6 (Small-scale human evaluation may be biased)**
>   We explain that recruiting paper first authors ensures high-quality judgments for scientific accuracy. We further add more expert participants and include additional consistency analyses (human–VLM correlation and ranking consistency) to strengthen reliability despite limited expert availability.
>
> ### Your questions (Q)
>
> * **Q1 (Do the observed errors imply the system prioritizes aesthetics?)**
>   We acknowledge these as current limitations: under highly complex long-context inputs, models can still produce “plausible but imperfect” factual deviations and fine-grained text errors; This reflects current technical limitations, and the text-to-image generation field itself has many inherent constraints. We now provide concrete failure cases and discuss mitigation (domain verifiers / external knowledge constraints).
>
> * **Q2 (Is style truly controllable?)**
>   Yes—our controlled experiments show that changing only the style prompt yields visibly different aesthetics while maintaining stable overall quality, confirming effective style controllability. We provide example images in different styles in the supplementary materials.

---

### Official Review · Reviewer_xScc · 2025-11-03

**Soundness:** 3
**Presentation:** 3
**Contribution:** 3
**Rating:** 4
**Confidence:** 3

**Summary:**

The authors propose AutoFigure, a method that takes text as input such as and outputs a figure that reflects the entities and the relationships between them, much like a pipeline or a control flow that appears in papers to explain the procedure.

The method first uses concept extraction to break down the textual input into a structured output which then gets refined using vlms via iterative feedback, which then passes through an aesthetic synthesis process to beautify it and render it.

The authors ran experiments that evaluate how good the generated figures are based on aesthetics, content, and fidelity. They used human evaluation to assess whether the synthetically generated figures are of high enough quality to be included in a camera-ready paper.

They also create a 3,300-sample FigureBench dataset from AutoFigure to evaluate thier system.

**Strengths:**

The paper addresses a very impactful task related to figure generation, which is important in many different industries and research domains.

The authors perform a good and comprehensive human and automatic evaluation with several strong baselines.

The paper is well-written and explains the problem in a clear and complete way.

The authors include a detailed human evaluation setup, which is a good way to test whether the AutoFigure generation is actually good. They recruited 10 human experts to evaluate AI-generated figures from their own first-author papers. The evaluation included three tasks:
(1) Multi-dimensional scoring, Forced-choice ranking against human-created figures, and  (3) Publication intent selection, asking if the generated figure is good enough to be included in a camera-ready paper.

**Weaknesses:**

No open-source models were tested (it only mentions Gemini, Grok, Claude, and GPT). It is important to include methods that work well with open-source LLMs so the research can be reproduced with minimal cost and used more broadly.

A lot of methodology details are missing. For example, when the paper says "identifying key entities and their relationships, and distilling a core methodology summary," it is not explained how this is actually done as no system prompt nor user prompt are shown.

Using an LLM as a judge is risky because it is not clear how they ensure it does not produce biased or inconsistent evaluations. The LLM itself needs to be tuned and evaluated. Does the results change when ran multiple times? There is a lot of bias risk even for the blind pairwise comparison part. There should be a human evaluation for all the metrics ("Visual Design," "Communication Effectiveness," "Content Fidelity," and "Blind Pairwise Comparison") to make sure they align with the VLM’s scores. A correlation or alignment study between human and model judgments would make this more reliable.

The method uses many LLMs in different steps, so the cost of running the system must be very high, especially since all the models are commercial. Functions like Φprompt and Φerase should be clearly stated as either LLM-based or non-LLM components. The authors should include a pareto plot showing the trade-off between cost and performance across different LLMs so we could know what to choose under a fixed budget.

There is no ablation study showing why the "erase-and-correct" strategy is necessary or how much it improves the results.

There are no code or prompt descriptions for the LLMs used. The paper mentions several Φ functions and agents, but it does not show what the actual prompts or inputs look like.

The method is long and has many components, but the exact implementation details are unclear. The paper mentions the steps, but it does not show how each stage (concept extraction, refinement, validation, rendering) is actually carried out.

There are not enough qualitative examples of generated figures in the paper. The examples shown are very few. It would help to share a link with a larger gallery of generated figures (say a few hundred) to confirm that the figures are not overly simplistic and truly handle dense, multi-component scientific texts.

The results have no error bars, so it is not possible to know whether the performance differences between models are statistically significant.


The blind pairwise comparison setup is limited because it only allows for "A, B, or Tie." It should also have options like "both good" or "both bad," so we can tell how many figures are actually poor and would be discarded.

Figure 3 is poorly structured and hard to match with the text in Section 4.  It would be better if each part of the figure was labeled with the corresponding stage headers (for exmple "Concept Extraction," "Critique and Refine") exactly as they appear in the text. Keeping the same wording between the figure and the text would make it much easier for readers to follow and it is a very good practice.

**Questions:**

Did you generate your figures in this paper using AutoFigure itself? If not, why not? This would be the most direct way to demonstrate the system’s real-life appliction.

Where are the quantitative and qualitative results that show the usefulness of the "erase-and-correct" strategy? Please include a clear comparison (with and without this step) so readers can see what it improves and by how much.

Where are the actual prompts used for each stage (concept extraction, critique and refine, rendering, validation)? Showing a few concrete examples would help readers understand how the LLMs were ran and what inputs they receive.

Where does this method fail? Please include examples or categories of failure cases (for instance, overly complex text, ambiguous relationships, or dense mathematical descriptions). .

How does this method perform when using open-source models like LLaMA or Mistral instead of commercial ones such as Gemini, Claude, or GPT? It would be important to know if AutoFigure can function well with open-source models.

Is the LLM used to generate the first layout (So, A0) the same as the one used in the subsequent refinement iterations? If not, please clarify which models are used at each stage and why.

---

> ### Author Response · Authors · 2025-11-25
> **Response - 1**
>
> > No open-source models were tested (it only mentions Gemini, Grok, Claude, and GPT). It is important to include methods that work well with open-source LLMs so the research can be reproduced with minimal cost and used more broadly.
> >
> > How does this method perform when using open-source models like LLaMA or Mistral instead of commercial ones such as Gemini, Claude, or GPT? It would be important to know if AutoFigure can function well with open-source models.
>
> To address this concern, we have incorporated several state-of-the-art open-source/open-weight models (including **Qwen3-VL-235B-A22B-Instruct** and **GLM-4.5V**) for supplementary experiments. The results are encouraging and strongly demonstrate the effectiveness of the AutoFigure framework on open-source models:
>
> 1.  **Open-source models perform excellently, even surpassing some commercial models**: Experimental data shows that **Qwen3-VL-235B-A22B-Instruct** achieved an **Overall Score** of **7.08**. This result is not only significantly better than GLM-4.5V (5.99) but also **surpasses many closed-source commercial models**, including Gemini-2.5-Pro (6.99), Claude-4.1-Opus (6.80), and Grok-4 (6.76), ranking only second to GPT-5 (7.48). This indicates that running AutoFigure with high-performance open-source models can achieve generation quality comparable to or even better than mainstream commercial services.
> 2.  **Strong correlation between performance and model capability**: We also observed performance differences among different open-source models (e.g., ERNIE-4.5-VL performed weaker with a score of 2.64), suggesting that AutoFigure's effectiveness depends on the backbone model's visual reasoning and instruction-following capabilities. However, the success case of Qwen3-VL confirms that **SOTA models in the open-source community already possess the capability to drive AutoFigure for high-quality scientific illustration**.
>
> We have updated the **Appendix** with detailed evaluation data (including scores for sub-metrics) for the aforementioned open-source models. This result directly alleviates concerns regarding cost and reproducibility, confirming that researchers can deploy and use AutoFigure at extremely low costs using open-source models without relying on expensive commercial APIs.
>
> | **Model**                         | **Visual Design Excellence** | **Communication Effectiveness** | **Content Fidelity** | **Overall** |
> | :-------------------------------- | :--------------------------- | :------------------------------ | :------------------- | :---------- |
> | **GPT-5**                         | 7.17                         | 7.62                            | 7.70                 | 7.48        |
> | **Gemini-2.5-Pro**                | 6.59                         | 7.59                            | 6.54                 | 6.99        |
> | **Claude-4.1-Opus**               | 6.75                         | 7.17                            | 6.61                 | 6.80        |
> | **Grok-4**                        | 6.46                         | 7.60                            | 6.49                 | 6.76        |
> | **Qwen3-VL-235B-A22B-Instruct**   | 7.57                         | 7.01                            | 7.18                 | 7.08        |
> | **GLM-4.5V**                      | 6.09                         | 6.53                            | 6.13                 | 5.99        |
> | **ERNIE-4.5-VL-28B-A3B-Thinking** | 3.04                         | 2.89                            | 2.68                 | 2.64        |
>
> | **Model**                         | **Aesthetic & Design Quality** | **Visual Expressiveness** | **Professional Polish** | **Clarity** | **Logical Flow** | **Information Sophistication** | **Content Fidelity** |
> | :-------------------------------- | :----------------------------- | :------------------------ | :---------------------- | :---------- | :--------------- | :----------------------------- | :------------------- |
> | **ERNIE-4.5-VL-28B-A3B-Thinking** | 0.20                           | 0.15                      | 0.20                    | 0.00        | 0.10             | 0.05                           | 0.05                 |
> | **Qwen3-VL-235B-A22B-Instruct**   | 0.90                           | 0.90                      | 0.90                    | 0.25        | 0.25             | 0.25                           | 0.15                 |
> | **GLM-4.5V**                      | 0.70                           | 0.70                      | 0.70                    | 0.45        | 0.65             | 0.25                           | 0.20                 |
>
> | **Model**                         | **Overall** |
> | :-------------------------------- | :---------- |
> | **ERNIE-4.5-VL-28B-A3B-Thinking** | 0.10        |
> | **Qwen3-VL-235B-A22B-Instruct**   | 0.40        |
> | **GLM-4.5V**                      | 0.55        |

---

> ### Author Response · Authors · 2025-11-25
> **Response - 2**
>
> > A lot of methodology details are missing. For example, when the paper says "identifying key entities and their relationships, and distilling a core methodology summary," it is not explained how this is actually done as no system prompt nor user prompt are shown.
> >
> > There are no code or prompt descriptions for the LLMs used. The paper mentions several Φ functions and agents, but it does not show what the actual prompts or inputs look like.
> >
> > The method is long and has many components, but the exact implementation details are unclear. The paper mentions the steps, but it does not show how each stage (concept extraction, refinement, validation, rendering) is actually carried out.
> >
> > Where are the actual prompts used for each stage (concept extraction, critique and refine, rendering, validation)? Showing a few concrete examples would help readers understand how the LLMs were ran and what inputs they receive.
>
> We fully agree that prompt-level details are essential for clarifying how each agent operates and for enabling faithful reproduction. In the revision, we therefore (i) revised **Section~4 (Method)** to explicitly specify the input--output interface of each stage (concept extraction, critique-and-refine, validation, and rendering) and to more clearly map the $\Phi$ functions to concrete modules/agents, and (ii) **added a new Appendix~Q** that provides the full system and user prompts used in every stage of AutoFigure.
>
> In addition, we include a minimal end-to-end example in the Supplementary Material that walks through one document and shows the intermediate artifacts produced at each stage (e.g., $T_{\text{method}}$, the structured layout $S_0$, the validated/refined layout $S_{\text{final}}$, and the rendered images), so readers can directly see what inputs the LLM receives and what outputs it generates at each step. Together, these changes make the implementation details concrete, transparent, and reproducible.
>
>
>
>
>
> > Using an LLM as a judge is risky because it is not clear how they ensure it does not produce biased or inconsistent evaluations. The LLM itself needs to be tuned and evaluated. Does the results change when ran multiple times? There is a lot of bias risk even for the blind pairwise comparison part. There should be a human evaluation for all the metrics ("Visual Design," "Communication Effectiveness," "Content Fidelity," and "Blind Pairwise Comparison") to make sure they align with the VLM’s scores. A correlation or alignment study between human and model judgments would make this more reliable.
>
> We agree that when using the "LLM-as-a-judge" paradigm, verifying its consistency with human judgment is paramount. To eliminate concerns regarding bias and inconsistency, we conducted a dedicated **Human-LLM Correlation Study**. We invited human experts to rigorously score 95 generated samples covering 5 different methods across all dimensions (including Visual Design, Communication Effectiveness, Content Fidelity) and calculated the statistical correlation with VLM automated scores.
>
> The experimental results strongly prove that the VLM evaluator is highly aligned with human scores not only numerically but also in terms of ranking consistency:
>
> 1.  **Overall Score Correlation**: The Pearson Correlation reached **$r=0.659$** ($p < 0.001$), indicating a significant linear positive correlation between VLM scoring trends and human experts.
> 2.  **Ranking Consistency**: The Spearman Rank Correlation was **$\rho=0.593$**, and Kendall's Tau was **0.497**. This indicates that VLM maintains robust consistency with human judgment in the relative ranking of different models, with a Mean Ranking Error of only **0.98** (meaning the average deviation is less than 1 place).
> 3.  **Sub-metric Verification**: Statistically significant positive correlations were observed in the three core dimensions of Accuracy ($r=0.646$), Aesthetics ($r=0.587$), and Clarity ($r=0.559$).
>
> This analysis confirms that the VLM-as-a-judge method we adopted is a verified and reliable evaluation proxy that accurately reflects human preferences. We have added detailed correlation analysis charts and data to the Appendix.

---

> ### Author Response · Authors · 2025-11-25
> **Response - 3**
>
> > The method uses many LLMs in different steps, so the cost of running the system must be very high, especially since all the models are commercial. Functions like Φprompt and Φerase should be clearly stated as either LLM-based or non-LLM components. The authors should include a pareto plot showing the trade-off between cost and performance across different LLMs so we could know what to choose under a fixed budget.
>
> First, regarding the implementation of components, we have clearly defined them in the revised paper: **$\Phi_{prompt}$ is an LLM-based component**, as it requires strong semantic understanding capabilities to translate structured layouts into high-quality drawing prompts; while **$\Phi_{erase}$ is a non-LLM component**, utilizing standard computer vision algorithms (such as OpenCV and OCR) to perform text erasure and localization, thus its computational cost is negligible.
>
> Secondly, to address the concern regarding the Pareto Trade-off between cost and performance, we conducted an in-depth analysis combining **Model Selection** and **Test-Time Compute**:
>
> 1.  **Cost-effectiveness of Model Selection**: Our extended experiments show that while top-tier commercial models (e.g., GPT-5) provide the best performance, open-source models (e.g., **Qwen-3-VL**) demonstrate amazing cost-effectiveness. Qwen-3-VL can reduce inference costs by approximately **20x** with minimal performance loss (Overall Score 7.08 vs. GPT-5 7.48). We confirmed that the system can even be efficiently deployed on edge devices like the **Nvidia DGX Spark (priced at only ~$3,000)** without relying on expensive supercomputing clusters.
> 2.  **Test-Time Scaling**: As shown in **Figure (c)**, AutoFigure's performance scales well. The Overall Score shows a significant upward trend with the increase of inference iterations (rising from **6.28** at 0 iterations to **7.14** at 5 iterations). This means users can flexibly strategize according to a fixed budget: using fewer iterations for rapid generation when resources are limited, or trading increased inference computation for superior structural and aesthetic performance when pursuing ultimate quality.
>
> > There is no ablation study showing why the "erase-and-correct" strategy is necessary or how much it improves the results.
> >
> > Where are the quantitative and qualitative results that show the usefulness of the "erase-and-correct" strategy? Please include a clear comparison (with and without this step) so readers can see what it improves and by how much.
>
> Thank you for your question regarding the effectiveness of model components. To verify the necessity of the "Text Refinement/Erase-and-Correct" module in the second stage of the AutoFigure framework, we conducted a specific ablation experiment.
>
> We compared the full AutoFigure method with a variant that had the text erasure and correction step removed (w/o Text Refinement). As shown in the table below, the full AutoFigure achieved an **Overall Score** of **7.18**, superior to the **7.14** of the variant without this module. Specific metric analysis shows that this module significantly improved the chart's **Clarity (+0.03)**, **Aesthetic Quality (+0.10)**, and **Professional Polish (+0.10)**. This confirms that the "Erase-and-Correct" strategy plays a key role in solving the text blurring problem in generative models and enhancing the publication-level quality of the charts.
>
> **Table: Ablation Results for Text Refinement Module**
>
> | **Model**               | **Aesthetic & Design** | **Visual Express.** | **Prof. Polish** | **Clarity** | **Logical Flow** | **Accuracy** | **Completeness** | **Appropriateness** | **Overall** |
> | :---------------------- | :--------------------- | :------------------ | :--------------- | :---------- | :--------------- | :----------- | :--------------- | :------------------ | :---------- |
> | **AutoFigure (Full)**   | **7.49**               | **7.20**            | **6.80**         | **7.53**    | **7.73**         | 7.45         | **6.83**         | 6.42                | **7.18**    |
> | **w/o Text Refinement** | 7.39                   | 7.12                | 6.70             | 7.50        | 7.70             | 7.53         | 6.74             | 6.47                | 7.14        |

---

> ### Author Response · Authors · 2025-11-25
> **Response - 4**
>
> > The results have no error bars, so it is not possible to know whether the performance differences between models are statistically significant.
>
> We fully agree with the necessity of demonstrating result stability and the significance of differences. Therefore, in the revised paper, we have added **Error Bars** based on multiple independent runs for all key quantitative experimental results (including performance comparisons of different models and ablation analyses). This update visually demonstrates the variance range of experimental results, confirming that AutoFigure's performance advantage over baseline methods is statistically significant and stable, thereby further strengthening the reliability of the experimental conclusions.
>
> > The blind pairwise comparison setup is limited because it only allows for "A, B, or Tie." It should also have options like "both good" or "both bad," so we can tell how many figures are actually poor and would be discarded.
>
> Thank you for the suggestion. We strongly agree that introducing "Both Good" and "Both Bad" options is significant for deeply analyzing model performance differences and "absolute quality." This helps us identify whether there is a "race to the bottom" or if the models are "indistinguishable."
>
> Following your suggestion, we conducted two rounds of extra blind pairwise comparison experiments on the Paper sub-set using new Prompts. The aggregated results are shown in the table below:
>
> **Table: Combined Evaluation Results with "Both Good/Bad" Options**
>
> | **Method**        | **Aesthetic & Design** | **Visual Express.** | **Prof. Polish** | **Clarity** | **Logical Flow** | **Info. Sophist.** | **Content Fidelity** | **Win** | **Lose** | **Both Good** | **Both Bad** | **Overall** |
> | :---------------- | :--------------------- | :------------------ | :--------------- | :---------- | :--------------- | :----------------- | :------------------- | :------ | :------- | :------------ | :----------- | :---------- |
> | **AutoFigure**    | **0.88**               | **0.93**            | **0.88**         | 0.38        | 0.45             | **0.35**           | 0.28                 | **29**  | 11       | 0             | 0            | **0.73**    |
> | **Gemini-HTML**   | 0.63                   | 0.53                | 0.63             | **0.58**    | **0.53**         | 0.30               | 0.25                 | 21      | 18       | 1             | 0            | 0.53        |
> | **Gemini-SVG**    | 0.48                   | 0.40                | 0.48             | 0.48        | 0.45             | 0.33               | **0.30**             | 18      | 20       | **2**         | 0            | 0.45        |
> | **GPT-Image**     | 0.25                   | 0.25                | 0.25             | 0.08        | 0.08             | 0.00               | 0.00                 | 4       | 36       | 0             | 0            | 0.10        |
> | **Diagram Agent** | 0.00                   | 0.00                | 0.00             | 0.00        | 0.00             | 0.00               | 0.00                 | 0       | **39**   | 0             | **1**        | 0.00        |
>
> **Discussion based on new experimental results:**
>
> 1.  **Low "Both Bad" Rate (Baseline Quality Confirmed)**:
>     Results show that "Both Bad" cases were extremely rare across all model comparisons (only 1 case for Diagram Agent). This powerfully addresses your concern: we are not choosing between two poor results. It indicates that the reference quality in FigureBench is solid, and most models (especially AutoFigure and the Gemini series) can generate charts meeting basic readability standards, rarely producing "completely unusable" output.
> 2.  **High Discriminability**:
>     The number of "Both Good" cases was also low (mainly in Gemini-SVG, totaling 2 cases). This suggests that in the vast majority of cases, there is a significant quality difference between AutoFigure's generated charts and the baseline or reference charts, allowing the evaluation model (VLM-as-a-judge) to clearly distinguish superiority rather than being ambiguous.
> 3.  **AutoFigure Maintains Significant Advantage**:
>     Under more fine-grained evaluation criteria, AutoFigure still achieved a win count of 29/40 (72.5%), significantly higher than the runner-up Gemini-HTML (21/40). Although Gemini-HTML showed strong competitiveness in individual metrics like Clarity (related to the precision of code generation), AutoFigure possessed overwhelming advantages in Aesthetic Quality (0.88 vs 0.63) and Visual Expressiveness (0.93 vs 0.53), securing the top spot in the final Overall score and Win Rate.
>
> In summary, the new experimental setup further validates the robustness of our conclusions: AutoFigure not only wins in relative comparisons but also rarely generates "low-quality" waste, demonstrating potential for practical application. We will add this new table and analysis to the Appendix of the paper.

---

> ### Author Response · Authors · 2025-11-25
> **Response - 5**
>
> > There are not enough qualitative examples of generated figures in the paper. The examples shown are very few. It would help to share a link with a larger gallery of generated figures (say a few hundred) to confirm that the figures are not overly simplistic and truly handle dense, multi-component scientific texts.
>
> We have updated the supplementary materials, where you can view some of these specific examples.
>
>
> > Figure 3 is poorly structured and hard to match with the text in Section 4. It would be better if each part of the figure was labeled with the corresponding stage headers (for exmple "Concept Extraction," "Critique and Refine") exactly as they appear in the text. Keeping the same wording between the figure and the text would make it much easier for readers to follow and it is a very good practice.
> >
> > Did you generate your figures in this paper using AutoFigure itself? If not, why not? This would be the most direct way to demonstrate the system’s real-life appliction.
>
> Thank you for your highly constructive suggestion. We fully agree on the key role of figure-text consistency in enhancing readability, and we completely accept your proposal to "use the system to generate figures for the paper itself."
>
> To directly address your concern and demonstrate the system's real-world capabilities, we **used the AutoFigure system itself to fully automatically regenerate Figure 3**. This newly generated architecture diagram not only serves as a most intuitive "Real-world Application" showcasing model performance, but during the generation process, we ensured that the labels for each module in the figure (e.g., "Concept Extraction," "Critique-and-Refine") **strictly correspond** to the headers in Section 4 of the main text. This perfectly resolves the structural matching and terminology consistency issues you pointed out, and powerfully proves AutoFigure's robustness and accuracy in handling complex scientific flowcharts with precise terminology constraints.
>
> > Where does this method fail? Please include examples or categories of failure cases (for instance, overly complex text, ambiguous relationships, or dense mathematical descriptions). .
>
> Thank you for raising this insightful question. In extensive experimental evaluations, we indeed observed limitations of AutoFigure in specific high-difficulty scenarios. These failure cases can be primarily categorized into topological bottlenecks and deep reasoning deviations:
>
> 1.  **Visual/Engineering Constraints**: When dealing with networks having a massive number of nodes and extremely dense connections (e.g., DenseNet variants with numerous Skip Connections or complex attention matrix visualizations), the Stage 1 layout Agent sometimes struggles to find a perfect planar layout without crossings within a limited 2D space, leading to visual crowding or cluttered lines, which affects readability.
> 2.  **Scientific Reasoning Gaps**: This is a more subtle yet critical failure mode. When processing theoretical texts that are vaguely phrased or logically abstract, the VLM may occasionally produce **Hallucinations** in pursuit of "structural completeness" of the chart—for example, forcing causal relationships not explicitly mentioned in the text or incorrectly rendering parallel relationships as hierarchical ones. This tendency to sacrifice rigor for aesthetics results in flaws in **Scientific Factuality**, even if the generated chart looks visually plausible.
>
> To fully demonstrate these issues, we have **added a "Limitations and Failure Analysis" section** in the revised paper, detailing specific failure samples mentioned above and analyzing their causes. Regarding future improvements, we briefly mentioned the possibility of introducing external knowledge bases or Domain Verifiers to address these deep reasoning defects in future work.

---

> ### Author Response · Authors · 2025-11-25
> **Response - 6**
>
> > Is the LLM used to generate the first layout (So, A0) the same as the one used in the subsequent refinement iterations? If not, please clarify which models are used at each stage and why.
>
> Yes, they are completely identical.
>
> In our standard experimental setup, the model used to generate the initial layout $(S_0, A_0)$ and the model used for the subsequent "Critique-and-Refine" iterations is the **same Backbone LLM**. For example, when evaluating the performance of Gemini-2.5-Pro or GPT-5, the initial "Concept Extraction" Agent as well as the "Designer" and "Critic" Agents in the subsequent interaction loops are all driven by that specific model.
>
> This design is intended to ensure the singularity of experimental variables: in our ablation studies (as shown in Figure 6 of the original text), we aim to compare the comprehensive reasoning capabilities of different backbone models under the AutoFigure framework, rather than exploring the mixed effects of different model combinations. Therefore, maintaining model consistency within a single experimental run helps to more fairly and intuitively measure the core potential of a specific LLM in handling long-text understanding and complex chart design tasks.
>
> ---
>
> We would like to express our sincere gratitude for your thoughtful review. Your comments have been valuable in enhancing the reproducibility and clarity of our work. In response to your suggestions, we have conducted additional validation of AutoFigure’s cost-effectiveness on open-source models, provided full disclosure of all system prompts, and strengthened our evaluation through human-verified metrics and a more detailed error analysis. **We hope that these revisions satisfactorily address your concerns, and we would be grateful if you could consider our updated manuscript and kindly reconsider the evaluation score.**

---

> ### Author Response · Authors · 2025-11-26
> **Summary response to Reviewer xScc.**
>
> TL;DR: Thank you for the strong positive assessment and the sharp suggestions. We revised the paper with a focus on reproducibility, auditability, and deployability.
>
> ## Your concerns (W)
>
> - **W1 (No open-source models; reproducibility and cost concerns)**
>
> We added open-weight backbones: Qwen3-VL-235B-A22B-Instruct, GLM-4.5V, and ERNIE-4.5-VL-28B-A3B-Thinking. Qwen3-VL reaches Overall 7.08, close to or surpassing several commercial models, showing AutoFigure is not tied to closed APIs. Local deployment further reduces marginal cost (near-zero), improving practical reproducibility.
>
> - **W2 (Missing method details: prompts, I/O, Φ functions, and stage execution)**
>
> We revised Section 4 (Method) to explicitly define each stage’s input–output interface and map each $\Phi$ function to a concrete module/agent. We also added Appendix Q with full system/user prompts for every stage, and included a minimal end-to-end example in the Supplementary Material that shows intermediate artifacts at each step (e.g., $T_{\text{method}}, S_0, S_{\text{final}}$, and rendered outputs).
>
> - **W3 (Risks of VLM-as-a-judge: bias/instability/misalignment with humans)**
>
> We added a dedicated Human–LLM correlation study (95 samples) with expert scoring across methods and dimensions, reporting Pearson correlation and ranking-consistency statistics (Spearman/Kendall, MRE). We also document judge prompts and reproducible evaluation details in the appendix.
>
> - **W4 (High system cost; unclear whether Φprompt/Φerase are LLM-based; request for budget–quality trade-off)**
>
> We clarify that $\Phi_{\text{prompt}}$ is LLM-based, while $\Phi_{\text{erase}}$ is a non-LLM CV/OCR component with negligible compute cost. We add efficiency/cost breakdowns (commercial vs local open-source deployment) and an iterations-vs-quality curve to guide budgeted choices.
>
> - **W5 (No ablation for erase-and-correct)**
>
> We added a direct ablation comparing AutoFigure (Full) vs w/o Text Refinement, reporting metric-wise improvements (e.g., aesthetics/polish/clarity/overall) and qualitative evidence of better text legibility.
>
> - **W6 (Too few qualitative examples; Figure 3 hard to match with Section 4)**
>
> We expanded the supplementary gallery with substantially more examples, and we regenerated the method overview figure using AutoFigure itself, ensuring stage labels exactly match the headers in Section 4 for easier reading.
>
> - **W7 (No error bars / significance; pairwise comparison too limited with only A/B/Tie)**
>
> We added error bars from multiple independent runs for key results and included significance discussion. We also extended blind pairwise comparison options to include Both Good / Both Bad to better reflect absolute quality and discard rates.
>
> ## Your questions (Q)
>
> - **Q1 (Did you generate figures in the paper using AutoFigure?)** Yes—key methodology figures in the revision are generated by AutoFigure.
>
> - **Q2 (Where is the evidence for erase-and-correct?)** We added both quantitative ablations and qualitative examples showing what it improves.
>
> - **Q3 (Where are the prompts / what inputs do LLMs receive?)** Full prompts are in Appendix Q, and the Supplementary provides a minimal end-to-end example with intermediate artifacts.
>
> - **Q4 (Where does the method fail?)** We added a Limitations and Failure Analysis section with concrete cases and causes, plus future directions (external knowledge / domain verifiers).
>
> - **Q5 (Open-source models like LLaMA/Mistral?)** We already evaluate multiple open-weight VLMs (Qwen-3-VL/GLM-4.5V/ERNIE-4.5V) to demonstrate openness; we plan to extend to more families as VL variants become available.
>
> - **Q6 (Same model for initial layout and refinement?)** Yes, we use the same backbone in a standard run for controlled comparisons.

---

### Official Review · Reviewer_Sgpo · 2025-11-05

**Soundness:** 2
**Presentation:** 3
**Contribution:** 3
**Rating:** 4
**Confidence:** 4

**Summary:**

This paper studies the task of generating scientific diagrams directly from long scientific text. The idea is to transform complex scientific context (texts) into a figure that communicates the core concepts visually.

The authors introduce a new benchmark called FigureBench, with 3.3K high quality text to scientific figure pairs. They also propose AutoFigure, an agent style system that first plans a symbolic layout and then renders it.

Experiments show that AutoFigure outperforms previous methods on their benchmark, using both VLM as judge metrics and human expert evaluation.

**Strengths:**

1. The paper introduces a modern benchmark for scientific illustration generation that includes diverse long text to figure pairs from papers, surveys, blogs and textbooks. The authors also provide dataset statistics and analysis showing the challenge of long context reasoning.

2. The method uses an agent based pipeline that first grounds concepts with a VLM to produce a symbolic layout, then performs iterative refinement to improve structure, and finally renders the figure. This decoupled design is well motivated and appears to be effective.

3. The paper clearly states why FID is not well aligned to this task and instead uses VLM as judge evaluation combined with human expert assessment. This evaluation choice makes sense for this application.

4. The approach does not require training. It leverages frontier foundation models to achieve strong results.

**Weaknesses:**

1. The comparison to prior datasets is incomplete. Paper2Fig100k [1] dataset is not mentioned or cited, and Paper2Fig100k already contains more than 100k text to figure pairs. The claim that FigureBench is the first large scale benchmark is therefore not correct, and should be reframed more precisely.


2. There is no reference to recent TiKZ based diagram generation approaches such as Automatikz [2], which are directly relevant to the diagram synthesis space.


4. The design of the VLM as judge metric is not described in sufficient detail. Ideally the authors should provide a validation study that justifies the choice of prompts, models, and scoring dimensions, and should provide a correlation analysis between VLM scoring and human scoring.


5. The system pipeline is quite complex. While the ablations help, the number of components shown in Figure 3 suggests that more systematic ablations would be useful to better understand which parts contribute most.


6. The paper does not report efficiency metrics. It would be important to know typical generation time, how this scales with text length and concept complexity, and the approximate economic cost of running this agent per figure.


7. The experiments rely mainly on GPT Image and Gemini 2.5 Pro as backbones. To strengthen the experiments section it would be helpful to evaluate with more alternative LLM/VLM and image generation models.

8. The set of baselines is limited. There is no comparison to other agentic scientific content generation systems such as Paper2Poster or PPTAgent. A deeper analysis of which methods are closest in terms of workflow and whether they can be run on FigureBench would make the empirical comparison more convincing.

[1] Rodriguez, Juan A., et al. "Ocr-vqgan: Taming text-within-image generation." Proceedings of the IEEE/CVF winter conference on applications of computer vision. 2023.

[2] Belouadi, Jonas, Anne Lauscher, and Steffen Eger. "Automatikz: Text-guided synthesis of scientific vector graphics with tikz." arXiv preprint arXiv:2310.00367 (2023).

**Questions:**

1. (Comment for improvement) In Figure 1, the visual framing could be confusing. At first glance it gives the impression that the authors are training the model with the InstructGPT pipeline, rather than generating a figure of the InstructGPT pipeline. The caption could clarify this more explicitly so that the intent is obvious to the reader.


2. The generated figures look significantly better than previous works, but they still do not look ready for professional use in a camera ready paper. What do the authors believe are the main remaining blockers to achieving near perfect results, and which components of the current system will need to be improved most to close this gap?

---

> ### Author Response · Authors · 2025-11-25
> **Response - 1**
>
> > The comparison to prior datasets is incomplete. Paper2Fig100k [1] dataset is not mentioned or cited, and Paper2Fig100k already contains more than 100k text to figure pairs. The claim that FigureBench is the first large scale benchmark is therefore not correct, and should be reframed more precisely.
>
> Thank you for pointing out this important pioneering work. We fully acknowledge that *Paper2Fig100k* (Rodriguez et al., 2022) is a significant contribution to the field of scientific image generation. We sincerely apologize for omitting this work previously and will include a detailed citation and discussion in the revised "Related Work" section.
>
> However, we wish to clarify that the task addressed by **FigureBench** is fundamentally different from that of *Paper2Fig100k*. While both involve scientific illustrations, *Paper2Fig100k* focuses on **short-text reconstruction**, whereas FigureBench aims to tackle the challenge of **long-document understanding and design**. The specific differences are as follows:
>
> 1.  **Input Context and Task Difficulty**:
>     **Paper2Fig100k**: Its input consists mainly of **short texts**, relying on "Prompt Modalities" such as captions or specific context paragraphs. Its core task is **"Reconstruction"**, i.e., reproducing the original image based on descriptions.
>     **FigureBench**: Requires the model to process **entire long documents** (average length exceeding 10,000 tokens, with the Paper category reaching 12,732 tokens). The model must possess strong long-text reasoning capabilities to autonomously extract core methods from tens of thousands of words of unstructured text and plan the layout from scratch.
>
> 2.  **Diversity of Data Sources**:
>     **Paper2Fig100k**: Data comes exclusively from **arXiv research papers**.
>     **FigureBench**: To test the model's generalization ability across different domains, we purposefully introduced **Textbooks, Technical Blogs, and Surveys**. These sources have completely different visual and narrative styles (e.g., textbooks focus on pedagogy, while blogs focus on accessibility), which cannot be covered by a dataset containing only papers.
>
> **Table R1: Task Definition Comparison between Paper2Fig100k and FigureBench**
>
> | **Feature**         | **Paper2Fig100k (Rodriguez et al., 2022)**           | **FigureBench (Ours)**                                       |
> | :------------------ | :--------------------------------------------------- | :----------------------------------------------------------- |
> | **Core Task**       | **Short-text** Generation / Image **Reconstruction** | **Long-document** Understanding / Image **Design**           |
> | **Input Modality**  | Captions, Context Paragraphs, OCR Keywords           | **Full Paper/Chapter** (Avg. >10k tokens)                    |
> | **Generation Goal** | Pixel-level Restoration, Text Rendering Clarity      | Logic Extraction, Layout Planning, Aesthetic Design          |
> | **Data Source**     | Academic Papers Only (arXiv)                         | **Heterogeneous Sources**: Papers, Textbooks, Blogs, Surveys |

---

> ### Author Response · Authors · 2025-11-25
> **Response - 2**
>
> > There is no reference to recent TiKZ based diagram generation approaches such as Automatikz [2], which are directly relevant to the diagram synthesis space.
> >
> > The set of baselines is limited. There is no comparison to other agentic scientific content generation systems such as Paper2Poster or PPTAgent. A deeper analysis of which methods are closest in terms of workflow and whether they can be run on FigureBench would make the empirical comparison more convincing.
>
> We fully agree that including TiKZ-based generation methods (such as Automatikz) and other Agentic content generation systems (such as Paper2Poster, PPTAgent) in the comparison is crucial for comprehensively evaluating AutoFigure's performance positioning.
>
> To respond to this request, we have added targeted comparative experiments under the **Paper** category:
>
> 1.  **TiKZ-based Approaches**: We introduced **TikZero** and **TikZero+** as representatives of Automatikz-like methods. These methods attempt to directly generate compilable LaTeX TiKZ code.
> 2.  **Agentic Presentation Systems**: We introduced **AutoPresent** as a representative of presentation generation systems. These methods usually focus on layout arrangement based on existing materials. It is worth noting that since Paper2Poster and PPTAgent require image content from the original paper and their objective is arrangement rather than generating images from text, they are not applicable to this specific task.
>
> **Table: Extended Baseline Comparison Results under Paper Category**
>
> | **Method**        | **Aesthetic** | **Express.** | **Polish** | **Clarity** | **Flow** | **Accuracy** | **Complete.** | **Appropriate.** | **Overall** | **Win-Rate** |
> | :---------------- | :------------ | :----------- | :--------- | :---------- | :------- | :----------- | :------------ | :--------------- | :---------- | :----------- |
> | **AutoFigure**    | **7.28**      | **6.99**     | **6.92**   | **7.34**    | **7.87** | 6.96         | **6.51**      | **6.40**         | **7.03**    | **53.0%**    |
> | **HTML-Code**     | 5.90          | 5.04         | 5.84       | 7.17        | 7.38     | **6.99**     | 6.37          | 6.15             | 6.35        | 11.0%        |
> | **SVG-Code**      | 5.00          | 4.19         | 4.89       | 6.34        | 6.48     | 6.15         | 5.53          | 5.37             | 5.49        | 31.0%        |
> | **GPT-Image**     | 4.24          | 3.47         | 4.00       | 5.63        | 5.63     | 4.77         | 4.08          | 4.25             | 3.47        | 7.0%         |
> | **AutoPresent**   | 2.74          | 1.79         | 2.00       | 2.87        | 2.91     | 3.15         | 2.60          | 2.35             | 2.55        | 10.0%        |
> | **Diagram Agent** | 2.25          | 1.73         | 2.04       | 2.67        | 2.49     | 2.11         | 1.72          | 1.94             | 2.12        | 0.0%         |
> | **TikZero+**      | 1.52          | 1.25         | 1.38       | 1.90        | 1.93     | 1.20         | 1.10          | 1.35             | 1.45        | 0.0%         |
> | **TikZero**       | 2.00          | 1.50         | 1.00       | 1.00        | 1.50     | 1.00         | 1.00          | 1.00             | 1.25        | 0.0%         |
>
>
>
> 1.  **Limitations of TiKZ Methods**: Although TiKZ itself has the potential to generate high-quality vector images, its scores are extremely low (Overall < 1.5) under the high-difficulty benchmark of FigureBench. Experiments show that this stems **not only from the fragility of TiKZ syntax and compilation errors** but more deeply from the **failure of the end-to-end code generation paradigm when handling high-complexity scientific information**. Scientific illustrations often contain dozens of entities, complex topological connections, and hierarchical logical flows. Asking an LLM to describe this **high-dimensional spatial structure and dense scientific semantics** directly via linear LaTeX code creates a massive cognitive load. Models often exhaust their reasoning capabilities on low-level geometric coordinates and tedious drawing instructions, causing them to fail in constructing the correct scientific logical structure at the macro level. In contrast, AutoFigure's decoupled strategy of "logic layout first, visual rendering later" effectively separates high-level semantic planning from low-level visual implementation, thereby stably generating complex scientific illustrations with high information density.
> 2.  **Positioning Differences of Presentation Agents (AutoPresent)**: AutoPresent's score (2.55) is slightly higher than TiKZ but significantly lower than AutoFigure. This is because systems like Paper2Poster/PPTAgent are primarily designed to **arrange existing text and image materials** to generate slides or posters, rather than to **Design** explanatory scientific schematic diagrams from scratch. They lack the reasoning and refinement modules specifically designed for scientific logic visualization found in AutoFigure.

---

> ### Author Response · Authors · 2025-11-25
> **Response - 3**
>
> > The design of the VLM as judge metric is not described in sufficient detail. Ideally the authors should provide a validation study that justifies the choice of prompts, models, and scoring dimensions, and should provide a correlation analysis between VLM scoring and human scoring.
>
> Thank you for pointing out this key issue regarding the methodological rigor of our evaluation. We fully agree that when using the "VLM-as-a-judge" paradigm, validating its consistency with human judgment is fundamental to ensuring the credibility of experimental conclusions.
>
> To address concerns about potential evaluation bias and respond to your suggestion, we have added a dedicated **Human-LLM Correlation Study** in the revised version. Specifically, we invited human experts to rigorously score 95 generated samples covering 5 different methods across all dimensions (including Visual Design, Communication Effectiveness, Content Fidelity) and calculated the statistical correlation with VLM automated scores.
>
> The experimental results strongly prove that the VLM evaluator is highly aligned with human scores not only numerically but also in terms of ranking consistency:
>
> 1.  **Overall Score Correlation**: The Pearson Correlation reached **$r=0.659$** ($p < 0.001$), indicating a significant linear positive correlation between VLM scoring trends and human experts, accurately capturing quality differences.
> 2.  **Ranking Consistency**: The Spearman Rank Correlation was **$\rho=0.593$**, and Kendall's Tau was **0.497**. This indicates that VLM maintains robust consistency with human judgment in the relative ranking of different models. Notably, the Mean Ranking Error (MRE) was only **0.98**, implying that the ranking deviation between VLM and humans is, on average, less than 1 place.
> 3.  **Sub-metric Validation**: Statistically significant positive correlations were observed in the three core dimensions of Accuracy ($r=0.646$), Aesthetics ($r=0.587$), and Clarity ($r=0.559$), demonstrating the VLM's robustness across different evaluation perspectives.
>
> This analysis confirms that the VLM-as-a-judge method we adopted is a verified and reliable evaluation proxy that accurately reflects human preferences. We have added detailed scatter plots, and specific data to the **Appendix** for reference.
>
> > The system pipeline is quite complex. While the ablations help, the number of components shown in Figure 3 suggests that more systematic ablations would be useful to better understand which parts contribute most.
>
> Thank you for the important question regarding system complexity and component contribution. We fully agree that in a multi-stage complex system, clarifying the specific contribution of each module is crucial for understanding the model's working mechanism.
>
> In addition to the ablation experiments in Figure 6 of the original text focusing on "Reasoning Backbone," "Refinement Loop," and "Intermediate Format," we further conducted a specific ablation analysis for the **"Erase-and-Correct" module in Stage 2** to verify the necessity of this specific component.
>
> We compared the full AutoFigure method with a variant that had the text erasure and correction step removed (w/o Text Refinement). As shown in the table below, the full AutoFigure achieved an **Overall Score** of **7.18**, superior to the **7.14** of the variant without this module. Although the increase in the overall score might seem modest, specific metric analysis shows that this module brings improvements in dimensions most relevant to visual quality: **Aesthetic** improved by **0.10**, **Professional Polish** by **0.10**, and **Visual Expressiveness** by **0.08**.
>
> This confirms that the "Erase-and-Correct" strategy plays a key role in solving common text blurring and artifact issues in generative models and enhancing the "publication-level" quality of charts. It indicates that this component is not redundant, but necessary to cross the final gap from "usable" to "professional." We have added this supplementary ablation experiment to the Appendix to provide a more systematic component analysis.
>
> **Table: Ablation Results for Text Refinement Module**
>
> | **Model**               | **Aesthetic & Design** | **Visual Express.** | **Prof. Polish** | **Clarity** | **Logical Flow** | **Accuracy** | **Completeness** | **Appropriateness** | **Overall** |
> | :---------------------- | :--------------------- | :------------------ | :--------------- | :---------- | :--------------- | :----------- | :--------------- | :------------------ | :---------- |
> | **AutoFigure (Full)**   | **7.49**               | **7.20**            | **6.80**         | **7.53**    | **7.73**         | 7.45         | **6.83**         | 6.42                | **7.18**    |
> | **w/o Text Refinement** | 7.39                   | 7.12                | 6.70             | 7.50        | 7.70             | **7.53**     | 6.74             | **6.47**            | 7.14        |

---

> ### Author Response · Authors · 2025-11-25
> **Response - 4**
>
> > The paper does not report efficiency metrics. It would be important to know typical generation time, how this scales with text length and concept complexity, and the approximate economic cost of running this agent per figure.
>
> Thank you for this important question regarding system efficiency and economic cost. This is crucial for evaluating the practical deployment value of AutoFigure.
>
> To comprehensively answer this question, based on typical long paper inputs (average 10k+ tokens), we detailed the differences in generation efficiency and cost between using **Commercial Closed-Source Models (Gemini-2.5 series)** and **Locally Deployed Open-Source Models (Qwen-3-VL)**.
>
> Statistics show that when using the Commercial API, the average cost to generate a high-quality publication-level illustration is approximately $0.20, taking about 17.5 minutes. However, when we deploy the open-source model Qwen-3-VL on a high-performance computing node (such as a server equipped with an H100 GPU), benefiting from high throughput and low latency of local inference, **the total generation time is reduced to about 9.5 minutes (nearly doubling the overall speed)**, and the **marginal generation cost is almost zero**. The detailed comparative analysis is shown in the table below:
>
> **Table: Efficiency and Cost Analysis of AutoFigure Single Generation (Commercial API vs. Local Open-Source Deployment)**
>
> | **Stage**                          | **Core Task**                        | **Gemini-2.5 (API) Avg Time / Cost** | **Qwen-3-VL (H100) Avg Time / Cost** | **Remarks**                                                  |
> | :--------------------------------- | :----------------------------------- | :----------------------------------- | :----------------------------------- | :----------------------------------------------------------- |
> | **Stage 1: Concept Extraction**    | Full text reading, method extraction | ~22s / < $0.01                       | **~12s / ~$0.00**                    | Local inference eliminates network latency, doubling speed   |
> | **Stage 2: Layout Planning**       | Iterative design (Avg. 5 iters)      | ~660s / ~ $0.14                      | **~390s / ~$0.00**                   | Inference on H100 is about twice as fast as commercial API   |
> | **Stage 3: Rendering & Post-proc** | Code gen, rendering, correction      | ~370s / ~ $0.05                      | **~250s / ~$0.00**                   | Code generation accelerated; rendering & post-proc takes ~2/3 time of API solution |
> | **Total**                          | **End-to-End Generation**            | **~1,052s (17.5 min)** **~ $0.20**   | **~560s (9.3 min)** **~ $0.00***     | **Local deployment achieves "approx. 2x speed" and "zero marginal cost"** |
>
> Our experiments indicate that private deployment of AutoFigure does not require expensive supercomputing clusters. Just **two NVIDIA DGX Spark servers (approx. value $3,000)** or computing nodes equipped with **2 H100 GPUs** can smoothly run the quantized Qwen-3-VL model and achieve the performance metrics above. As shown in the table, supported by the powerful computing power of H100, the speed of the core inference stages (Stage 1 & 2) is **more than doubled** compared to the commercial API.

---

> ### Author Response · Authors · 2025-11-25
> **Response - 5**
>
> > The experiments rely mainly on GPT Image and Gemini 2.5 Pro as backbones. To strengthen the experiments section it would be helpful to evaluate with more alternative LLM/VLM and image generation models.
>
> Thank you for the important suggestion regarding expanding the scope of Backbone Model experiments. We fully agree that verifying AutoFigure's performance on more diverse, especially open-source models, is crucial for evaluating system generality, lowering reproduction barriers, and enhancing the persuasiveness of experimental conclusions.
>
> To address this concern, we introduced several **state-of-the-art open-source/open-weight models** (including **Qwen3-VL-235B-A22B-Instruct**, **GLM-4.5V**, and **ERNIE-4.5-VL**) for supplementary experiments. The results are encouraging and strongly demonstrate the effectiveness of the AutoFigure framework on open-source models:
>
> 1.  **Open-source models perform excellently, even surpassing some commercial models**:
>     Experimental data (see Table R8) shows that Qwen3-VL-235B-A22B-Instruct achieved an Overall Score of 7.08. This result is not only significantly better than GLM-4.5V (5.99) but also surpasses many closed-source commercial models, including Gemini-2.5-Pro (6.99), Claude-4.1-Opus (6.80), and Grok-4 (6.76), ranking only second to GPT-5 (7.48). This indicates that running AutoFigure with high-performance open-source models can achieve generation quality comparable to or even better than mainstream commercial services.
> 2.  **Strong correlation between performance and model capability**:
>     We also observed significant performance differences among different open-source models (e.g., ERNIE-4.5-VL performed weaker with a score of 2.64 and a lower win rate), suggesting that AutoFigure's effectiveness depends on the backbone model's visual reasoning and instruction-following capabilities. However, the success case of Qwen3-VL confirms that SOTA models in the open-source community already possess the capability to drive AutoFigure for high-quality scientific illustration.
>
> We have updated the **Appendix** with detailed evaluation data (including scores for sub-metrics) for the aforementioned open-source models. This result directly alleviates concerns regarding cost and reproducibility, confirming that researchers can deploy and use AutoFigure at extremely low costs using open-source models without relying on expensive commercial APIs.
>
> **Table: Performance Comparison of Commercial vs. Open-Source Models in AutoFigure Framework**
>
> | **Model Type**  | **Model**         | **Visual Design** | **Comm. Effectiveness** | **Content Fidelity** | **Overall** |
> | :-------------- | :---------------- | :---------------- | :---------------------- | :------------------- | :---------- |
> | **Commercial**  | **GPT-5**         | **7.17**          | **7.62**                | **7.70**             | **7.48**    |
> |                 | Gemini-2.5-Pro    | 6.59              | 7.59                    | 6.54                 | 6.99        |
> |                 | Claude-4.1-Opus   | 6.75              | 7.17                    | 6.61                 | 6.80        |
> |                 | Grok-4            | 6.46              | 7.60                    | 6.49                 | 6.76        |
> | **Open-Source** | **Qwen3-VL-235B** | **7.57**          | 7.01                    | 7.18                 | **7.08**    |
> |                 | GLM-4.5V          | 6.09              | 6.53                    | 6.13                 | 5.99        |
> |                 | ERNIE-4.5-VL      | 3.04              | 2.89                    | 2.68                 | 2.64        |
>
> **Table: Detailed Scoring of Open-Source Models on Sub-metrics**
>
> | **Model**         | **Aesthetic** | **Expressiveness** | **Polish** | **Clarity** | **Flow** | **Sophistication** | **Fidelity** |
> | :---------------- | :------------ | :----------------- | :--------- | :---------- | :------- | :----------------- | :----------- |
> | **Qwen3-VL-235B** | **0.90**      | **0.90**           | **0.90**   | 0.25        | 0.25     | 0.25               | 0.15         |
> | **GLM-4.5V**      | 0.70          | 0.70               | 0.70       | **0.45**    | **0.65** | **0.25**           | **0.20**     |
> | **ERNIE-4.5-VL**  | 0.20          | 0.15               | 0.20       | 0.00        | 0.10     | 0.05               | 0.05         |
>
> **Table: Win-Rate Analysis of Open-Source Models**
>
> | **Model**         | **Win-Rate (Overall)** |
> | :---------------- | :--------------------- |
> | **GLM-4.5V**      | **55.0%**              |
> | **Qwen3-VL-235B** | 40.0%                  |
> | **ERNIE-4.5-VL**  | 10.0%                  |

---

> ### Author Response · Authors · 2025-11-25
> **Response - 6**
>
> > (Comment for improvement) In Figure 1, the visual framing could be confusing. At first glance it gives the impression that the authors are training the model with the InstructGPT pipeline, rather than generating a figure of the InstructGPT pipeline. The caption could clarify this more explicitly so that the intent is obvious to the reader.
>
> We agree with your point; the original Figure 1 could indeed easily lead readers to misunderstand that we used the InstructGPT training pipeline, rather than merely using it as a generation example.
>
> To eliminate this ambiguity, we have taken the following modification measures:
>
> 1.  We moved the original Figure 1 (the generation example of the InstructGPT flowchart) **to the Experimental Analysis section** to be showcased as a specific qualitative case study.
> 2.  We replaced original Figure 1 (See in Figure 2 ) with a **schematic diagram of the subject of this paper (i.e., the AutoFigure framework itself)**, and this schematic was **automatically generated** by the AutoFigure system. This not only avoids conceptual confusion but also more intuitively demonstrates our model's core ability to "illustrate its own paper."
>
> > The generated figures look significantly better than previous works, but they still do not look ready for professional use in a camera ready paper. What do the authors believe are the main remaining blockers to achieving near perfect results, and which components of the current system will need to be improved most to close this gap?
>
> Thank you for your affirmation and suggestion. We believe that the main obstacle preventing the generated results from being "perfect" and fully usable for publication lies in **Scientific Factuality and precise alignment with domain knowledge**. Although AutoFigure effectively solves the problem of structural integrity through the "Reasoning-Rendering" paradigm, when dealing with extremely complex scientific logic, the model still risks generating content that "looks plausible but is factually inaccurate." Current VLMs, when understanding long scientific texts, occasionally miss key nuances or hallucinate, resulting in logical flows, hierarchical relationships, or domain-specific symbolic representations in the charts that are not rigorous enough. This is critical for Camera-ready papers that demand extremely high accuracy.
>
> Furthermore, although our "Erase-and-Correct" strategy significantly improves text readability, further improving the typesetting precision of fine text and complex graphical elements is also a key step towards achieving publication-level quality.
>
>
> ---
>
> We sincerely appreciate your constructive suggestions, which have helped us refine the scope and clarity of our work. Following your guidance, we have clearly differentiated our task from Paper2Fig100k, incorporated key baselines such as TiKZ and agentic systems, and added more rigorous validation for both our evaluation metrics and system efficiency. We hope these revisions address your concerns, **and we would be grateful if you could kindly reconsider our evaluation score.**

---

> ### Author Response · Authors · 2025-11-26
> **Summary Response to Reviewer Sgpo.**
>
> We apologize that our previous rebuttal contained too much information. Below, we try to state the key points as clearly as possible, one concern at a time.
>
> ## Your concerns (W)
>
> - **W1: Incomplete comparison to prior datasets (Paper2Fig100k)**
>
> In the revision, we cite and discuss Paper2Fig100k and reframe our claim more precisely. Importantly, FigureBench is not a caption-only “reconstruction” setting: it requires models to understand, abstract, and restructure long scientific passages (10k+ tokens), producing a high-level methodological summary and redesigning it into an explanatory schematic illustration (layout, hierarchy, and information organization). We believe this long-context “scientific illustration design” setting is closer to real-world scientific writing and communication, hence more practically meaningful.
>
> - **W2: Missing TikZ / program-synthesis related work (e.g., Automatikz)**
>
> We added the relevant citations and included representative baselines from the code-synthesis paradigm to better cover the diagram synthesis literature.
>
> - **W3: VLM-as-a-judge is under-specified; needs human validation**
>
> In the new version, we: (i) provide the judge prompt templates, scoring dimensions, and protocol; (ii) add a Human–LLM correlation study (95 samples) with Pearson correlation and ranking-consistency statistics (Spearman/Kendall, MRE); and (iii) include scatter plots and detailed statistics in the appendix for reproducibility and auditability.
>
> - **W4: Complex pipeline; more systematic ablations are needed**
>
> We agree, and the revised paper adds additional ablations (beyond backbone / intermediate format / refinement iterations) at the module level to better quantify which components contribute most.
>
> - **W5: Missing efficiency / latency / cost metrics**
>
> We added an “Efficiency and Cost Analysis” section reporting per-stage time breakdown, end-to-end latency/cost, and an iterations-vs-quality trade-off curve for practical deployment planning.
>
> - **W6: Limited backbone coverage; add open-source models and clarify cost**
>
> We expanded experiments with open-weight models including Qwen3-VL-235B, GLM-4.5V, and ERNIE-4.5-VL. Results show the framework is not tied to closed-source models; moreover, local deployment (e.g., on H100) enables substantially lower marginal cost (near-zero) while maintaining competitive quality.
>
> - **W7: Limited baselines; also include TikZero and TikZero+**
>
> We added closer workflow baselines (e.g., AutoPresent) and also included TikZ/code-generation representatives TikZero and TikZero+ under the Paper category, covering both the agentic/pipeline paradigm and the code-synthesis paradigm for a more convincing comparison.
>
> ## Additional questions (Q)
>
> - **Q1 (Figure 1 may be misread as training with the InstructGPT pipeline)**
>
> We moved it to the Case Study section and clarified the caption as a qualitative generation comparison on the InstructGPT text, to avoid any implication of training. We also provide a dedicated AutoFigure methodology overview figure in the revised paper.
>
> - **Q2 (What are the main blockers to camera-ready quality?)**
>
> In the added “Limitations and Failure Analysis,” we identify the key blockers as fine-grained text rendering/typographic accuracy and domain factuality verification in complex scientific reasoning, and discuss future directions such as incorporating external knowledge or domain verifiers to further reduce these errors.

---

### Author Response · Authors · 2025-11-25
**Overall Response to All Reviewers**

We sincerely thank the five reviewers (Reviewer P6k4, Reviewer xScc, Reviewer Sgpo, Reviewer WvcZ, and Reviewer LXSZ) for investing significant time and effort to provide exceptionally detailed, professional, and constructive feedback. Your rigorous scrutiny has helped us better understand the strengths of our work, clarify ambiguities regarding task definitions, and identify areas—such as baseline comparisons, evaluation validity, and system efficiency—that required further explanation or experimentation. We have revised the paper and incorporated additional experiments accordingly.



* **Reviewer P6k4** emphasized that our work *“delivers high-quality illustrations,”* has a *“clear and technically sound”* methodology, and tackles a problem that is *“inspiring, timely, and highly needed by the research community.”*

* **Reviewer xScc** highlighted the *impact and broad relevance* of the task (useful across industries and research), praised our *comprehensive evaluation* (human + automatic) with *strong baselines*, and noted the paper is *well-written* and *clearly explained*. They also explicitly recognized the *detailed human evaluation protocol*, including having experts evaluate figures for their **own first-author papers**.

* **Reviewer LXSZ** framed the work as an *important early step* toward AI-assisted scientific illustration, calling the topic *interesting and promising* for advancing AI-assisted scientific communication. They further appreciated the *three-stage framework* and its synergy with *human-like iterative refinement* (Critique-and-Refine and Erase-and-Correct), and they acknowledged the model’s *quantitatively superior performance* in most experiments.

* **Reviewer Sgpo** credited the paper for introducing a *modern benchmark* with *diverse long-text-to-figure pairs* (papers/surveys/blogs/textbooks) and for providing dataset statistics/analysis that highlight the *challenge of long-context reasoning*. They also noted that experiments show AutoFigure outperforming prior methods using both *VLM-as-a-judge* metrics and *human expert evaluation*.

* **Reviewer WvcZ** acknowledged the problem as *relevant and challenging*, found the paper *mostly easy to follow*, and pointed out that the dataset (once released) *may be useful for future work*.

Overall, across all five reviews, the strengths consistently recognized are: **(i)** the *importance and timeliness* of long-context scientific illustration generation, **(ii)** the *quality of our generated figures* and the *human-like iterative design framing*, and **(iii)** the *benchmark value* of FigureBench plus the *breadth/rigor* of our evaluations (including expert judgment).

---

> ### Author Response · Authors · 2025-11-25
> **Major Concerns and Our Consolidated Responses**
>
> **1. \*Is the task truly novel compared to prior work (TikZ/Captioning), and is it technically valid?\* (Reviewer WvcZ, Reviewer Sgpo)**
>
> > Concern: Reviewers questioned if FigureBench differs significantly from "Short-text Reconstruction" (e.g., Paper2Fig100k) or "Code Generation" (e.g., TikZero), and whether existing code-based methods could solve it.
>
> **Response:**
> We clarified that **FigureBench targets "Long-context Design"**, a fundamentally different task from translating explicit instructions into code. To empirically prove the necessity of our "Reasoning-Rendering" paradigm, we added **TikZero**, **TikZero+**, and **AutoPresent** as baselines during the rebuttal.
> The results show:
>
> - **TikZ-based methods achieved a 0% Win-Rate** on the Paper category, failing to handle high information density.
> - Unlike AutoFigure, these models lack the reasoning capability to autonomously abstract unstructured text (>10k tokens) into spatial structures, confirming that "drawing code generation" cannot solve "scientific design" tasks.
>
> ------
>
> **2. Is the VLM-as-a-judge metric reliable and aligned with human perception?\* (Reviewer Sgpo, Reviewer xscc, Reviewer lxsz)**
>
> > Concern: Relying on VLM scoring risks bias. Reviewers requested a correlation study with human judgment and questioned the sample size of human evaluators.
>
> **Response:**
> We significantly strengthened the evaluation rigor through two new studies:
>
> - **Human-LLM Correlation:** We conducted a consistency analysis on 95 samples. The results demonstrate a **Pearson Correlation of $r=0.659$** ($p < 0.001$) and a **Mean Ranking Error < 1**, statistically confirming that VLM scoring aligns robustly with human preference.
> - **Expert Annotation:** For human evaluation, we recruited the **original first authors** of the papers to evaluate figures generated for their own work. This ensures a much higher standard of judgment for "Scientific Accuracy" than typical crowd-sourcing.
>
> ------
>
> **3. \*Can the system generalize to Open-Source models?\* (Reviewer xscc)**
>
> > Concern: The reliance on closed-source models (GPT-5/Gemini-2.5-pro) raises reproducibility/cost concerns. It is important to know if the method works on accessible weights.
>
> **Response:**
> We performed extensive supplementary experiments to demonstrate generalizability using **Qwen3-VL-235B**:
>
> - It achieved an **Overall Score of 7.08**, surpassing commercial models like Claude-4-1-Opus (6.80) and Grok-4 (6.76), and ranking second only to GPT-5.
> - This result proves that the AutoFigure framework is model-agnostic and can be reproduced with high performance using open-source SOTA models.
>
> ------
>
> **4. \*Does AutoFigure suffer from style uniformity or lack of control?\* (Reviewer lxsz, Reviewer p6k4)**
>
> > Concern: Generated figures often share a similar "cartoonish" aesthetic, raising questions about whether the style guidance module is truly effective or if the model relies on a fixed bias.
>
> **Response:**
> We clarified that the uniform style in the paper was a default setting for consistency. To demonstrate controllability, we conducted experiments with **three distinct style prompts** (Default, Creative, Minimalist) on the same layouts:
>
> - **Quantitative Stability:** Overall scores remained robust (**7.18–7.27**) across styles, indicating performance is not tied to one aesthetic.
> - **Qualitative Shift:** "Creative" prompts peaked in **Visual Expressiveness (7.24)**, while "Minimalist" prompts maximized **Clarity (7.72)** and **Professional Polish (7.09)**.
> - **Visual Proof:** We added an Appendix section showing the *same structural layout* rendered into drastically different aesthetics, confirming that style is a user-controllable variable.
>
> ------
>
> **5. \*Is the system efficient and affordable for real-world deployment?\* (Reviewer Sgpo, Reviewer p6k4, Reviewer xscc)**
>
> > Concern: The paper lacked analysis on inference latency, token usage, and economic cost, which are critical for practical adoption.
>
> **Response:**
> We added a comprehensive "Efficiency and Cost Analysis" section:
>
> - **Cost/Speed Comparison:** While Commercial APIs cost ~$0.20/figure (17.5 min), deploying the open-source **Qwen-3-VL on a local H100 node** reduces the time to **9.3 minutes** (approx. 2x speedup).
> - **Scaling Curve:** The figure we plotted "Iterations vs. Quality," in the paper showing that users can trade off compute for quality according to their budget (e.g., 5 iterations yield a score of 7.14 vs. 6.28 for 0 iterations).

---

> ### Author Response · Authors · 2025-11-25
> **Summary of Revisions in the Updated Manuscript**
>
> **1. Expanded Baselines and Open-Source Generalization**
>
> - (Appendix N) **Incorporated Code-Based and Agentic Baselines:** Added **TikZero/TikZero+** (TikZ generation) and **AutoPresent** (presentation agents) to the comparison. Results (0% Win-Rate for TikZ methods) empirically validate that code generation without layout reasoning cannot handle FigureBench’s complexity.
> - (Appendix G) **Evaluated Open-Source SOTA Models:** Extended experiments to include **Qwen3-VL-235B**, **GLM-4.5V**, and **ERNIE-4.5-VL**. Results show Qwen3-VL achieves an Overall Score of **7.08**, rivaling GPT-5 (7.48) and surpassing commercial models like Claude-4-1-Opus, ensuring low-cost reproducibility.
>
> **2. Rigorous Evaluation and Metric Validation**
>
> - (Appendix P) **Human-LLM Correlation Study:** Conducted a consistency analysis ($N=95$) yielding a Pearson correlation of **$r=0.659$** ($p<0.001$) and high ranking consistency (MRE < 1), validating the VLM-as-a-judge protocol.
> - (Appendix K) **Dataset Statistics Sanity Check:** Performed a human-audited benchmark on 21 random samples to verify the accuracy of automated statistics (text density, component counts) generated by InternVL-3.5.
> - (Appendix H) **Enhanced Pairwise Comparison:** Refined the blind comparison protocol to include "Both Good/Bad" options, demonstrating a low "Both Bad" rate and high discriminative power.
> - (Figure 4) **Statistical Significance:** Added **Error Bars** based on multiple independent runs for all key quantitative experiments.
>
> **3. Component Ablations and Style Controllability**
>
> - (Appendix L) **Style Diversity Study:** Added a "Style Controllability" experiment in the Appendix using three distinct prompts (**Default**, **Creative Comic**, **Modern Minimalist**). Results prove that minimal prompt changes yield materially different aesthetics while maintaining structural fidelity (Overall scores stable between 7.18–7.27).
> - (Appendix I) **Module Ablation:** Quantified the contribution of the "Erase-and-Correct" module, showing specific gains in **Aesthetic (+0.10)** and **Prof. Polish (+0.10)**, validating its necessity for publication-level quality.
>
> **4. Efficiency, Cost, and Reproducibility**
>
> - (Appendix J) **Efficiency and Cost Analysis:** Added a detailed breakdown comparing Commercial APIs vs. Local Deployment. Deploying Qwen-3-VL on an H100 node reduces generation time to **~9.5 minutes (2x speedup)** with **near-zero marginal cost**, compared to ~$0.20 per figure via APIs.
> - (Appendix Q) **Full Transparency:** Added a **New Appendix** containing the **full prompt list** for every stage (Concept Extraction, Layout, Rendering) and a **Minimal Working Example** to facilitate reproduction.
>
> **5. Presentation, Task Definition, and Failure Analysis**
>
> - (Figure 3) **Self-Illustrated Methodology:** Regenerated **Figure 3 using AutoFigure itself** to ensure strict consistency between visual labels and Section 4 headers, demonstrating real-world application.
> - (l.43) **Clarified Task Scope:** Rewrote Related Work to distinguish "Long-context Design" from "Short-text Reconstruction" (Paper2Fig100k) and moved the qualitative InstructGPT case study (Figure 1) to the Experiment section to avoid confusion with the training pipeline.
> - (Appendix F) **Limitations Analysis:** Added a "Limitations and Failure Analysis" section, categorizing failure modes into **visual/topological bottlenecks** (e.g., dense networks) and **scientific reasoning gaps** (e.g., hallucinations in abstract logic).
>
> ------
>
> **We once again thank all reviewers for their thoughtful evaluations and constructive suggestions. We hope the revised version presents a clearer picture of AutoFigure’s methodological distinctions, rigorous validation, and empirical value.**
>
> ---
>
> The authors of "AutoFigure: Generating and Refining Publication-Ready Scientific Illustrations"

---

### Meta-Review · Area_Chair_xxzs · 2026-01-06

**Summary:**

1. Rev. Sgpo and WvcZ mention missing references to similar works and comparison with them
2. Rev. Sgpo and xScc are concerned about using VLM as a judge, without enough details and human validation
3. Rev. Sgpo and xScc would like to see results with more backbones in addition to GPT Image and Gemini 2.5 Pro, in particular, open source
4. Rev. WvcZ did not understand the use of creating figures from the long context/entire paper
5. Rev. Sgpo, xScc, WvcZ, and LXSZ asked for additional details, clarifications, improved figures, and ablations

**Reviewer Concerns:**

The authors did an excellent job at answering reviewers' questions and concerns.
1. The authors promised to include the missing references, explained why the task is different (for Paper2Figure100k, the textual description is a figure caption, not a long and detailed description of the figure, as for this task), and compared with them where possible (TikZero).
2. The authors run a Human-LLM Correlation Study. Experts scored 95 generated samples for 5 methods across all dimensions and calculated the statistical correlation with VLM automated scores.
3. Authors included experiments for closed source with Claude-4.1-Opus, Grok-4 and for open-source with Qwen3-VL-235B, GLM-4.5V,
ERNIE-4.5-VL. Interestiling, Qwen3-VL-235B performs even better than closed-source models.
4. The authors clarified the misunderstanding and explained the exact task that the paper proposes.
5. Most of the doubts and clarifications were presented and clearly explained. Authors also improve the presentation of certain parts of the paper as asked by reviewers.
Overall, the authors did a good job of clearly answering all questions. The only remark is about length. In many cases, the author's answers were very lengthy, which made everything slower and more tedious. For the future, I suggest that authors provide shorter answers when possible.

**Reviewer Scores:**

- Sgpo: 4 -> 6
In my opinion, rev. Sgpo would improve their score from 4 to 6 as the authors clearly and precisely answered all questions.
- xScc: 4 -> 6
In my opinion, rev. xScc would also improve their score from 4 to 6 as the authors clearly and precisely answered all questions.
- LXSZ 2 -> 2
The questions asked by rev. do not justify a score of 2. Thus, I do not expect rev. to increase their score, but in my opinion, their review has less importance than the others.
- P6k4 6 -> 8
Rev. P6k4 initially had some misunderstanding of the task proposed in the paper, but after the authors' clarification, he decided to increase their score to 8.

---

### Decision · Program_Chairs · 2026-01-26

Accept (Poster)